**RESEARCH**

# Dynamic co-evolution of transposable elements and the piRNA pathway in African cichlid fishes

Miguel Vasconcelos Almeida[1,2]* , Moritz Blumer[3†], Chengwei Ulrika Yuan[1,2,3†], Pío Sierra[3], Jonathan L. Price[1,2], Fu Xiang Quah[1,3], Aleksandr Friman[4,5], Alexandra Dallaire[1,2,6], Grégoire Vernaz[2,3,7], Audrey L. K. Putman[1,2,3], Alan M. Smith[8], Domino A. Joyce[8], Falk Butter[9,10], Astrid D. Haase[4], Richard Durbin[3,11], M. Emília Santos[12] and Eric A. Miska[1,2,11]*

†Moritz Blumer and Chengwei Ulrika Yuan contributed equally to this work.

*Correspondence:
mdd34@cam.ac.uk; eam29@cam.ac.uk

¹ Department of Biochemistry, University of Cambridge, Tennis Court Road, Cambridge CB2 1GA, UK
Full list of author information is available at the end of the article

## Abstract

**Background:** East African cichlid fishes have diversified in an explosive fashion, but the (epi)genetic basis of the phenotypic diversity of these fishes remains largely unknown. Although transposable elements (TEs) have been associated with phenotypic variation in cichlids, little is known about their transcriptional activity and epigenetic silencing. We set out to bridge this gap and to understand the interactions between TEs and their cichlid hosts.

**Results:** Here, we describe dynamic patterns of TE expression in African cichlid gonads and during early development. Orthology inference revealed strong conservation of TE silencing factors in cichlids, and an expansion of *piwil1* genes in Lake Malawi cichlids, likely driven by PiggyBac TEs. The expanded *piwil1* copies have signatures of positive selection and retain amino acid residues essential for catalytic activity. Furthermore, the gonads of African cichlids express a Piwi-interacting RNA (piRNA) pathway that targets TEs. We define the genomic sites of piRNA production in African cichlids and find divergence in closely related species, in line with fast evolution of piRNA-producing loci.

**Conclusions:** Our findings suggest dynamic co-evolution of TEs and host silencing pathways in the African cichlid radiations. We propose that this co-evolution has contributed to cichlid genomic diversity.

## Introduction

The East African Great Lakes are home to prolific cichlid radiations, the most species-rich and phenotypically diverse adaptive radiations in vertebrates [1, 2]. In the last 10 million years, more than 1,700 species of cichlid fishes (Cichlidae family) have evolved in virtually every lacustrine and riverine ecological niche in Lakes Victoria,

Tanganyika, Malawi and surrounding bodies of water. The explosive diversification of East African cichlids is particularly striking in the haplochromine tribe and has resulted in astonishing variation in morphologies, colouration, diets, and behaviours [1, 2]. The genetic and epigenetic basis for such phenotypic variability is of great interest and remains, by and large, unknown.

Initial genomic studies suggested very low genetic variability amongst East African cichlids [3]. In Lake Malawi cichlids, for example, the reported average single nucleotide polymorphism divergence between species pairs was 0.1–0.25% [3, 4]. These low estimates were derived from approaches aligning short-read sequence data to a linear reference genome and generally ignore the contribution of structural variation. We have recently complemented these estimations using a pangenomic approach and long-read genome assemblies of representative Lake Malawi species [5]. With this approach, we estimated that 4.73–9.86% of Lake Malawi cichlid genomes can be attributed to interspecific structural variation [5]. Importantly, transposable elements (TEs) account for up to 74.65% of structural variant sequence. Thus, TEs comprise an underestimated source of genetic variability in East African cichlids.

TEs are diverse mobile genetic elements that inhabit nearly all eukaryotic genomes sequenced to date [6]. While most extant TEs and novel TE mobilisation events are selectively neutral or slightly deleterious to their hosts [7], several examples of TEs providing adaptive benefits to their hosts have been reported [8–10]. The TE landscapes of teleost fish genomes are highly dynamic [11–17], and cichlid genomes are no exception, as they contain varied TE populations with signs of recent transpositional activity [16, 18]. TEs may be an important source of (epi)genetic variability that has fuelled the cichlid radiations. Consistent with this notion, presence/absence variation of TEs is associated with pigmentation traits [19, 20], sex determination [21], and modulation of endogenous gene expression [18, 22]. It has recently been shown that differentially methylated regions enriched in young TEs are associated with transcriptional changes [23], further supporting a role for TEs in modulating gene expression in cichlids. The same study found widespread DNA methylation at TEs, but besides this, little is known about the silencing pathways that direct TE silencing in cichlids and lead to the deposition of DNA methylation.

Several pathways have evolved in animals to silence TEs, particularly in the germline and early development to protect the next generations from deleterious effects of TE activity [8, 24–30]. Here, we focus on the Piwi-interacting RNA (piRNA) pathway, a class of non-coding small RNAs (sRNAs) 21–35 ribonucleotides long, which drive silencing of TEs in the animal germline, including in fishes [27, 31–35]. piRNAs bind to Piwi Argonaute proteins and guide them to target RNAs with base complementarity, leading to post-transcriptional and/or transcriptional silencing of their targets [26, 27]. The latter can be achieved by piRNA-directed DNA methylation of targets. piRNA biogenesis is complex, requires a variety of co-factors, and can be conceptualised as two collaborating pathways that create sequence diverse piRNA populations in the animal germline: the ping-pong and phased biogenesis pathways [26, 27, 32, 36–39]. These pathways depend mainly on the slicer activity of Piwi proteins, and endonucleolytic activity of Zucchini/PLD6 acting on long piRNA precursor transcripts.

The co-evolution of TE silencing factors and TEs is often thought to occur in the form of an arms race. TE silencing factors, including those of the piRNA pathway, often have signatures of fast, adaptive evolution that are interpreted as a consequence of such an arms race [24, 25, 40–43]. These signatures include positive selection and lability in terms of copy number variation, with recurrent gene duplications and turnover. Little is known about the co-evolution of TE silencing pathways and TEs in East African cichlids and whether these arms races could help fuel cichlid radiations.

Here, we describe dynamic TE expression in the gonads and early development of African cichlids. We identify cichlid orthologs of known factors required for TE silencing in vertebrates and discover an expanded repertoire of *piwil1* genes in Lake Malawi cichlids, which may have been driven by PiggyBac TEs. The additional *piwil1* paralogs retain amino acid residues required for the catalytic activity of the PIWI domain and have signatures of adaptive evolution, suggesting acquisition of novel regulatory functions. TE silencing factors are expressed in cichlid gonads, alongside an abundant piRNA population with signatures consistent with active piRNA-driven TE silencing. Lastly, we observe divergence in the genomic origins of piRNA production in closely related Lake Malawi cichlids.

## Results

### TE transcriptional activity in cichlid gonads and early development

To profile TE expression in African cichlids, we sequenced mRNAs of representative species of haplochromine cichlids from each of the major East African Great Lakes (Fig. 1A). We chose *Pundamilia nyererei* (PN) as a representative for Lake Victoria, *Astatotilapia burtoni* (AB) for Lake Tanganyika, and *Astatotilapia calliptera* (AC) for Lake Malawi. To compare closely related species within the same Lake, we included two species from Lake Malawi, alongside AC: *Maylandia zebra* (MZ), and *Tropheops* sp. 'mauve' (TM). In addition, we included *Oreochromis niloticus* (ON, commonly known as Nile tilapia) as an outgroup. ON is a representative of the tilapine tribe that has a broad geographical distribution in Africa and is not as phenotypically diverse as haplochromines [44]. We profiled TE expression in cichlid gonads, as these contain the germline, where the arms race between TEs and their silencing factors is most apparent in other animals [8, 27]. For a comprehensive analysis of younger TE populations in Lake Malawi, we created an additional curated TE annotation for AC, which we used throughout this work alongside the uncurated annotation (Additional File 1: Fig. S1A, see Methods).

We found that 515–746 (86–93%) cichlid TE families show detectable expression in gonads (Additional File 1: Fig. S1B). Two trends are recognisable when considering the expression of TE families grouped by class. First, long terminal repeat (LTR) families have the highest median expression (Additional File 1: Fig. S1C-D). This trend is reversed when TE expression is quantified based on the curated TE annotation of AC, which has more annotated LTR families (Additional File 1: Fig. S1B) and where LTR annotations were improved, including both the long-terminal repeats and intervening genes. This suggests that uncurated LTR annotation may lead to an overestimation of LTR expression. Second, TE families of the same class tend to be more highly expressed in testes rather than ovaries, revealing differences in TE expression between sexes (Additional File 1: Fig. S1C). These differences extend to many TE superfamilies (Additional

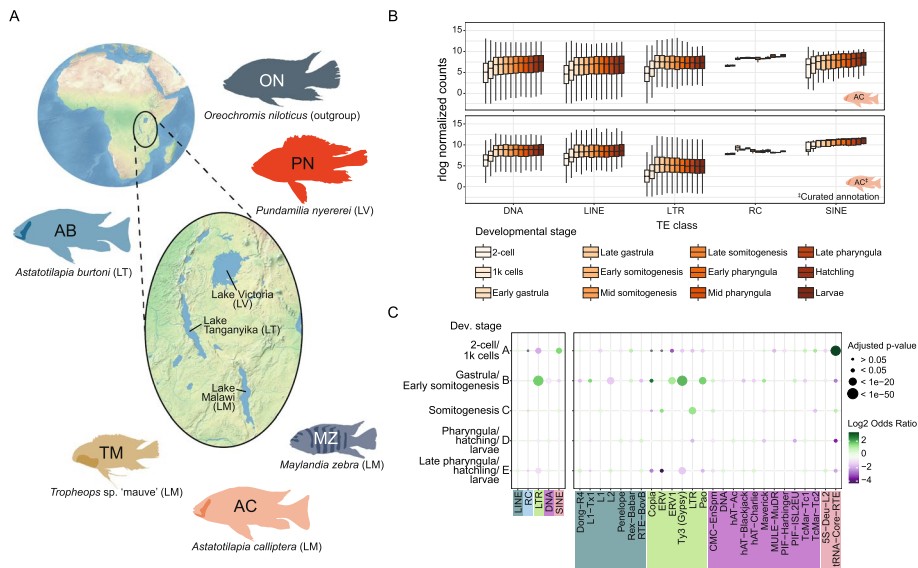

**Fig. 1** Dynamic patterns of TE expression during cichlid early development. **A** The East African Great Lakes and surrounding bodies of water, along with the species used in this study, each representative of a major lake. *Oreochromis niloticus* (Nile tilapia) is used as an outgroup to the radiations of the Great Lakes. For Lake Malawi, we use three species to address within-lake dynamics of TE expression and epigenetic silencing. *Astatotilapia calliptera* is a generalist omnivore, which inhabits shallow water environments in the lake and surrounding rivers and streams [4, 44], while *Maylandia zebra* and *Tropheops* sp. 'mauve' are Mbuna rock-dwelling cichlids specialised in eating algae [4, 44]. Maps obtained from Natural Earth, naturalearthdata. com. **B** Expression of TE families belonging to major TE classes throughout early development of *A. calliptera*, displayed as regularised log (rlog) normalised counts. TE Expression was calculated using the default (panel above) and curated annotations (panel below). **C** Enrichment of TE classes and superfamilies in particular developmental stages, according to clusters of differentially expressed TEs defined in Figure S2D. Only TE superfamilies significantly enriched/depleted in at least one developmental stage are depicted. Grey dots represent lack of significant enrichment. Analysis done as in Chang et al., 2022 [13], using SQuIRE counts mapped to the curated TE annotation of AC. AB, *Astatotilapia burtoni*; AC, *Astatotilapia calliptera*; LM, Lake Malawi; LT, Lake Tanganyika; LV, Lake Victoria; MZ, *Maylandia zebra*; ON, *Oreochromis niloticus*; PN, *Pundamilia nyererei*; rlog, regularised log; TM, *Tropheops* sp. 'mauve'

File 1: Fig. S1D). Higher median expression of annotated protein-coding genes was also observed in cichlid testes (Additional File 1: Fig. S1E), suggesting the sex biases in TE expression may follow general sex biases in transcriptional output.

Embryogenesis and early development are periods known to display signs of TE transcriptional activity [8, 10, 13]. We therefore conducted bulk mRNA sequencing in early developmental stages of Lake Malawi cichlids and found that 91–94% of cichlid TE families have detectable expression during early development (Additional File 1: Fig. S2A). Expression in these developmental stages is overall identical between AC and TM, apart from Ty3 LTRs (also known as Gypsy elements, Fig. 1B and Additional File 1: Fig. S2B-C). The temporal expression pattern of most TE classes and superfamilies is similar: lower expression before gastrulation rising to peak or near peak expression at early gastrula followed by relatively constant levels of transcriptional activity (Fig. 1B and Additional File 1: Fig. S2C). These trends in expression are observed when gene expression of entire TE families is grouped together and may mask more heterogeneous expression patterns of individual TEs. To address heterogeneity, we quantified TE expression at the locus level. This analysis revealed the overall expression pattern at the family level is not universal, as several individual TEs have expression patterns specific to distinct

developmental stages (Additional File 1: Fig. S2D). Interestingly, we find a major enrichment of ERV1, Ty3 and Pao LTRs in gastrula stage and early somitogenesis (Fig. 1C, cluster B), and SINE (Short INterspersed Element) enrichment at the earliest stages (Fig. 1C, cluster A). These enrichment patterns may have important implications for fish development (see Discussion). With this analysis we determined the proportion of counts mapping to TEs in relation to the annotated genome features they overlap with. This analysis is useful to assess the proportion of TE counts related to read-through transcription from expressed protein-coding genes. The majority of the counts (between 38.5–49.3%) map to TEs within intergenic regions, and are thus unlikely to originate from read-through transcription from expressed genes (Additional File 1: Fig. S2E). Overall, these results support diverse and dynamic transcriptional TE activity in gonads and during early development of African cichlids.

### An expanded repertoire of *piwil1* genes in Lake Malawi cichlids

Given the dynamic TE expression patterns observed, we reasoned that active silencing pathways must be in place in cichlids to counteract TE activity. First, we identified cichlid orthologs of sRNA-based TE silencing factors conserved in animals (Additional File 2: Table S1) [8]. With three exceptions, all genes are present in cichlid genomes (Additional File 1: Fig. S3A and Additional File 2: Table S1).

Then, we addressed whether these factors are expressed in the germline by performing quantitative proteomics on gonads of representative cichlid species. TE silencing factors are detected most prominently in testes (Additional File 1: Fig. S3B). Abundant yolk proteins, from the substantial yolk fraction of cichlid eggs [45], precluded protein detection in ovary samples at a depth similar to other organs (Additional File 1: Fig. S3C). Despite the influence of the yolk, Piwil1, a core piRNA pathway factor was detected in the ovaries of all species (Additional File 1: Fig. S3B). Somatic roles for the piRNA pathway have been increasingly recognized in animals, including in brain and nervous system [46]. We also profiled the proteome of brain tissues of the representative cichlid species, but obtained no consistent evidence supporting expression of core piRNA factors in the brain of all cichlid species (Additional File 1: Fig. S3B). These results point to strong conservation of germline-expressed TE silencing factors in African cichlids.

While inspecting TE silencing factor orthologs, we detected multiple copies of *piwil1* genes, homologs of zebrafish *ziwi* [33], in cichlids representative of Lake Malawi, but not in representatives of Lakes Tanganyika and Victoria (Fig. 2A-B, Additional File 2: Table S1). While fishes generally have one *piwil1* copy [33, 47–49], AC has four *piwil1* copies, which we named *piwil1.1–1.4* (Additional File 2: Table S1). Two of these are full-length copies, whereas the other two are truncations containing only the PIWI domain (Fig. 2A-B). *piwil1.1* of AC is located in the conserved syntenic context of vertebrate *piwil1* genes (Additional File 1: Fig. S4A), indicating that *piwil1.1* is the ancestral cichlid *piwil1* gene. By aligning all additional *piwil1* copies of AC to the coding sequence of *piwil1.1* and projecting the coding sequence to the aligned paralogs, we observe that the full-length paralog *piwil1.2* likely contains stop codons that are bypassed in existing gene annotations produced by automated annotation pipelines (Fig. 2A). Also, *piwil1.2* is expressed at negligible levels in cichlid gonads and brain (Additional File 1: Fig. S4B) and is therefore likely a pseudogene.

*piwil1.2*, *piwil1.3*, and *piwil1.4* reside in genomic regions rich in TEs (Fig. 2A). The 3' regions of *piwil1.2*, *piwil1.3* and *piwil1.4* share a PiggyBac TE insertion (Fig. 2A and Additional File 1: Fig. S4C). PiggyBac is a DNA TE family known to be very proficient at carrying large DNA segments upon transposition, a quality that has promoted its use in genome engineering [51, 52]. Autonomous PiggyBac TEs consist of two terminal inverted repeats (TIRs) flanking a transposase gene [53]. Like other DNA TEs with TIRs, PiggyBacs mobilise when two transposase proteins each bind to one of the TIRs [6]. The *piwil1*-associated PiggyBacs have mutations that preclude production of a functional transposase (Additional File 1: Fig. S4C). These *piwil1*-associated PiggyBac belong to the same TE family (PiggyBac-1), of which we identified 315 high quality copies in the AC reference genome (with a RepeatMasker Smith-Waterman alignment score > 1000, see Methods). Considering the genome size of > 880 Megabase, one PiggyBac-1 element is expected, on average, every 2.8 Megabase. A phylogeny of all high-confidence PiggyBac-1 TE copies in the AC genome shows that the three *piwil1*-associated PiggyBac TEs are closely related, particularly the PiggyBacs associated with *piwil1.3* and *piwil1.4* (Additional File 1: Fig. S4D). Finding all three *piwil1* paralogs on different chromosomes with closely related flanking PiggyBac-1 insertions either directly 3' adjacent (*piwil1.2* and *piwil1.4*) or 7 kb downstream (*piwil1.3*) is therefore highly unlikely to be coincidental.

(See figure on next page.)

**Fig. 2** An expansion of *piwil1* paralogs in Lake Malawi cichlids likely mediated by PiggyBac TEs. **A** Detailed schematics of the four *piwil1* loci in the *A. calliptera* reference genome. Exons and TEs are shown, along with other relevant sequence features, such as start and stop codons, deletions, and insertions. The sequences of the putative PiggyBac TIRs (terminal inverted repeats), TIR-like sequences and preferred insertion sites are shown in white boxes, from 5' to 3'. Of note, the putative TIR and insertion site sequences distal to the PiggyBac are the reverse complement of the TIR and insertion site sequences on the right flank of the PiggyBac. The dotted lines represent the borders of duplicated regions, according to multiple sequence alignment. Region S marks the genomic region shared by all *piwil1* genes. The image in the lower portion of the panel is a zoomed-out snapshot of the multiple sequence alignment, colour-coded by nucleotide. The putative stop codons in *piwil1.2* were identified manually from an alignment of the genomic regions of all *piwil1* copies with the coding sequence of *piwil1.1*, the *piwil1* gene most conserved in vertebrates. No putative premature stop codons were found in *piwil1.3* and *piwil1.4*. **B** Schematics of the domain structure of the five Piwi proteins annotated in the *A. calliptera* genome, including the expanded Piwil1 protein repertoire. Due to the putative stop codons found in the *piwil1.2* locus, as shown in (A), it is likely that the protein is misannotated and that the full-length protein will not be produced. **C** Presence (green)/absence (black) of each *piwil1* gene in genomes of Lake Malawi and Tilapia cichlids. Presence of *piwi*-associated PiggyBac TEs is indicated in orange. Presence/absence of *piwil1* genes and PiggyBac TEs was ascertained from long-read sequencing of 12 individuals and short-read sequencing of 79 individuals spanning all the major eco-morphological clades in Lake Malawi. The cladogram of the Malawi radiation reflects the current understanding of the radiation based on genomic studies [4]. The proposed model for *piwil1* gene evolution involves gene expansion early in the Lake Malawi radiation, followed by partial losses in particular lineages. **D** Neighbour-joining tree representing the Hamming distance between the non-coding regions of the *piwil1* genomic sequences of *A. calliptera* along with the genomic sequence of *piwil1* of *O. niloticus* (*Onpiwil1*) as an outgroup. In specific, the non-coding regions within region S shared by all *piwil1* genes, as shown in (A), were used for the multiple sequence alignment. **E** The plots show genome-wide results of Raised Accuracy in Sweep Detection (RAiSD) [50]. μ is a metric incorporating three selective sweep signatures, with higher μ values indicative of a stronger signature of selection. Upper panels show μ across the entire chromosome, or entire scaffold in case of *piwil1.2*. Lower panels are insets of the *piwil1* gene regions ± 1 Megabase (Mb). As the entire scaffold where *piwil1.2* resides is less than 2 Mb, no inset is shown. We calculated per-gene μ for all genes (see Methods), and with this approach *piwil1.3* and *piwil1.4* are in the 99th and 93rd percentile, respectively, of per-gene μ

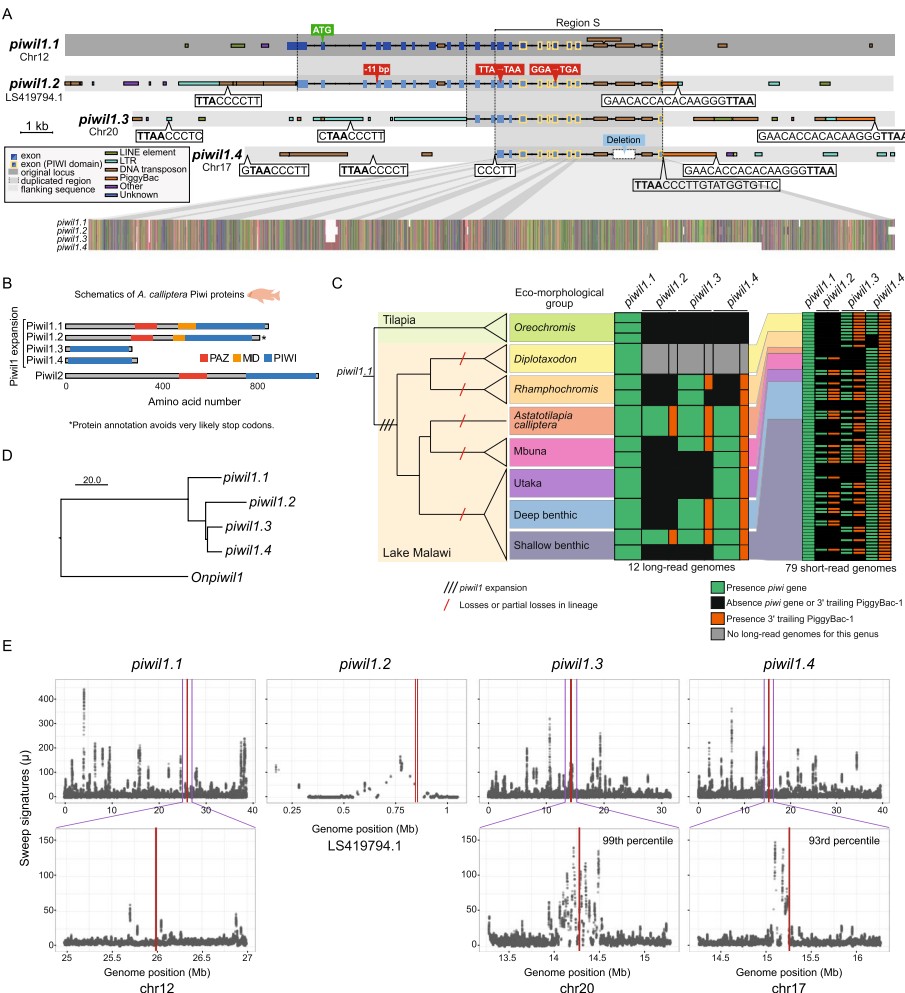

**Fig. 2** (See legend on previous page.)

Given the presence of related PiggyBac-1 TEs associated with all three *piwil1* paralogs, we reasoned that the initial expansion of *piwil1* genes in Lake Malawi cichlids was likely driven by transposition of PiggyBac-1, either at a time when its transposase was active, or in a non-autonomous fashion using the transposase of other PiggyBacs. This could have happened if a PiggyBac transposase used one of its own TIRs together with an alternative TIR-like sequence from the *piwil1* locus. To address this, we searched for sequence signatures of PiggyBac mobilisation: the preferred insertion sequence (5'-TTAA-3'), directly preceding the complete PiggyBac TIR sequences of this PiggyBac family, according to our curated TE annotation (left TIR: 5'-CCCTTGTATGGTGTTC-3'; right TIR: 5'-GAACACCACACAAGGG-3') [53, 54]. We found complete TIRs adjacent to the PiggyBac-1 elements, and close to the border of the *piwi* duplications distal to the PiggyBac (Fig. 2A). Putative PiggyBac insertion signatures distal to the PiggyBac-1 element of *piwil1.2* and *piwil1.3* could not be conclusively identified with BLASTN of the complete TIR sequences (Additional File 2: Table S1). This may be partially because of erosion and additional transposition in that area that could have pushed the PiggyBac sequence signature further upstream from *piwil1* (Fig. 2A). Several sequences similar (allowing

only one mismatch) to the target sequence 5'-TTAA-3' followed by 5'-CCCTT-3' could be identified upstream of *piwil1* genes, potentially corresponding to alternative TIR-like sequences. Furthermore, we could find a 5'-CCCTT-3' sequence directly upstream of *piwil1.4*, but the downstream 5'-TTAA-3' insertion sequence may have eroded.

Next, we tested whether cichlid *piwi* genes are prone to be targeted by PiggyBac elements by being enriched with the PiggyBac's preferred insertion sequences (5'-TTAA-3'). While the regions flanking *piwi* genes ($\pm 5$ kilobase) in the AC genome do not have significantly more 5'-TTAA-3' than the flanking regions of other genes ($\pm 5$ kilobase; Welch's t-test, $p$-value $= 0.86$, Additional File 1: Fig. S4E), the *piwi* gene sequence proper does have slightly more 5'-TTAA-3' than the sequences of other genes (Welch's t-test, $p$-value $= 0.018$, Additional File 1: Fig. S4F). This could reflect a higher tendency of *piwi* genes to be targets of TEs with 5'-TTAA-3' preferred insertion sites. Of note, *piwil1.1* has the greatest number of 5'-TTAA-3' sites of all *piwi* genes, but no associated PiggyBac TEs. This observation argues against active targeting of *piwi* genes by these TEs.

The consistent association of closely related PiggyBac TEs to *piwil1* paralogs, and the presence of putative TIR sequences flanking the genes are compatible with a model whereby PiggyBac-1 transposition mediated, at least partially, the expansion of *piwil1* genes in Lake Malawi cichlids.

### Evolution and functional potential of *piwil1* genes in Lake Malawi cichlids

Next, we assessed the prevalence of each *piwil1* paralog in the major eco-morphological clades of Lake Malawi cichlids. We mapped genomic reads to the AC reference genome (which contains all four *piwil1* copies) and manually assessed the presence or absence of each *piwil1* gene from mapped reads. We used 12 sets of long reads and 79 sets of short reads of Lake Malawi cichlids, corresponding to 80 species (Additional File 2: Table S1). We did not find any of the three extra *piwil1* paralogs in tilapias, which form an outgroup to the haplochromine radiations (Fig. 2C). However, the *piwil1* paralogs have a broad distribution across all major eco-morphological clades of the Lake Malawi radiation (Fig. 2C). *piwil1.1* and *piwil1.4* are most widespread, with *piwil1.1* identified in all individuals and *piwil1.4* found in 82/88 individuals (exceptions are 6/7 individuals of the *Rhamphochromis* genus, Fig. 2C and Additional File 2: Table S1). Conversely, *piwil1.2* and *piwil1.3* have a patchier distribution (27/88 and 46/88 individuals). We found support for a 3' trailing PiggyBac-1 TE in the vast majority of *piwil1.2*, *piwil1.3*, and *piwil1.4* copies (153/155, Fig. 2C and Additional File 2: Table S1). In 8 individuals we found support for a 3' trailing PiggyBac-1 TE in their expected location 3' of *piwil1.3* and *piwil1.4*, but found no support for the *piwil1* gene itself (Fig. 2C and Additional File 2: Table S1). This observation may reflect rare events of *piwil1* gene elimination by recombination processes.

Inspection of alignments of all AC *piwil1* paralogs revealed that *piwil1.2*, *piwil1.3* and *piwil1.4* all share variation that is not shared with *piwil1.1* (Fig. 2A). Moreover, *piwil1.3* and *piwil1.4* share the most variation. This, together with the relatedness of the *piwil1*-associated PiggyBac-1 elements (Additional File 1: Fig. S4D), suggests that *piwil1.2* was the first paralog to duplicate via transposition and that *piwil1.3* and *piwil1.4* originated from *piwil1.2.* A tree representing the distance between the non-coding regions shared by all four *piwil1* genes of AC (within region S in Fig. 2A) and *piwil1* of ON (as

an outgroup) support this hypothesis (Fig. 2D). A similar tree created from the exons shared by these same *piwil1* genes (within region S in Fig. 2A) did not produce a tree topology congruent with the non-coding tree (compare Fig. 2D with Additional File 1: Fig. S4G). We suggest that this discrepancy could reflect selective processes acting on the coding sequences of *piwil1* genes.

Following gene duplication, paralogs can undergo a number of evolutionary routes, including towards sub- or neofunctionalisation [55], with distinct signatures of selection. To learn about the selective pressures at play, we tested for the presence of signatures of selective sweeps in 79 Lake Malawi cichlid genomes (Additional File 2: Table S1). While the genomic region of *piwil1.1* does not display a clear signature of selective sweep (Fig. 2E, left panels), *piwil1.3* and *piwil1.4* are in the 99th and 93rd percentiles, respectively, of genes with highest values of integrative sweep signatures, supporting positive selection at these loci (Fig. 2E and Additional File 2: Table S1). Moreover, we found evidence of positive selection in cichlid Piwil1 proteins beyond Lake Malawi, particularly in amino acid residues in the PIWI domain or immediately C-terminally adjacent to the annotated domain (Additional File 1: Fig. S5A and Additional File 2: Table S1). These results are in line with positive selection acting on cichlid Piwi proteins, most notably in the expanded Piwi repertoire of Lake Malawi cichlids. Overall, the data suggests a scenario consistent with *piwil1* expansion early in the radiation, followed by positive selection and gene losses.

Next, we sought to determine whether the expanded copies of *piwil1* genes in Lake Malawi are expressed. We excluded *piwil1.2* from further analysis, because both the premature stop codons in conserved exons (Fig. 2A) and low expression (Additional File 1: Fig. S4B), suggest that it is a pseudogene. First, we interrogated *piwil1* gene expression at the mRNA level. We also probed the expression of *piwil2*, the *piwi* gene homolog of zebrafish *zili* [33], which did not undergo gene duplication. *piwil1.1* and *piwil2* are strongly expressed in gonads but not in brain (Fig. 3A and Additional File 1: Fig. S5B), in line with known TE silencing roles in the germline of other organisms [8, 27, 33]. *piwil1.4* was expressed in gonads, and lowly expressed in brain. During early development of Lake Malawi cichlids, we detected strong maternal deposition of *piwil1.1* and *piwil2* transcripts (Fig. 3B). In contrast, *piwil1.4* seems to be expressed mainly after gastrulation, likely after the onset of zygotic expression. No expression of *piwil1.3* was detected in these organs and in early development (Fig. 3A-B).

To gain further insights into the potential function of these Piwil1 proteins, we analysed their protein sequence and structure. As Piwil1.3 and Piwil1.4 have only the PIWI domain (Fig. 2B), we focused on the portion of Piwil1 proteins encompassing this domain. We found low overall variation in African cichlid Piwil1 proteins, but the Lake Malawi truncations showed higher divergence than their full-length orthologs (Fig. 3C). This divergence is not expected to disrupt protein structure, as the predicted structures of full-length Piwil1.3 and Piwil1.4 proteins align well with the known structures of Piwi proteins of *Drosophila melanogaster* and *Bombyx mori* [56, 57], and the predicted PIWI domain of Piwil1.1 (Additional File 1: Fig. S5C). The PIWI domain is a ribonuclease H-like domain, the catalytic centre of Argonaute proteins responsible for their slicer activity. Within the PIWI domain, a DDE motif of amino acid residues is required for Argonaute cleavage [58, 59]. Despite the higher divergence of Piwil1.3

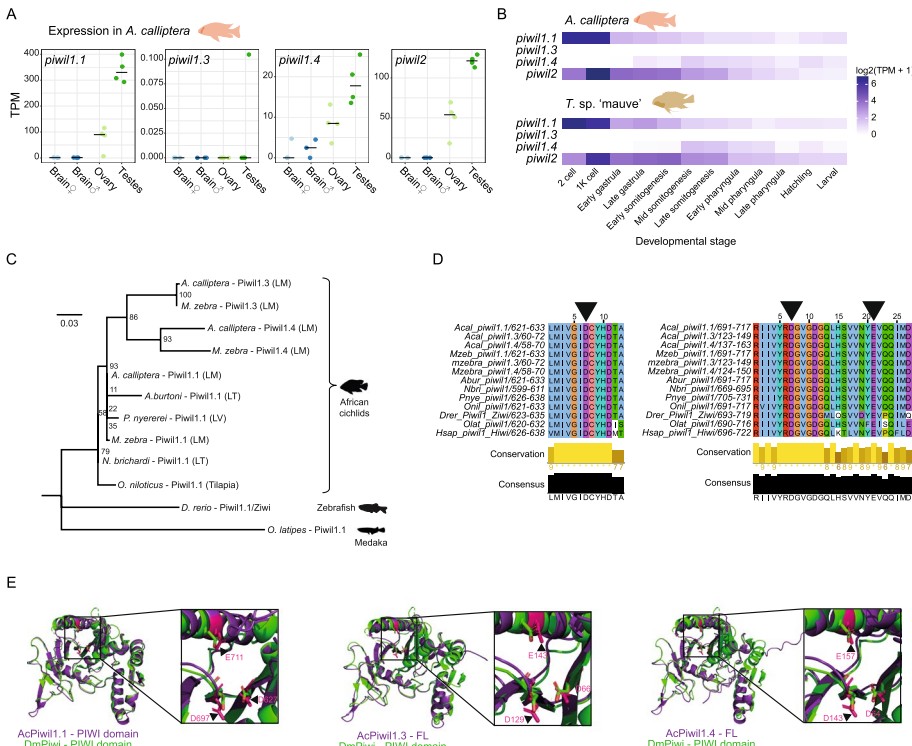

**Fig. 3** Expression and functional potential of Piwil1 proteins in Lake Malawi. **A** Expression, in Transcripts per Million (TPM), of *piwil1* paralogs and *piwil2* in gonads and brain of *A. calliptera*. Data points represent distinct biological replicates. **B** Expression of the three *piwil1* genes and *piwil2* throughout early development of *A. calliptera* and *Tropheops* sp. 'mauve', another Lake Malawi cichlid. **C** Phylogenetic tree constructed from an alignment of the PIWI domain of Piwil1 proteins of African cichlids, using zebrafish and medaka as outgroups. Branch support numbers are shown at the tree nodes and were calculated with 10,000 ultrafast bootstrap replicates. **D** Specific regions of the multiple sequence alignment of several PIWI domains, surrounding the integral residues of the catalytic triad, indicated with black arrowheads, the catalytic residues within the PIWI domain known to be important for Piwi-mediated cleavage. These residues are conserved in Piwil1 proteins of African cichlids, including in *piwil1.3* and *piwil1.4* in Lake Malawi. **E** Structural alignments of the PIWI domain of *Drosophila melanogaster* (Dm) Piwi protein and AlphaFold predictions of Piwil1.1 (using only PIWI domain, left), Piwil1.3 (full-length, centre), and Piwil1.4 (full-length, right) of *A. calliptera*. Regions of the structural alignment encompassing the catalytic triad are augmented in the insets and the triad residues are highlighted with black or white arrowheads

and Piwil1.4, they retain a conserved DDE motif, as Piwil1.1 (Fig. 3D). Furthermore, the PIWI domain structures of Lake Malawi Piwil1 proteins are predicted to be identical to those of *D. melanogaster* and *B. mori* Piwi proteins, including the relative position of the DDE motif residues (Fig. 3E and Additional File 1: Fig. S5D). These data indicate the genomes of Lake Malawi cichlids encode three Piwil1 proteins with potentially catalytically active PIWI domains.

## Cichlid gonads express piRNAs

To characterise the piRNA cofactors of cichlid Piwi proteins, we sequenced sRNAs from gonads of the selected cichlid species (Fig. 1A). The sRNA length distribution profiles in gonads have prominent peaks at lengths of 21–22 nucleotides, likely corresponding to microRNAs (Additional File 1: Fig. S6A). Contrary to microRNAs, piRNAs have high sequence diversity [27]. When sRNA reads are collapsed into unique sequences,

we observed prominent sRNA populations between 24–31 nucleotides long, consistent with the length distribution of piRNAs (Fig. 4, left panels). In testes, sRNA populations peaked at lengths of 26–27 nucleotides, whereas in ovaries the peak was shifted to 28–29 nucleotides long (Fig. 4 and Additional File 1: Fig. S6B).

We selected sRNAs between 24–35 nucleotides long for subsequent analysis and searched for the two typical sequence signatures of piRNAs: a bias for uridine at position 1 (1U), and a bias for an adenine at position 10 (10A) [27, 31, 32, 36, 37, 60]. Unique sRNA sequences between 24–35 nucleotides long clearly show both the 1U and 10A biases in cichlid gonads, as well as additional signatures consistent with active piRNA ping-pong and phased biogenesis pathways (Fig. 4 and Additional File 1: Fig. S6B-G). Of note, while phased piRNA biogenesis is pervasive in cichlid testes, ovary sRNAs display no clear signatures of phased biogenesis, except in AC ovaries (Fig. 4 and Additional File 1: Fig. S6F-G). This lack of phased biogenesis signatures cannot be explained

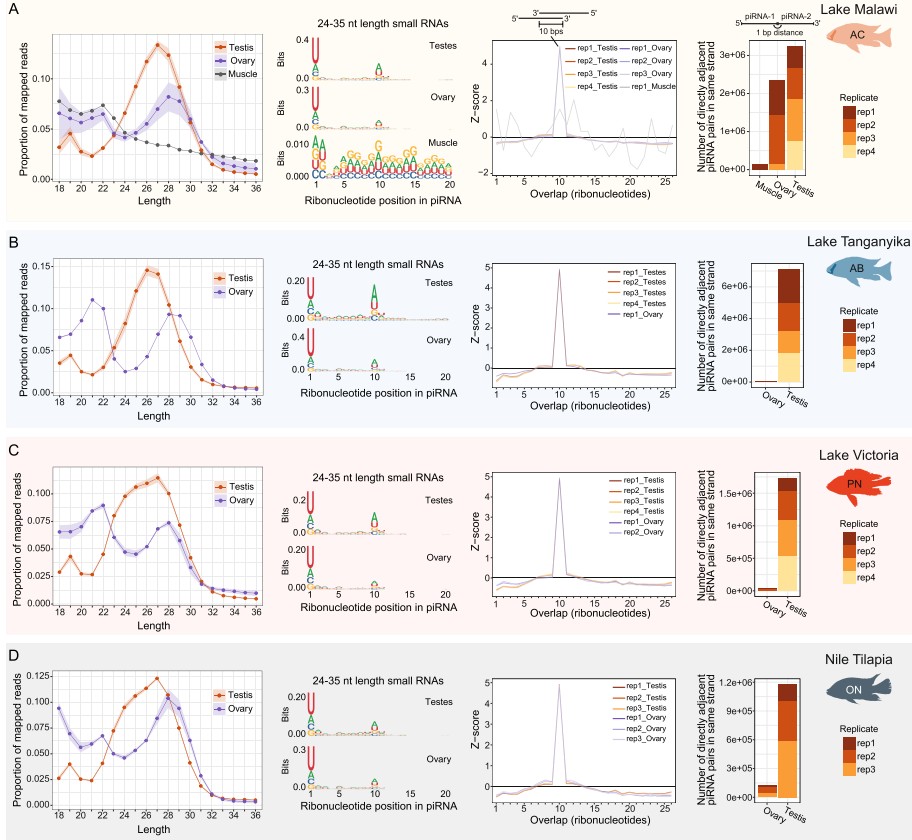

**Fig. 4** African cichlids express piRNAs in gonads. **A**-**D** sRNA length distribution profiles and piRNA sequence signatures in sRNAs 24–35 nucleotides long. sRNA length profiles shown here (left-most panels) comprise only reads of unique sequence. The shading in the sRNA length distribution profiles indicates standard deviation of replicates (no shading for *A. burtoni* ovary, as only one replicate is shown). Sequence logos (second set of panels from left) denote the 1U bias typical of piRNAs, and the 10A signature of ping-pong amplification in gonad sRNAs but not in muscle tissues of *A. calliptera* (A). Third set of panels from left show ping-pong signature with a robust overlap of 10 ribonucleotides in piRNA pairs. Right-most panels show number of piRNA pairs in same orientation that are directly adjacent, indicative of phased piRNA biogenesis. Signature is observable in the testes of all species, but in ovaries it is detectable only in AC. AB, *Astatotilapia burtoni*; AC, *Astatotilapia calliptera*; CPM, Counts Per Million; ON, *Oreochromis niloticus*; PN, *Pundamilia nyererei*

by the expression of *mov10l1* and *pld6*, factors known to be required for this process in other species [27] (Additional File 1: Fig. S6H). As a control, we sequenced sRNAs from muscle, as a representative somatic tissue of AC and found no prominent population of sRNAs in the piRNA length range with piRNA signatures (Fig. 4A and Additional File 1: Fig. S6E-F). Thus, cichlid gonads express sRNA populations consistent in length and sequence signatures with an active piRNA pathway.

### The genomic origins of cichlid piRNAs

piRNAs are often created from discrete genomic regions termed piRNA clusters [26, 27, 31, 32]. To finely map piRNA clusters, we used piRNA Cluster Builder (piCB) [61], a novel computational approach that identifies piRNA clusters by incorporating information from uniquely- and multi-mapping reads in a stepwise manner (see Methods). We restricted the analysis to sRNA sequencing data of Lake Malawi (AC, TM, and MZ, all mapped to the AC genome) and ON, because these chromosomal level assemblies allow us to define the piRNA clusters within genomic coordinate systems and to understand their biological context. We benchmarked this approach in AC testes samples to identify appropriate parameter settings (Additional File 1: Fig. S7A-F and Additional File 3: Table S2). Then, we extended our analysis and identified thousands of genomic sources of piRNAs in Lake Malawi cichlids (Fig. 5A, Additional File 1: Fig. S8A and Additional File 3: Table S2, between 3,091–3,251 in ovaries and 3,494–4,252 in testes) and ON (Fig. 5B, Additional File 1: Fig. S8B, and Additional File 3: Table S2, between 3,194–7,352 in ovaries and 4,053–4,781 in testes). The clusters explain 65−80% of piRNA reads in the library (Additional File 1: Fig. S9A). Although the total number of clusters are comparable in testes of distinct Lake Malawi species

(See figure on next page.)

**Fig. 5** Fluid genomic origins of cichlid piRNAs. **A** Left panel shows the mean number of clusters identified in Lake Malawi cichlid gonads. Error bars represent standard deviation. Right panel depicts the number of clusters shared between the replicates of the organs indicated. **B** Left panel represents the mean number of clusters identified in *O. niloticus* gonads. Error bars represent standard deviation. Right panel shows the shared clusters between the replicates indicated. **C** Circos plot showing the TE distributions (tracks 1–2) from non-curated (track 1) and non-curated annotations (track 2), plus the chromosomal locations of shared and divergent piRNA clusters in the Lake Malawi cichlids whose piRNAs were profiled. Tracks 3–8 are bar plots with the number of clusters present in genome bins. Track 3 shows the piRNA clusters shared by Lake Malawi gonads, while track 4 shows the piRNA clusters that are not present in both *A. calliptera* testes and ovaries. Track 5 displays the clusters shared in all testes samples of the three Lake Malawi species profiled (AC/MZ/ TM). Tracks 6–8 display the piRNA clusters identified in the testes of one of the Lake Malawi species, but not in the testes of other species. **D** Circos plot showing the TE distribution (track 1) and chromosomal locations of shared (track 2) and divergent (track 3) piRNA clusters in the gonads of *O. niloticus*. Tracks 2–3 are bar plots with the number of clusters present in genome bins. **E**–**F** Strand biases in piRNA production, shown as the ratio of sense over antisense piRNAs intersecting each piRNA cluster in Lake Malawi (E) or *O. niloticus* (F). The grey violin plot represents all piRNA clusters identified, while the orange violin plot represents the sense/ antisense ratio normalised according to cluster productivity. The purple region highlights piRNA clusters with piRNA production less than 100-fold different between the sense and anti-sense strands. Thus, values that fall within this range likely account for piRNA clusters producing piRNAs from both strands. **G-I** Genome tracks with examples of clusters identified in Lake Malawi cichlids (G-H) and in *O. niloticus* (I). Blue and red tracks represent 24–35 nucleotide long piRNAs, in Counts per Million (CPM), mapping to the plus and minus strands, respectively. TE tracks are color-coded according to TE class, color key is on the right of panel (I). (G) shows a piRNA cluster identified in the testes of Lake Malawi cichlids. (H-I) are examples of piRNA clusters identified in ovaries and testes of Lake Malawi cichlids (H) and *O. niloticus* (I). AC, *Astatotilapia calliptera*; CPM, Counts Per Million; MZ, *Maylandia zebra*; ON, *Oreochromis niloticus*; TM, *Tropheops* sp. 'mauve'

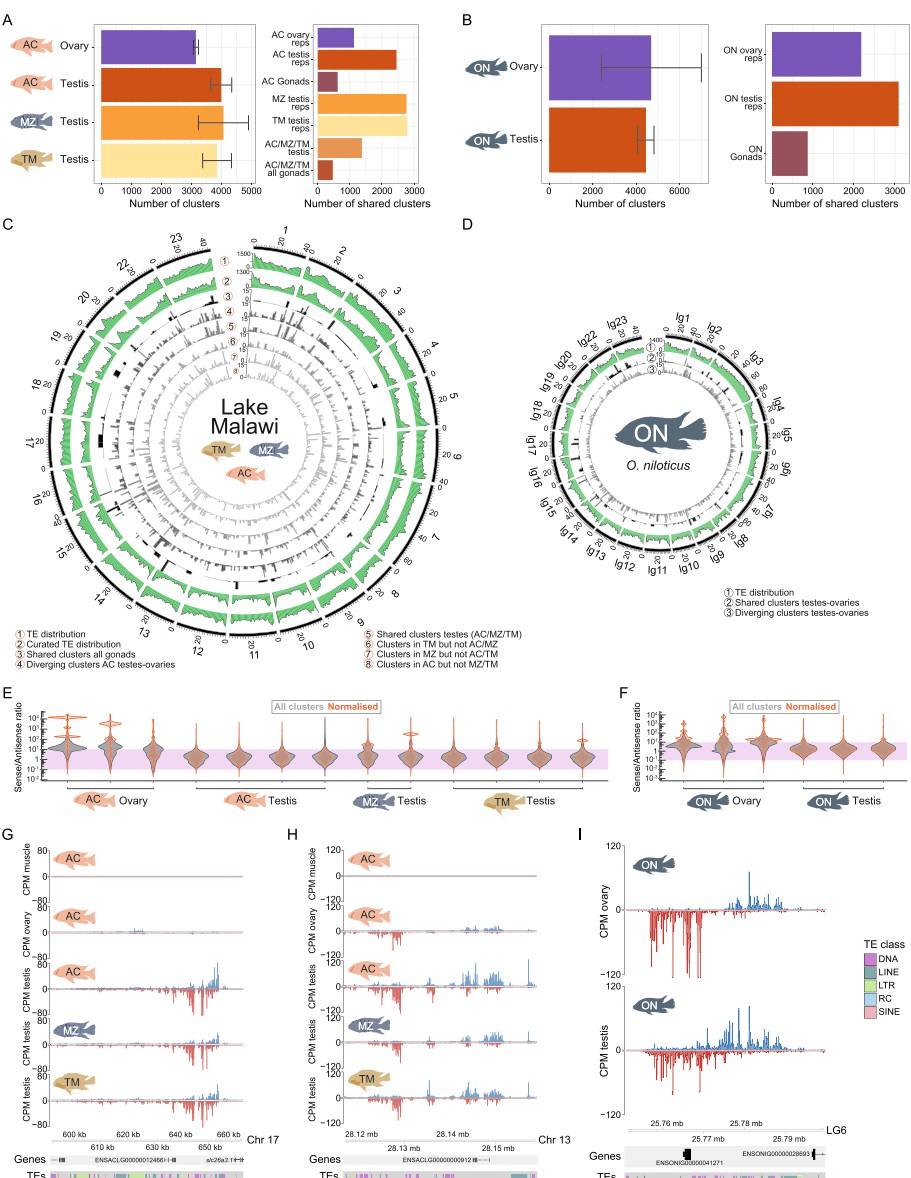

**Fig. 5** (See legend on previous page.)

(Fig. 5A, compare testis of AC, MZ, and TM), only a fraction of clusters is shared between all three species (Fig. 5A, C, 1,377 shared clusters), revealing variation in piRNA production in closely related Lake Malawi cichlids. Moreover, the even lower number of clusters shared between testes and ovaries illustrates sex differences in piRNA production (Fig. 5A-D, 622 shared clusters in AC gonads, 469 clusters shared across all Lake Malawi testes and ovaries, and 872 shared clusters in ON gonads). As it has been noted that productivity affects reproducibility of cluster identification [61] (Additional File 1: Fig. S7F), we repeated the piRNA cluster intersections using only the 50% most productive clusters identified in each library. The resulting intersections are identical to those reported above, indicating the extensive divergence in piRNA production observed is not attributable to lowly productive piRNA clusters

(Additional File 1: Fig. S9B-C). Overall, these results suggest considerable fluidity in the sources of piRNA production in cichlids, even amongst species endemic to the same Lake.

We explored additional features of the cichlid piRNA clusters identified. Most piRNA clusters are shorter than 50 kb (Additional File 1: Fig. S9D-E). In testes, clusters tended to be larger than in ovaries (Additional File 1: Fig. S9D-E, median length of 12.3 kb in AC testes versus 2.89 kb in AC ovaries, and median length of 13.70 in ON testes versus 4.62 kb in ON ovaries). Within Lake Malawi, median cluster lengths in testes were consistent in AC, MZ, and TM (Additional File 1: Fig. S9D). piRNA clusters are spread throughout the entire genome and do not tend to be in close proximity (Additional File 1: Figs. S8 and S9F). piRNA clusters tend to produce piRNAs from both strands, although in ovaries there is a bias for sense piRNAs (Fig. 5E-F). In terms of productivity, we found that a fraction of clusters generate the majority of the piRNAs in the library (Additional File 1: Fig. S9A). We found no relationship between the productivity and length of piRNA clusters (Additional File 1: Fig. S9G). Examples of large, highly productive piRNA clusters are shown in Fig. 5G-I.

**Cichlid piRNAs target TEs and have signatures of active silencing**

Having identified the sites of piRNA production, next we mapped the genomic features and TE classes that these sites overlap with. The majority of piRNA clusters in AC and ON overlap with intergenic regions and TEs, and the observed overlap with TEs is higher than expected by chance (Fig. 6A-B and Additional File 1: Fig. S10A). Of all TEs, LTRs are significantly enriched in piRNA clusters, whereas LINEs and DNA TEs are generally depleted (Fig. 6A-B and Additional File 1: Fig. S10A). piRNAs are generally depleted from UTRs, exons and promoters (Fig. 6A-B and Additional File 1: Fig. S10A). piRNAs that do not map to TEs may have functions beyond TE silencing, which should be explored in the future. The species-variable piRNA clusters detected in the testes of one but not the other Lake Malawi species (as shown in Fig. 5A, C) tend to follow similar enrichment trends (Additional File 1: Fig. S10B). This observation suggests that species-variable piRNA clusters do not display a clear enrichment for particular TE classes besides LTRs. Furthermore, these species-variable piRNA clusters are enriched in genomic regions of Lake Malawi cichlids associated with structural variation defined in recent work [5] (Additional File 1: Fig. S10C). This suggests the existence of piRNA-producing sequences that are structural variants in Lake Malawi cichlids. We overlapped all 24–35 nucleotide long piRNAs of AB and PN with genome features and TE classes and observed enrichments similar to those of AC and ON piRNA clusters (Additional File 1: Fig. S10D-E).

Next, we further explored sequence signatures of TE-mapping piRNAs. We found 1U and 10A signatures in piRNAs mapping sense and antisense relative to TEs, consistent with active targeting of TEs (Fig. 6C-D and Additional File 1: Fig. S10F-G). Sense piRNAs have higher 10A bias and lower 1U bias than piRNAs antisense to TEs (Fig. 6C-D and Additional File 1: Fig. S10F-G). These signatures are absent from muscle (Fig. 6C). We did not find consistent differences in piRNAs mapping to TEs between testes and ovaries across all species (Additional File 1: Fig. S10H), contrasting with the sex differences

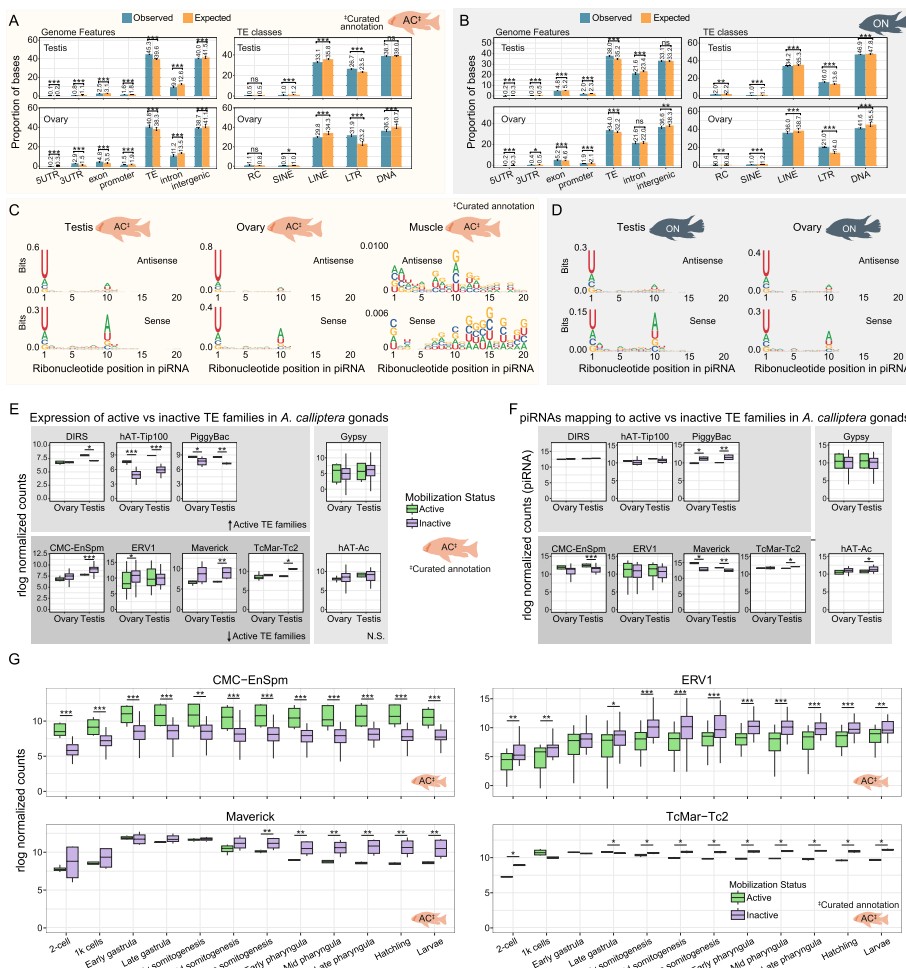

**Fig. 6** Cichlid piRNAs target TEs. **A**-**B** Observed and expected percentage overlaps of piRNA cluster sequences with genome features and TE classes in *A. calliptera* (A) and *O. niloticus* (B). ns, not statistically significant. In (A) the TE annotation was from a curated library for *A. calliptera*. The observed percentage overlaps were compared against the expected random null distributions approximated by shuffling the piRNA coordinates, with 95% confidence intervals indicated. **C-D** Sequence logos of 24–35 nucleotide long piRNAs mapping sense or antisense in regard to TE orientation in *A. calliptera* (C) and *O. niloticus* (D). The 1U and 10A signatures are observable in gonads but not in muscle. For *A. calliptera*, TE features were extracted from the curated TE annotation. **E** The mRNA expression in cichlid gonads of TE families likely to have been transpositionally active in recent evolutionary time (see Methods for a description of how this was defined) vs inactive families of the same TE superfamily. Panels above: superfamilies where the likely active families are significantly more highly expressed than the inactive families in ovaries and/or testis. Panels below: TE superfamilies with higher expression of inactive families in ovaries and/or testis. The panels on the right display TE superfamilies with no significant difference in expression between active and inactive families. *P*-values were calculated with Wilcoxon rank-sum tests (using Benjamini & Hochberg correction) comparing TE families with distinct mobilisation status in each gonad. The expression data was quantified using a curated annotation of Lake Malawi TEs. **F** piRNAs mapping to TE families likely to have been transpositionally active in recent evolutionary time (see Methods) vs inactive families of the same TE superfamily. *P*-values were calculated with Wilcoxon rank-sum tests (using Benjamini & Hochberg correction) comparing TE families with distinct mobilisation status in each gonad. piRNAs were mapped to a curated TE annotation in Lake Malawi cichlids. Disposition of panels and TE superfamilies as in (E). **G** The mRNA expression in early *A. calliptera* development of TE families likely to have been transpositionally active in recent evolutionary time (see Methods) vs inactive families of the same TE superfamily. A curated annotation of Lake Malawi TEs was used to calculate expression data. *P*-values were calculated with Wilcoxon rank-sum tests (using Benjamini & Hochberg correction) comparing TE families with distinct mobilisation status in each developmental stage. (A-B, E–G) Significance notation as follows: *$0.01 \leq$ *p*-value $< 0.05$; **$0.001 \leq$ *p*-value $< 0.01$; *** *p*-value $< 0.001$. AB, *Astatotilapia burtoni*; AC, *Astatotilapia calliptera*; ON, *Oreochromis niloticus*; PN, *Pundamilia nyererei*; rlog, regularised log; sRNA, small RNA

observed in TE expression (Additional File 1: Fig. S1C-D). These results suggest ongoing targeting of TEs by piRNAs.

We identified 15 TE families whose low allele frequencies coupled with high number of polymorphisms are strongly suggestive of recent transpositional activity in AC (Additional File 4: Table S3 and see Methods). Next, we compared expression in AC gonads of TE families with strong support for recent transpositional activity with transpositionally inactive families of the same superfamily. In two superfamilies, the expression levels did not differ between active and inactive TE families (Fig. 6E, right panel, Additional File 4: Table S3). We found three TE superfamilies in which the active families have significantly higher expression in at least one organ (Fig. 6E, upper panel, Additional File 4: Table S3). Conversely, in four TE superfamilies, we observed lower expression of the active families, in line with ongoing silencing (Fig. 6E, lower panel, Additional File 4: Table S3). Two of these TE superfamilies (CMC-EnSpm and Maverick) showed significant opposite correlation in terms of piRNA levels: active TE families were targeted by higher piRNA levels (Fig. 6F, lower panels, Additional File 4: Table S3), further supporting ongoing piRNA-driven TE silencing. Additional data agree with robust targeting of CMC-EnSpm and Maverick families by piRNAs (Fig. S10I-J). The four superfamilies with expression signatures suggestive of silencing in gonads showed dynamic expression patterns in early development (Fig. 6G). Likely active families of CMC-EnSpm (DNA TEs) are more highly expressed throughout early development than their inactive relatives (Fig. 6G), a reverse pattern to that observed in gonads (Fig. 6E, lower panel). Expression of likely active ERV1 (LTR class), Maverick and TcMar-Tc2 (DNA TEs) families did show expressions patterns consistent with the establishment of epigenetic silencing during early development (Fig. 6G). These data suggest a complex relationship between TEs and host piRNAs.

In relation to the three TE superfamilies in which active TE families were more highly expressed than their related inactive families (DIRS, hAT-Tip100, and PiggyBac), there was only an anti-correlation with piRNA levels for the PiggyBac superfamily (Fig. 6F, upper panels, Additional File 4: Table S3). In this case, the trend of higher expression and less piRNA targeting of active families may reflect that piRNA silencing of these active families has not yet been established. In line with this, during early development the active families of DIRS (LTR class), hAT-Tip100 and PiggyBac (DNA TEs) are all significantly more highly expressed than their related inactive families (Additional File 1: Fig. S11). These data support the existence of TE families likely transpositionally active in recent evolutionary time that are evading silencing by the piRNA pathway.

## Discussion

In this work, we describe four main findings that altogether suggest dynamic TE co-evolution with host control mechanisms in East African cichlids: 1) dynamic TE expression; 2) fast evolution of *piwil1* genes; 3) fast evolution of piRNA clusters; and 4) evidence of ongoing targeting of TEs by the piRNA pathway, including of TE families likely to have been transpositionally active in recent evolutionary time. We will elaborate on these points below.

First, hundreds of TEs families are dynamically expressed in gonads and early development of cichlids (Fig. 1, Additional File 1: Figs. S1 and S2). Given the extensive shared

polymorphism in African cichlids due to hybridisation [3, 4, 18, 62–65], we adopted a more conservative approach by initially quantifying TE expression at the family level. We found sex-biased expression patterns of TEs and protein-coding genes (Additional File 1: Fig. S1C-D), with higher median expression in testes versus in ovaries. This asymmetry is likely to be the result of overall higher transcriptional output in testes. Interestingly, this sex asymmetry does not consistently extend to piRNAs mapping to TEs (Additional File 1: Fig. S10H), suggesting that piRNA precursor transcription may not follow general transcription trends. Testes contain the male germline and are a relevant organ in the context of genetic conflict between TEs and host silencing factors [8, 27]. Higher gene and TE expression in testes is consistent with previous studies describing more widespread expression and increased transcriptome complexity in mammalian testes [66, 67]. TEs have been shown to contribute to transcriptome complexity in the mammalian germline [68] and in fish testes [69]. It may be worth exploring in depth whether cichlid testes, much like mammalian testes, have increased transcriptome complexity and diversity, and if this has contributed to the cichlid radiations. Indeed, gonad transcriptomes are evolving faster than transcriptomes of other organs in Lake Tanganyika cichlids [70]. In early development, we found that the majority of TE families exhibit higher expression during gastrulation, a period that may coincide with the maternal-to-zygotic transition in cichlids. Zygotic transcription of TE silencing factors may initiate concomitant to the onset of general zygotic transcription, leading to zygotic TE silencing. Expression of transpositionally active Maverick, ERV1, and TcMar-Tc2 TEs in early development could illustrate just that (Fig. 6G). Expression analysis of individual TE loci revealed TEs with expression in discrete developmental times (Fig. 1C and Additional File 1: Fig. S2D). As expected given their evolutionary distance, the TE classes enriched in particular developmental stages in cichlids differ substantially from those enriched in the same developmental stages in zebrafish [13]. However, a striking similarity to zebrafish is enrichment of TEs belonging to ERV1, Ty3, and Pao LTR superfamilies in gastrula stages (Fig. 1C and Additional File 1: Fig. S2D). Such similarities in expression might suggest that TEs of those LTR superfamilies are required in specific gene regulatory networks, through regulation of host gene expression. It will also be relevant to investigate how the maternal-to-zygotic transition and/or epigenetic reprogramming affect LTR transcription and transposition during early fish development.

Second, we find an expanded repertoire of *piwil1* genes in Lake Malawi cichlids and signatures of positive selection on the novel copies (Fig. 2, Additional File 1: Fig. S5A, and Additional File 2: Table S1). Lability in copy number and positive selection on TE silencing factors are two signatures associated with arms races between TEs and their animal hosts [24, 25, 40–43]. These findings also add to the notion that piRNA pathway factors, including *piwil1* genes, evolve fast in teleosts [47, 49]. Interestingly, TEs, the targets of Piwi proteins, likely have mediated, at least partially, the expansion of *piwil1* genes in Lake Malawi cichlids. We found closely related PiggyBac elements associated with the three novel *piwil1* genes, but not with the *piwil1* copy sharing synteny with other vertebrate *piwil1* genes, presumably the original copy (Fig. 2, Additional File 1: Fig. S4, and Additional File 2: Table S1). We also found TIRs flanking the PiggyBac and putative TIR fragments distal to the PiggyBac and 5' to the *piwil1* copies. The non-coding differences of the four *piwil1* genes suggest the succession of events underlying the

expansion: first a duplication of *piwil1.1* creating *piwil1.2*, followed by creation of one of the truncated copies from *piwil1.2*, and its subsequent duplication (Fig. 2D). Given the PiggyBac TIR signatures, it is likely that at least the first duplication was mediated by transposition, but we cannot exclude that subsequent duplications were driven by a recombination-based mechanism. By leveraging available genomic resources we determined that *piwil1.1* and *piwil1.4* seem to be fixed or nearly fixed in Lake Malawi, whereas *piwil1.2* and *piwil1.3* are less widespread (Fig. 2C). *piwil1.3* seems to have negligible expression in the germline and early development (Fig. 3A-B). It is possible that *piwil1.3* is expressed and functional in other organs beyond the gonads and brain, or in juvenile developmental stages between larval stage and sexual maturity. An alternative is that *piwil1.3* is a pseudogene, similar to *piwil1.2*. Although we presented gene expression data of *piwi* paralogs in AC individuals, future studies could entail a comprehensive profiling of *piwi* expression across Lake Malawi to better understand the roles of these paralogs at a population scale.

The exact function of *piwil1.3* and *piwil1.4* remains to be determined. Knock-outs of *piwil1.1* and *piwil1.4* will be key to inform on their function. The annotated Piwil1.3 and Piwil1.4 proteins are predicted to encode a catalytically competent PIWI domain (Figs. 2A-B and 3D-E), the catalytic centre of Argonaute proteins responsible for slicer activity [58, 59]. The Argonaute domains lacking in Piwil1.3 and Piwil1.4, the MID and PAZ domains (Fig. 2B), are predicted to serve as binding pockets for the 5' and 3' ends of the piRNA, respectively [71–73]. Without these domains, Piwil1.3 and Piwil1.4 are most likely not able to bind to piRNAs or other sRNAs, and will probably function independently of piRNAs. Thus, these truncated Piwi proteins were likely repurposed for a piRNA-independent gene regulatory role, related, or not, to TE silencing.

Third, we find fast evolution of piRNA clusters in cichlids. The majority of piRNAs were produced from intergenic regions and TEs (Figs. 5 and 6) and 65–80% of these sequences can be grouped into discrete piRNA-producing clusters (Additional File 1: Fig. S9A). We identify piRNA clusters with sex-biased expression, and, interestingly, variation in piRNA clusters even in testes of closely related Lake Malawi cichlids (Fig. 5A, C). These observations indicate that piRNA clusters are fast-evolving modules in Lake Malawi. An in-depth population-wide analysis of piRNA populations and piRNA clusters in Lake Malawi will be useful to determine just how rapidly these units are evolving in cichlids. In terms of piRNA biogenesis, we find conserved differences in cichlid piRNA populations with peaks at 26–27 nucleotides long piRNAs in testes versus 28–29 nucleotide long piRNAs in ovaries (Fig. 4). These piRNA size differences may be driven by the relative amounts of Piwi Argonautes in gonads and their favoured piRNA length. The most striking difference in terms of piRNA biogenesis however, is the lack of consistent phasing signature in the ovaries of East African cichlids outside Lake Malawi, which is not correlated with the expression of *pld6* and *mov10l1*, factors required for phased biogenesis in other species [27] (Fig. 4, Additional File 1: Figs. S3 and S6). It will be interesting to determine the factor(s) inhibiting or inactivating phased biogenesis in cichlid ovaries.

Fourth, piRNAs and the identified piRNA clusters substantially overlap with TEs (Fig. 6A-B and Additional File 1: Fig. S10A-B). TE-mapping piRNAs display 1U and 10A signatures consistent with ongoing piRNA biogenesis from TE template transcripts.

The relationship between piRNA levels and expression of given families is complex, in particular regarding the transpositional activity status of the families. For the active families of CMC-EnSpm and Maverick TE superfamilies, the anti-correlation between expression and piRNA levels in gonads is suggestive of ongoing epigenetic silencing. The anti-correlation observed for piRNAs and expression of active PiggyBac families is also worthy of further exploration, as it may reflect active PiggyBacs evading the piRNA pathway. These dynamics could support ongoing PiggyBac transposition in Lake Malawi cichlids and the participation of these TEs, at least partially, in *piwil1* expansion. It will be relevant to revisit the interplay between TEs and piRNAs in an experimental setup where transposition can be experimentally verified.

It is relevant to note that besides the piRNA pathway, other pathways are known to be required for TE silencing, including the human silencing hub (HUSH) complex [28, 29], and Krüppel-associated box (KRAB) domain-containing zinc finger proteins [8, 24]. Fishes have homologs of HUSH complex factors [8], and although the transcriptional repressor KRAB domain is absent in fishes, a recently identified FiNZ domain is associated with zinc fingers in fishes and these proteins were proposed to act analogously to KRAB domain-containing zinc finger proteins [30]. It is important to determine whether these factors are relevant to silence particular classes of TEs in specific tissues/organs or periods of development, and whether these complement or collaborate with the piRNA pathway.

Three sets of observations point towards TEs as key genetic elements contributing to cichlid diversification: 1) TEs represent a previously underestimated source of genetic diversity in African cichlids [5]; 2) TEs have been linked with pigmentation and vision traits, sex determination, and gene expression changes [18, 20–23]; and 3) the ongoing dynamic TE-host co-evolution and arms races that our findings suggest. It remains unclear how the latter connects with cichlid phenotypic diversification. We expect it does not come down to the number of TE families or the proportion of the genome comprised by TEs. In this regard, zebrafish provides a much more striking example, with nearly 2,000 distinct TE families, occupying more than 50% of its genome [6, 13], versus 557–828 TE families and 16–41% of the genome in cichlids (Additional File 1: Figs. S1B and S2A). However, the *Danio* genus of zebrafish did not diversify nearly as prolifically as East African cichlids despite its massive TE content [6, 14, 16].

What led to the unparalleled rates of phenotypic diversification observed in East African cichlids? Recent work on the cichlid radiation of Lake Victoria suggests that ecological versatility is the key [74, 75]. Key features contributing to cichlid versatility include strong sexual selection, highly plastic jaw structures, and abundant interspecific hybridisation [1, 74]. The regulatory consequences of hybridisation are one possible avenue to pursue to study the influence of TE-host co-evolution in cichlid radiations. Genomic studies have elucidated a complex evolutionary history of East African cichlids, marked by substantial amounts of gene flow occurring through hybridisation [4, 18, 62–65]. It will be important to determine how interspecific cichlid hybrids tolerate regulatory mismatches driven by genetic conflict between TEs and the piRNA pathway.

## Conclusions

This study is the first in-depth profiling of TE and piRNA expression in cichlid species representative of the East African Great Lakes, using genomic, transcriptomic, and proteomic data. We profiled gonads and early development and found evidence consistent with TEs and host epigenetic silencing pathways engaged in conflict. In addition to serving as a valuable new resource, this research provides an initial understanding of TEs and piRNAs as two co-evolving modules, creating a platform to investigate additional hypotheses on their roles in African cichlid radiations. Going forward, learning about the co-evolution of these modules in the context of recurring hybridisation has the potential to give valuable insights into the genetic and molecular basis of cichlid diversification.

## Methods

### Animal sampling and housing conditions

*Astatotilapia calliptera* and *Tropheops* sp. 'mauve' animals were grown in 220 Litre tanks, with pH 8, at approximately 28 °C, and with a 12 h dark/light cycle. Males and females of each species were housed only with conspecifics. Feeding, housing, and handling were conducted in strict adherence to local regulations and with the protocols listed in Home Office project license PP9587325. Fish were fed twice a day with cichlid flakes and pellets (Vitalis). Tank environment was enriched with plastic plants, plastic hiding tubes, and sand substrate. Aquaria grown animals were euthanised with approved Home Office schedule 1 protocols, namely using 1 g/L MS-222 (Ethyl 3-aminobenzoate methanesulfonate, Merck #E10521) and subsequent exsanguination by cutting the gill arches, in accordance with local regulations. Afterwards, gonads, brain and dorsal muscle tissue were carefully dissected, swiftly snap frozen in dry ice and stored at approximately −80 °C.

Dominant adult male *Maylandia zebra* bred and raised in captivity were obtained from commercial supplier Kevs Rifts and culled in Cambridge animal facilities, following an ethically approved post–transport adjustment period. *M. zebra* animals were euthanised using approved Home Office schedule 1 protocols as above. *Pundamilia nyererei* animals were raised in stock tanks of dimensions 59 cm(L) × 45 cm(B) × 39 cm(H) and moved to larger tanks 177 cm (L) x 45 cm(B) x 39 cm(H) once they reached approx. 7 cm long. Temperatures were kept at 26 °C, with constant daily water change of about 10% and 12:12 light dark regime. Frozen tissue samples of *Astatotilapia burtoni* were provided by Hans Hofmann and Caitlin Friesen (University of Texas at Austin, Austin, TX, USA). *Oreochromis niloticus* frozen tissue samples were provided by David Penman, Alastair McPhee, and James F. Turnbull (Institute of Aquaculture, University of Stirling, Stirling, Scotland, UK).

### Orthology analysis

To identify orthologs of conserved factors involved in TE silencing pathways, we used OrthoFinder [76, 77] v2.3.12. We used Ensembl proteomes (downloaded on 02/06/2020) of Homo sapiens (GRCh38), *Mus musculus* (GRCm38), *Oryzias latipes* (ASM223467v1), *Danio rerio* (GRCz11), *Takifugu rubripes* (fTakRub1.2), *Gasterosteus aculeatus* (BROADS1), *Amphilophus citrinellus* (Midas_v5), *Oreochromis aureus* (ASM587006v1),

*Oreochromis niloticus* (O_niloticus_UMD_NMBU), *Astatotilapia burtoni* (AstBur1.0), *Neolamprologus brichardi* (NeoBri1.0), *Pundamilia nyererei* (PunNye1.0), *Astatotilapia calliptera* (fAstCal1.2) and *Maylandia zebra* (M_zebra_UMD2a). OrthoFinder was run on proteomes containing the longest protein isoform, parsed using a script provided with OrthoFinder (https://github.com/davidemms/OrthoFinder/blob/master/tools/primary_transcript.py). Initially, we ran OrthoFinder with the fish genomes above as inputs (except *M. zebra*), using option -f. Afterwards, we added human, mouse, and an additional Lake Malawi cichlid species *M. zebra* to this analysis using options -b and -f. We subsequently pinpointed the orthogroups containing known human, mouse and zebrafish TE silencing factors and extracted the gene IDs of their cichlid orthologs.

### Piwil1 evolutionary analysis

Piwil1 protein orthologs were identified with OrthoFinder (see Orthology analysis above). Schematic of domain structure of Piwil1 proteins was plotted in R [78], with packages drawProteins [79] and tidyverse [80]. Coordinates of the MID domain were manually added to Piwil1 proteins, as this information was not present in Uniprot, which drawProteins relies on. MID domain coordinates in *A. calliptera* Piwil1 proteins were inferred from the MID domain coordinates of zebrafish Ziwi in Uniprot, through a multiple sequence alignment of *A. calliptera* Piwil1 proteins and Ziwi.

To determine the presence and absence of *piwil1* copies and their 3' trailing Piggy-Bac-1 TEs across Lake Malawi cichlid eco-morphological groups and genera, we probed the reads of 74 previously published short-read genomes [4], 5 new short-read genomes, as well as 12 long-read genomes (Additional File 2: Table S1). Short-read genomes were aligned to the *A. calliptera* reference genome (fAstCal1.2, GCA_900246225.3) using bwa mem v0.7.17-r1188 (arguments: -C -p) using default settings [81]. Using samtools v1.9 [82], the resulting alignment files were then further processed with fixmate (arguments: -m), sort (arguments: -l0) and mardup. Long-read genomes were aligned to the same reference using minimap2 v2.17-r974-dirty [83] (arguments: -ax map-pb --MD) and then sorted and indexed using samtools v 1.16–9-g99f3988. We manually checked whether read alignments showed robust support in specific eco-morphological groups/genera for the presence of each *piwil1* paralog and 3' trailing piggyBac copy using IGV v2.9.4 [84]. Next, we manually determined the exact features of these regions using the *piwil1* gene annotations of fAstCal1.2 [85], our TE annotation created from a curated TE library (see section Transposable element annotations), and genomic alignments of the entire regions encompassing all *piwil1* paralogs. Initial alignments of the paralog loci were generated by aligning the fAstCal1.2 reference genome to itself using Winnowmap2 [86] (options: -ax asm5 --MD). Potential stop codons in *piwil1* paralogs were assessed in a multiple sequence alignment between *piwil1.1*, *piwil1.2* (reverse complement), *piwil1.3*, and *piwil1.4* (reverse complement) genomic regions, which was created MUSCLE v3.8.31 [87] using default settings and then curated manually in AliView v1.27 [88]. The exons of ENSACLT00000021959, the canonical ENSEMBL isoform of *piwil1.1*, the best evolutionarily conserved *piwil1* gene, was projected to the aligned sequences of the paralogs. A second alignment was created analogously, which additionally included the homologous *piwil1* sequence from *Oreochromis niloticus*. Based on the latter alignment, we calculated Hamming distances (github.com/ssciwr/hammingdist) separately

for intronic and exonic regions and built neighbour joining trees (github.com/scikit-bio/scikit-bio). Alignment files can be found online [89]. The *A. calliptera* TE annotation created from a curated TE library was used to identify the PiggyBac-1 TE and its complete terminal inverted repeats (available in DFAM [53]: https://dfam.org/family/DF003 571810/model). We used BLASTN v2.16.0 (blastn-short options -evalue 10 -word_size 7 -gapopen 1 -gapextend 1) with the complete TIR sequences as queries and 50 kilobase regions of the *A. calliptera* genome encompassing the *piwil1* genes as subjects to identify TIR sequences neighbouring *piwil1* genes. The BLASTN results are included in Additional File 2: Table S1 and the 50 kilobase *piwil1* regions used as search subjects were deposited online [89]. Due to the lack of high confidence BLAST hits for the complete TIR upstream of the *piwil1* genes, we instead searched for 5'-TTAACCCTT-3' sequences (with the PiggyBac TTAA target site and extremity of the TIR) upstream of the *piwil1* genes, allowing one mismatch. All hits we obtained using this approach were annotated in Fig. 2A. We quantified the number of TTAA motifs in the 5 kb flanking regions of the *A. calliptera piwi* genes and performed a Welch's t-test to compare these to the distribution of TTAA motifs in the flanking regions of all other genes, finding no significant differences ($p$-value $= 0.8552$). We also quantified the total amount of TTAA motifs in each gene, including introns, normalized by length of the gene. The *piwi* genes seem to have a slightly larger quantity of TTAA motifs than other genes (Welch's t-test, $p$-value $= 0.01754$), but this could be simply due to the number of introns, as a large number of annotated genes have no introns and seem to be depleted of TTAA motifs.

For the selection analysis we restricted our existing callset of more than 2,000 whole-genome sequenced Lake Malawi cichlids (github.com/tplinderoth/cichlids/tree/master/callset), which are all aligned against the chromosome level fAstCal1.2 reference genome [85], to the 79 individuals used in Fig. 2C (Additional File 2: Table S1). The subset was generated with bcftools view [90], v.1.16–9-g99f3988) (arguments --types snps -m2 -M2 -f PASS -S $sample_list) to retain exclusively biallelic SNPs that passed all filters. Chromosome-scale VCFs along with the four largest contigs (>1 Mbp) were concatenated into a single VCF using bcftools concat and served as the input for the selection analysis. A selection scan was performed using Raised Accuracy in Sweep Detection (RAiSD) v2.9 [50] (with arguments: -f -M 3 -y 2 -m 0 -R -I). We used the μ values obtained by RAiSD analysis to calculate a per-gene μ. This was done by traversing the RAiSD output, intersecting it with the start/end of the largest transcript of every gene (only including "RAiSD windows" fully enclosed by the transcript start/end coordinates), and calculating the median value. PiggyBac-1 sequences adjacent to *piwil1* genes were extracted according to their annotation coordinates, and aligned with the PiggyBac-1 family consensus from the curated TE library using MAFFT v7.475 [91] with option --auto. L-INS-i was the alignment method automatically selected. Alignment visualisation was optimised in Jalview v2.11.2.7 [92]. To expand the analysis and identify high quality copies, we extracted all the PiggyBac-1 sequences annotated in the *A. calliptera* reference genome on scaffolds ≥ 1 Mbp (according to the curated TE annotation) with a RepeatMasker SWscore > 1000 (sequences available online [89]), and aligned them with MUSCLE v3.8.31 [87]. We further filtered the alignment to contain only the region encompassed by the PiggyBac1 elements associated with *piwil1.2*, *piwil1.3*, and *piwil1.4*, and removed alignment columns consisting almost exclusively of missing data (filtered alignment with

315 high quality copies available online [89]). A phylogenetic tree was constructed with IQ-TREE v2.1.2 [93], option -B 1000. TPM2 + F + R2 was the best fit model. Trees were visualised and annotated in FigTree v1.4.4 (https://github.com/rambaut/figtree).

The sequences of Piwil1 protein orthologs were collected from Ensembl. For *Piwil1* genes encoding more than one protein isoform, the longest isoform was chosen for analysis. As *A. calliptera piwil1.2* may be a pseudogene, we did not include its predicted protein sequence in the subsequent analysis. Fish Piwil1 proteins were aligned with MAFFT v7.475 [91], using option --auto, and L-INS-i was the alignment method automatically selected. We trimmed the alignment manually, keeping only 296 sites corresponding to the C-terminal region of the proteins with excellent alignment score, which includes the PIWI domain. Original protein sequences and alignment files can be found online [89]. IQ-TREE v2.1.2 [93] was used to construct phylogenetic trees from these two alignments with options -B 10000 -o {medaka and zebrafish Piwil1 proteins were defined as outgroups}. -B parameter refers to ultrafast bootstrap approximation [94]. PMB + G4 was the best fit model. To test for selection, we redid the alignment using a smaller subset of the proteins, including only Piwil1 proteins of African cichlids. Alignment of protein sequences was performed with MAFFT v7.475 [91], using option --auto. L-INS-i was the chosen alignment model. Next, we used pal2nal v14 [95] to produce a reverse alignment from an alignment of the protein sequences to an alignment of the coding sequences. The resulting reverse alignment was used as input for selection tests in Datamonkey [96]. A gene-wide test was first performed using Branch-site Unrestricted Statistical Test for Episodic Diversification (BUSTED) [97]. We conducted the test in two ways, testing for selection across all branches and testing for selection only in radiating cichlids, with *O. niloticus* as an outgroup. BUSTED reported very strong support for positive selection in both cases (*p*-value = 0). Additional File 1: Fig. S5A shows the subsequent analysis to identify residues very likely to be under positive selection according to Mixed Effects Model of Evolution (MEME) [98]. In addition, we tested these same protein-coding sequences for positive selection using the branch-site test implemented in codeml from the PAML package v4.10.7 [99, 100]. Control files and input alignments used can be found online [89].

To pinpoint catalytic residues of cichlid Piwil1 proteins, we first added the sequence of human PIWIL1 (HIWI) to the list of fish Piwil1 proteins used in the alignments above, and redid the alignment using MAFFT v7.475 [91] with option --auto (L-INS-i was the model automatically chosen). The alignment was visualised in Jalview v2.11.2.7 [92] and the catalytic residues were manually pinpointed based on their known positions in HIWI [58, 59]. Structural alignments were performed with open-source PyMOL v2.5.0 using the align command. We aligned AlphaFold predictions of Piwil1.1 (Uniprot ID A0A3P8PWP0), Piwil1.3 (Uniprot ID A0A3P8NS09), and Piwil1.4 (Uniprot ID A0A3P8NRZ4) of *A. calliptera*, downloaded from AlphaFold Protein Structure Database [101, 102], with crystal structures of *bombyx mori* Siwi (PDB ID 5GUH) [56] and *Drosophila melanogaster* Piwi (PDB ID 6KR6) [57]. As we focus on the PIWI domain, we aligned only the PIWI domains of *A. calliptera* Piwil1.1 (residues 550–856), *D. melanogaster* Piwi (residues 537–843), and *B. mori* Siwi (residues 593–899). As Piwil1.3 and Piwil1.4 of

*A. callliptera* are truncations encompassing only the Piwi domain, we used their full-length structure for the alignments.

### RNA extractions

Frozen brain, muscle, and gonad tissues were partitioned on a mortar positioned on dry ice, quickly to avoid thawing, and weighed. Biological replicates were created by collecting a similar mass of the same organ/tissue from size-matched individuals of the same species. 15–30 mgs of brain tissue, 26 mg of dorsal muscle tissue, and 14–144 mgs of gonad tissue were used, according to the specific tissue, tissue availability, and size of the specimen, which varied per species. Tissue pieces were transferred to BeadBug tubes prefilled with 0.5 mm Zirconium beads (Merck, #Z763772) and 500–600 µl of TRIzol (Life Technologies, #15596026) was added to the tubes and mixed vigorously. Afterwards, we conducted the homogenisation using a BeadBug microtube homogeniser (Sigma, #Z764140) at approximately 4 °C (in cold room). Each sample was homogenised with five BeadBug runs at maximum speed (4,000 rpm) for 60 s each. No sample was run on BeadBug more than two consecutive times to avoid overheating. Other than the run time inside the BeadBug, samples were left on ice. After homogenisation, lysates were centrifuged for 5 min at 18,000 G at 4 °C. Supernatant was then removed into a clean 1.5 mL tube. Centrifuged the lysates again, this time at maximum speed (approximately 21,000 G) for 5 min at 4 °C. Transferred supernatant into a clean tube without disturbing the pellet and tissue debris. Mixed supernatant thoroughly 1:1 with 100% ethanol, pipetted the mix into a column provided in the Direct-zol RNA Miniprep Plus kit (Zymo Research, #R2072) and followed manufacturer's instructions, using the recommended in-column DNase I treatment.

### Library preparation and sequencing

#### mRNA sequencing

Library preparation (directional, with poly-A enrichment) and sequencing (Illumina, PE150) of *A. calliptera, M. zebra, T.* sp. 'mauve', *A. burtoni,* and *O. niloticus* gonads was performed by Novogene. Libraries of *P. nyererei* gonads and *A. calliptera* brain tissues were prepared and sequenced as follows. Initial quality control was done using a Qubit Fluorometer (Invitrogen) and Qubit RNA HS Assay Kit (Invitrogen, #Q32855), and Agilent RNA TapeStation reagents (Agilent, #5067–5576; #5067–5577; #5067–5578). 50–250 ng of total RNA were used for library production with the NEBNext® Poly(A) mRNA Magnetic Isolation Module (NEB, #E7490), in conjunction with the NEBNext® Ultra™ II Directional RNA Library Prep Kit for Illumina® (NEB, #E7760) and the NEBNext® Multiplex Oligos for Illumina® (96 Unique Dual Index Primer Pairs, NEB #E6440). Quality control of the libraries was done with the Qubit dsDNA HS Assay Kit (Invitrogen, #Q32854) and Agilent DNA 5000 TapeStation reagents (Agilent, #5067–5588; #5067–5589). Samples were then pooled in equimolar amounts according to the TapeStation results and sequenced on a NovaSeq 6000 system (PE150 on one lane of an S1 Flowcell).

### Small RNA sequencing

Initial quality control was conducted using a Qubit Fluorometer (Invitrogen) and the Qubit RNA HS Assay Kit (Invitrogen, #Q32855), and Agilent RNA TapeStation reagents (Agilent, #5067–5576; 5067–5577; 5067–5578). Samples were processed according to the NEXTFLEX® Small RNA-Seq Kit v4 with UDIs (PerkinElmer, #NOVA-5132–32) with a 1 μg starting input and 12 cycles of PCR. Quality control of the libraries was done with Qubit dsDNA HS Assay Kit (Invitrogen, #Q32854) and Agilent DNA 5000 TapeStation reagents (Agilent, #5067–5588; #5067–5589). Samples were then pooled in equimolar amounts according to the TapeStation results and sequenced on a Novaseq 6000 system (PE50 on one lane of an SP Flowcell).

## Transposable element annotations

In each respective cichlid genome, transposable elements and repeats were first modelled and identified using RepeatModeler v1.0.11 in combination with the recommended programmes RECON v1.08, RepeatScout v1.0.6, TRF v4.0.9 and NCBI-RMBlast v2.14, and then annotated using RepeatMasker v4.0.9 in combination with NCBI-RMBlast v2.14, TRF v4.0.9 and the custom libraries of modelled repeats, Dfam3.0 and Giri-Repbase-20170127 [103]. The curated TE library for Lake Malawi cichlids was created following a previously described protocol [104], it is available through DFAM [53], and will be described in detail elsewhere (P. Sierra & R. Durbin, in preparation). This library was used as input to RepeatMasker v4.1.2-p1 [103] with options -e rmblast -no_is -gff -lib -a to generate a final TE annotation for the *A. calliptera* genome fAstCal1.2. GTF files with TE annotations amenable to be used for TEtranscripts (see below Bioinformatic analysis, mRNA-sequencing analysis section) were created using custom scripts (available online [89]).

## Inference of transpositional activity of transposable elements from genomic data

To compare the genetic diversity of TEs in the different *A. calliptera* populations (from Lake Masoko or Lake Kingiri), we calculated the intrapopulation TE diversity $\pi_{TE}$ and interpopulation TE diversity $D_{xyTE}$, based on the allele frequencies of the polymorphisms in a similar way as described in Chase et al., 2021 [105], but using biallelic TE insertions rather than single nucleotide polymorphisms. p denotes the frequency of the reference allele and q the frequency of the alternate allele, S is the number of TE polymorphisms and L is the length of the genome:

$$\pi_{TE} = \frac{\sum_{i=1}^{S} 2p_i q_i}{L}$$

The interspecies TE diversity uses a similar formula where 1 and 2 refer to different species:

$$D_{xyTE} = \frac{\sum_{i=1}^{S} p_{1i}q_{2i} + p_{2i}q_{1i}}{L}$$

Then, the top 10 TE families with highest values of the $\frac{\pi_{TE}}{D_{xyTE}}$ ratio in either Lake Masoko or Lake Kingiri populations of *A. calliptera* were selected as having strong signs

of transpositional activity in recent evolutionary time (see Additional File 4: Table S3). We compared gene expression of these likely active TE families with gene expression of their related TE families (of the same superfamily), using DESeq2-normalised regularised log counts for each TE family (output of TEtranscripts, see below).

### Bioinformatic analysis

#### mRNA-sequencing analysis

Illumina adapters and reads with low-quality calls were filtered out with Trimmomatic v0.39 [106] using options SLIDINGWINDOW:4:28 MINLEN:36. Quality of raw and trimmed fastq files was assessed with fastQC v0.11.9 (https://www.bioinformatics.babraham.ac.uk/projects/fastqc/) and summarised with multiQC v1.11 [107]. Gene expression was quantified from trimmed reads using salmon v1.5.1 [108], with options --seqBias --gcBias --validateMappings -l A. Salmon indexes were prepared for each species separately, and used as input (in the -i option) for gene expression quantification in the respective species. DESeq2 [109] and custom scripts (available online [89]) were used to calculate normalised and TPM counts, generate plots and conduct statistical tests on an R framework [78]. See R packages used below, in the end of this section.

Trimmed fastq files were mapped to the cichlid genomes using HISAT2 v2.2.1 [110] with options -x -1 -2 -S. Reads from *A. burtoni*, *P. nyererei* and *O. niloticus* were mapped to their respective Ensembl genomes (AstBur1.0, GCA_000239415.1; PunNye1.0, GCA_000239375.1; O_niloticus_UMD_NMBU, GCA_001858045.3). Reads from all Lake Malawi cichlid species used (*A. calliptera*, *M. zebra* and *T.* sp. 'mauve') were mapped to *A. calliptera* Ensembl genome fAstCal1.2 (GCA_900246225.3). Mapping statistics in Additional File 5: Table S4. SAM alignment files were converted to BAM format, sorted and indexed with samtools v1.10 [82]: 1) samtools view -bS; 2) samtools sort; and 3) samtools index. To create bigwig files, the BAM alignment files were used as input to bamCoverage v3.5.1, part of the deepTools package [111], using options --normalizeUsing CPM -of bigwig --binSize 10. Bigwig files of biological replicates of the same organ were combined using WiggleTools [112] mean and wigToBigWig v4 [113]. Genome tracks were plotted with custom scripts (available online [89]) using the Gviz [114] and GenomicFeatures [115] packages on an R framework [78].

To quantify TE expression at the TE family level, we mapped trimmed reads using STAR v2.5.4b [116] with options --readFilesCommand zcat --outSAMtype BAM SortedByCoordinate --outFilterType BySJout --outFilterMultimapNmax 150 --winAnchorMultimapNmax 150 --alignSJoverhangMin 8 --alignSJDBoverhangMin 3 --outFilterMismatchNmax 999 --outFilterMismatchNoverReadLmax 0.04 --alignIntronMin 20 --alignIntronMax 10,000,000 --alignMatesGapMax 100,000,000. As above, reads from *A. burtoni*, *P. nyererei* and *O. niloticus* were mapped to their respective Ensembl genomes (AstBur1.0, GCA_000239415.1; PunNye1.0, GCA_000239375.1; O_niloticus_UMD_NMBU, GCA_001858045.3) and reads from all Lake Malawi cichlid species used (*A. calliptera*, *M. zebra* and *T.* sp. 'mauve') were mapped to *A. calliptera* Ensembl genome fAstCal1.2 (GCA_900246225.3). Mapping statistics in Additional File 5: Table S4. The resulting BAM files were used as inputs for TEtranscripts v2.2.1 [117] with options --stranded reverse --SortByPos. TEtranscripts was run separately for each species, using gene annotations of the respective species downloaded from Ensembl (March 2021) and TE annotations described above (see Transposable element annotations

section). For Lake Malawi cichlids, TEtranscripts was ran using *A. calliptera* gene and TE annotations (both default and curated versions). A TE family was defined as having detectable expression if it had > 10 counts in at least 2 samples. This is a low cutoff purposely used to identify TE families with detectable expression. Detectable levels of expression should not be interpreted as high levels of expression. DESeq2 [109] and custom scripts (available online [89]) were used to calculate normalised counts, generate plots and conduct statistical tests on an R framework [78]. We have used the following R packages: tidyverse [80], lattice [118], eulerr [119], genefilter [120], pheatmap [121], reshape2 [122], ggrepel [123], biomaRt [124], tximport [125], RColorBrewer [126], ashr [127], ggpubr [128], GenomicFeatures [115], patchwork [129].

### mRNA-sequencing analysis of Lake Malawi cichlid embryogenesis datasets

The embryogenesis datasets will be reported in detail elsewhere (Chengwei Ulrika Yuan & Eric A. Miska, in preparation). Data and metadata (with information on dataset collection and experimental design) are publicly available, see Availability of data and materials section below. Trimmomatic-0.39 [106] was used to trim the Illumina adapters. Salmon v0.14.2 [108] was used to quantify expression of protein-coding genes (--seqBias --validateMappings --gcBias). TEtranscripts analysis on embryo samples was performed as described above (mRNA-sequencing analysis subsection), with one exception: option –stranded no. Locus-specific TE expression levels were analysed with SQuIRE (v0.9.9.9a-beta) [130]. For squire Count the option --strandness '0' was run as default for unstranded Illumina data. Reads were mapped to the *A. calliptera* genome (Ensembl, fAstCal1.2), and the TE annotation created from the curated TE library was used (see above, Transposable element annotations section). Tot_counts was used in downstream analysis from the Squire output. Only expressed TEs were kept (defined as > 5 reads in at least 2 samples). Heatmap and enrichment plots were made from SQuIRE output with code adapted from Chang et al., 2022 [13].

### Small RNA-sequencing analysis

CutAdapt v1.15 [131] was used to remove adapters and reads shorter than 18 nucleotides with options -a TGGAATTCTCGGGTGCCAAGG --minimum-length 18. Quality of raw and trimmed fastq files was assessed with fastQC v0.11.9 (https://www.bioinformatics.babraham.ac.uk/projects/fastqc/) and summarised with multiQC v1.11 [107]. Of note, ovary samples were sometimes problematic, and did not pass QC, and thus were excluded from subsequent analysis. This is most likely due to the vast amount of yolk in ovary, which may interfere with RNA purification. Next, we mapped the trimmed reads to the genome using STAR v2.5.4b [116], with options --readFilesCommand zcat --outMultimapperOrder Random --outFilterMultimapNmax 100 --outFilterMismatchNmax 2 --alignIntronMax 1 --outSAMtype BAM SortedByCoordinate --outFilterType BySJout --winAnchorMultimapNmax 100 --alignEndsType EndToEnd --scoreDelOpen −10,000 --scoreInsOpen −10,000 --outSAMmultNmax 1 --outFileNamePrefix. As above, reads from *A. burtoni*, *P. nyererei* and *O. niloticus* were mapped to their respective Ensembl genomes (AstBur1.0, GCA_000239415.1; PunNye1.0, GCA_000239375.1; O_niloticus_UMD_NMBU, GCA_001858045.3) and reads from all Lake Malawi cichlid species used (*A. calliptera*, *M. zebra* and *T.* sp. 'mauve') were mapped to *A. calliptera*

Ensembl genome fAstCal1.2 (GCA_900246225.3). Mapping statistics in Additional File 5: Table S4. An in-house custom script [132, 133] was used, with the BAM files of the alignment as inputs, to create sRNA length distribution profiles in the range of 18–36 nucleotides, and to report 5'-nucleotide frequency, normalised to all mapping reads. The script creates separate sRNA length distribution profiles for 1) collapsed and 2) uncollapsed reads. The first profile keeps only one read of each unique sequence to remove abundance bias, while the second profile keeps all reads. Lastly, the script also produces a FASTA file with the collapsed sequences. With the outputs of the scripts, plots of sRNA length distribution profiles and first nucleotide composition plots were created on an R framework [78] with the packages tidyverse [80], reshape2 [122], and RColor-Brewer [126].

Next, we selected sRNAs in the piRNA size range, between 24 and 35 nucleotides long, for further analysis. We have done this size selection on the trimmed reads using CutAdapt v1.15 [131] with options --minimum-length 24 --maximum-length 35. We mapped 24–35 nucleotides long sRNAs to the genome with the same settings as discriminated in the previous paragraph (mapping statistics in Additional File 5: Table S4). Next, we used "Small RNA Signatures" v3.5.0 [134] of the Mississippi Tool Suite from the web-based analysis tool Galaxy (available here: https://mississippi.sorbonne-universite.fr/) to calculate z-scores of overlapping sRNA pairs. For this analysis, alignment BAM files of 24–35 nucleotide long reads were used as input, along with following options: Min size of query sRNAs 24, Max size of query sRNAs 35, Min size of target sRNAs 24, Max size of target sRNAs 35, Minimal relative overlap analyzed 1, Maximal relative overlap analyzed 26. To find signatures of phased piRNA biogenesis, BAM files of 24–35 nucleotide long reads were loaded into R as Genomic Ranges [115] and using RSamtools [135], the Follow function was used to identify the next mapping piRNA pair and distances between the 5' and 3' were calculated for plotting. To create sequence logos, we first ran the custom script described above [132, 133] to produce a FASTA file with the 24–35 nucleotide long collapsed reads (unique sequences). Then, we created a new FASTA file with all these reads trimmed from the 3' end to a total length of 20 nucleotides, and concatenated together the FASTA files of the biological replicates for each species and organ. The FASTA file with the concatenated and trimmed sequences was in turn used to generate sequence logos in R (scripts available online [89]), with packages ggseqlogo [136], phylotools [137], and tidyverse [80]. This process was repeated to generate sequence logos of piRNAs mapping sense or antisense in regard to TE orientation using BAM files with 24–35 nucleotide long reads, which were created as follows: 1) samtools view -b -f (16 or 0); 2) bedtools intersect (-s or -S); 3) samtools merge; 4) samtools sort; 5) samtools index.

To quantify piRNA counts associated with TEs, we used featureCounts v1.6.0 [138] with options -t exon -M. The 24–35 nucleotide long BAM file was used as input. The TEtranscripts-compatible TE annotations described above (see Transposable element annotations) were provided as the intersecting features. For Lake Malawi cichlids, featureCounts analysis was performed twice, using *A. calliptera* default and curated TE annotations, although the curated annotations were used in final figures (see mapping statistics in Additional File 5: Table S4). After obtaining the tables of counts, DESeq2 [109] and custom scripts (available online [89]) were used to calculate normalised

counts, generate plots and conduct statistical tests on an R framework [78], with packages tidyverse [80], lattice [118], eulerr [119], genefilter [120], pheatmap [121], reshape2 [122], ggrepel [123], biomaRt [124], tximport [125], RColorBrewer [126], ashr [127], ggpubr [128], GenomicFeatures [115], patchwork [129]. To create bigwig files, the 24–35 nucleotide long BAM alignment files were used as inputs to bamCoverage v3.5.1, part of the deepTools package [111], using options --normalizeUsing CPM -of bigwig --binSize 5. Bigwig files of biological replicates of same organ were combined using WiggleTools [112] mean and wigToBigWig v4 [113]. Genome tracks were plotted with custom scripts (available online [89]) using the Gviz [114] and GenomicFeatures [115] packages on an R framework [78]. We used these bigwig files to produce sRNA metagene profiles with deepTools [111] computeMatrix scale-regions v3.5.1 (options -b 1000 -a 1000 --regionBodyLength 2000 --averageTypeBins median --missingDataAsZero --binSize 5) and plotProfile v3.5.1 (--plotType se --averageType mean --perGroup). To generate metagene profiles against particular TE classes or superfamilies, TE annotations were subsetted by TE class or superfamily and converted to bed format with grep and awk utilities. The resulting bed files contained the regions to plot and were used as input for computeMatrix.

To define piRNA clusters, we first re-mapped trimmed reads 24–35 nucleotides long to the *A. calliptera* (fAstCal1.2) or *O. niloticus* (O_niloticus_UMD_NMBU) genomes using STAR v2.5.4b [116], with options: --readFilesCommand zcat --outFilterMultimapNmax 100 --outFilterMismatchNmax 2 --alignIntronMax 1 --outSAMtype BAM SortedByCoordinate --outFilterType BySJout --alignSoftClipAtReferenceEnds No --winAnchorMultimapNmax 100 --alignEndsType EndToEnd --scoreDelOpen −10,000 --scoreInsOpen −10,000 --outSAMmultNmax 100 --outSAMattributes All. We used the piRNA Cluster Builder (piCB) package to identify piRNA clusters. The method and code related to this package is detailed elsewhere [61]. In short, the BAM files were loaded into R environment using GenomicAlignments package [115]. For each BAM file the alignments were sorted into three categories: unique mapping alignments, primary multimapping alignments, and secondary multimapping alignments [61]. The reference genome was split into sliding windows [115] with size and step between starting position depending on the alignments category. For unique mapping alignments the windows were 350 nt (window size) starting at every 35 nt (window step) of genome length. For each of these windows the number of overlapping unique mapping alignments was counted. If the number was at least 2 FPKM (RPKM), the window was called. The called windows were reduced into genomic intervals named "seeds", indicating the genomic origin of uniquely mapping piRNAs. Seeds that were shorter than 800 nt were discarded to reduce false positives, which can be caused by individual degradation fragments of abundant structural RNAs or other cellular transcripts. Next, we incorporated multimapping piRNA reads, considering first their primary alignments and then all possible alignments (up to 100 according to the parameters used for genome mapping). We counted primary multimapping alignments using 350 nt long sliding windows (window size) located at every 35 nt (window step) of genome length. Windows overlapping with more than 4 FPKM (RPKM) with each other and with previously established 'seeds', were reduced into intervals named 'cores'. Each 'core' was required to overlap with at least one seed. Finally, we integrated all secondary multimapping alignments using 1000 nt long sliding windows

(window size) with 100 nt step (window step). We requested read coverage greater or equal 0.2 FPKM (RPKM) as threshold. Overlapping windows were reduced into 'clusters' when they overlapped with at least one 'core'. All clusters contain strand information and predict one or multiple piRNA precursor transcripts from a defined genomic strand. Intersection [115] of genomic 'cluster' coordinates from different samples or biological replicates take strand information into account. Of note, individual seeds can generate a core even without any multi-mapping reads, and the same applies to a core capable of generating a cluster. Thus, genomic loci covered exclusively by uniquely-mapping reads can be identified as clusters as well. Results were plotted on a R framework [78], using packages: tidyverse [80], reshape2 [122], and ggpubr [128]. A compilation of piRNA clusters identified can be found in Additional File 3: Table S2. Cluster intersections were calculated with the function PICBCombine() of the piCB package, which is in turn based on GenomicRanges::intersect(). Divergent sites were identified based on subsetByOverlaps(invert = T). First we intersected with PICBCombine the clusters identified in all the libraries of the same species and organ. Then, these were intersected in the different combinations reported in the figures to determine shared and divergent piRNA clusters. Circos plots were created with Circos v0.69–8 [139]. Density tracks are displayed on the circos plots as the number of features per mega-base.

We defined the observed overlap of piRNA clusters with genomic features and TEs as the proportion of bases of piRNA cluster sequences that intersect with such regions. We opted for the percentage overlap, because RepeatMasker TE annotations are not necessarily reliable and can result in over-fragmented annotations, which can lead to an overestimation of the number of piRNA cluster-TE intersections. 95% confidence intervals for the observed values were estimated by recomputing these overlaps for 10 bootstraps where 20% of piRNA ranges were randomly omitted. The observed distribution was then compared to the expected random null distribution where the piRNA cluster coordinates were randomly shuffled within ± 18,000 bp of their original positions, against which Z-test significance values were calculated.

### Protein preparations and mass spectrometry

Frozen brain and gonad tissues were partitioned on a mortar positioned on dry ice, and weighed. This was done quickly to avoid thawing. A similar mass of the same tissue was collected from size-matched individuals of the same species to create biological replicates. 6–50 mgs of brain tissue, and 8–120 mgs of gonad tissue were used, according to the specific tissue, tissue availability, and size of the specimen, which varied per species. Partitioned tissues were transferred to BeadBug tubes prefilled with 0.5 mm Zirconium beads (Merck, #Z763772) together with 150 µl (if using 6–20 mg of tissue) or 250 µl (if using > 20 mg of tissue) of modified RIPA buffer (50 mM Tris HCl pH 7.5, 150 mM NaCl, 1% IGEPAL CA-630, 1% Sodium Deoxycholate, supplemented with cOmplete EDTA-free protease inhibitor cocktail tablets, Roche #4693132001). Next, homogenisation was conducted using a BeadBug microtube homogeniser (Sigma, #Z764140) at approximately 4 °C (conducted in cold room). Each sample was homogenised with five BeadBug runs at maximum speed (4,000 rpm) for 60 s each. Did not run any sample more than two consecutive times to avoid overheating. Other than the run time inside the BeadBug, samples were left on ice. After homogenisation, lysates were centrifuged

for 5 min at 18,000 G at 4 °C. Supernatant was then removed into a clean 1.5 mL tube. Centrifuge the lysates again, this time at maximum speed (approximately 21000 G) for 5 min at 4 °C. Transfer supernatant into a clean tube without disturbing the pellet and tissue debris. Measured protein concentration using Bradford (Bio-Rad, Protein Assay Dye Reagent Concentrate, #5000006) and prepared a final sample by combining 150 µg of lysate, $1 \times$ LDS (prepared from NuPAGE LDS Sample Buffer 4x, Thermo Scientific, #NP0007) and 100 mM DTT and boiling for 10 min at 95 °C. Half of the sample was sent for mass spectrometry.

In-gel digestion for mass spectrometry was performed as previously described [140]. Samples were boiled at 70 °C for 10 min prior to loading on a 4%−12% NuPAGE Bis−Tris gel (Thermo Scientific, #NP0321). The gel was run in $1 \times$ MOPS buffer at 180 V for 10 min and subsequently fixed and stained with Coomassie G250 (Carl Roth). Each lane was minced and transferred to a 1.5 mL reaction tube, destained with 50% EtOH in 50 mM ammonium bicarbonate buffer (pH 8.0). Gel pieces were dehydrated with 100% acetonitrile and dried in a Concentrator Plus (Eppendorf, #5305000304). Then, samples were reduced with 10 mM DTT / 50 mM ABC buffer (pH 8.0) at 56 °C and alkylated with 50 mM iodoacetamide / 50 mM ABC buffer (pH 8.0) in the dark. After washing with ABC buffer (pH 8.0) and dehydration with acetonitrile the proteins were digested with 1 µg mass spectrometry-grade Trypsin (Serva) at 37 °C overnight. The peptides were purified on stage tips as previously described [141]. Peptides were analysed by nanoflow liquid chromatography using an EASYnLC 1200 system (Thermo Scientific) coupled to an Exploris 480 (Thermo Scientific). Peptides were separated on a C18-reversed phase column (60 cm, 75 µm diameter), packed in-house with Reprosil aq1.9 (Dr. Maisch GmbH), mounted on the electrospray ion source of the mass spectrometer. Peptides were eluted from the column with an optimized 103-min gradient from 2 to 40% of a mixture of 80% acetonitrile/0.1% formic acid at a flow rate of 250 nL/min. The Exploris was operated in positive ion mode with a data-dependent acquisition strategy of one mass spectrometry full scan (scan range 300–1650 m/z; 60,000 resolution; normalised AGC target 300%; max IT 28 ms) and up to 20 MS/MS scans (15,000 resolution; AGC target 100%, max IT 28 ms; isolation window 1.4 m/z) with peptide match preferred using HCD fragmentation. Mass spectrometry measurements were analysed with MaxQuant v1.6.10.43 [142] with the following protein databases (downloaded from Ensembl): Haplochromis_burtoni.AstBur1.0.pep.all.fa (35,619 entries, from *A. burtoni*), Oreochromis_niloticus.O_niloticus_UMD_NMBU.pep.all.fa (75,555 entries, from *O. niloticus*), Astatotilapia_calliptera.fAstCal1.2.pep.all.fa (41,597 entries, from *A. calliptera*), and Pundamilia_nyererei.PunNye1.0.pep.all.fa (32,153 entries, from *P. nyererei*). Missing values were imputed at the lower end of LFQ values using random values from a beta distribution fitted at 0.2–2.5%. Prior to further analysis, protein groups with contaminants, reverse hits and only identified by site were removed.

## Supplementary Information

Additional file 1. Supplementary figures: Figs S1-S11.

Additional file 2. Table S1. Results of orthology analysis, PiggyBac-1 TIR BLASTN results, selection test results, and metadata on all the genomic resources used to ascertain presence/absence of *piwil1* genes in Lake Malawi cichlid species.

Additional file 3. Table S2. Results of piRNA Cluster Builder (piCB) benchmarking and analysis.

Additional file 4. Table S3. Polymorphism and expression of TE families.

Additional file 5. Table S4. Mapping statistics of small RNA- and mRNA-sequencing data.

Additional file 6. Peer review history.

## Acknowledgements

We are grateful to all members of the Miska, Durbin, Santos, and Haase groups for discussion and suggestions. We are especially grateful to Chenxi Zhou for input on confirmation of *piwil1* duplications, and Navin B. Ramakrishna for analytical input and critically reading the manuscript. We thank all who provided fish or fish samples: Kevs Rifts for *M. zebra* fishes; Hans Hofmann and Caitlin Friesen (University of Texas at Austin, TX, USA) for *A. burtoni* samples; David Penman, Alastair McPhee, and James F. Turnbull (Institute of Aquaculture, University of Stirling, Scotland, UK) for *O. niloticus* samples. We are thankful to the animal caretakers of the cichlid facility of the University of Cambridge for diligently ensuring the maintenance of our *A. calliptera* and *T.* sp. 'mauve' stocks. The Malawi cichlid samples were collected ethically under prescribed permits, and the results and data are published under an Access and Benefit Sharing agreement with the Government of Malawi, administered by the Department of Fisheries. We acknowledge the contributions of the Malawi Department of Fisheries and the people and Government of Malawi for their assistance in the collection of samples and the generation of data and results. We thank Maike Paramor and Vicki Murray (Wellcome-MRC Cambridge Stem Cell Institute) for NGS library preparation.

### Review history
The review history is available as Additional file 6.

### Peer review information
PD Dr. Jürgen Schmitz and Tim Sands were the primary editors of this article and managed its editorial process and peer review in collaboration with the rest of the editorial team.

### Authors' contributions
Conceptualisation: M.V.A. and E.A.M.; Data curation: M.V.A., M.B., and J.L.P.; Formal analysis: M.V.A., M.B., C.U.Y., P.S., J.L.P., and F.X.Q.; Funding acquisition: M.V.A., A.D.H., R.D. and E.A.M.; Investigation: M.V.A., and C.U.Y.; Project administration: M.V.A. and E.A.M.; Resources: P.S., A.F., A.D., G.V., A.L.K.P., A.M.S., D.A.J., F.B., A.D.H., R.D., and M.E.S.; Software: A.F. and A.D.H.; Supervision: M.V.A., A.D.H., R.D., M.E.S., and E.A.M; Visualisation: M.V.A., M.B., C.U.Y., and J.L.P.; Writing – original draft: M.V.A.; Writing – review & editing: all authors contributed.

### Funding
M.V.A. is funded by the European Union's Horizon 2020 research and innovation programme under the Marie Skłodowska-Curie grant agreement No. 101027241. M.B. is funded by a Harding Distinguished Postgraduate Scholarship of the University of Cambridge. C.U.Y., F.X.Q., and A.L.K.P. were funded by the Cambridge Commonwealth, European and International Trust. P.S. is funded by the European Union's Horizon 2020 research and innovation programme under Marie Skłodowska-Curie grant agreement No. 956229. F.X.Q. is supported by a Wellcome studentship (108864/B/15/Z). A.D. is supported by the Royal Botanic Gardens Kew. G.V. acknowledges funding from Wolfson College—University of Cambridge and the Genetics Society London. A.D.H.'s research group is supported by the intramural research program of the National Institute of Diabetes and Digestive and Kidney Diseases (ZIA DK075111-07). R.D. is supported by a Wellcome Investigator Award (207492/Z/17/Z). M.E.S is supported by NERC IRF NE/R01504X/1. This work was supported by the following grants to E.A.M.: Wellcome Trust Senior Investigator Award (219475/Z/19/Z) and CRUK award (C13474/A27826). The authors also acknowledge core funding to the Gurdon Institute from Wellcome (092096/Z/10/Z, 203144/Z/16/Z) and CRUK (C6946/A24843).

### Data availability
The mass spectrometry proteomics data have been deposited to the ProteomeXchange Consortium via the PRIDE [143] partner repository with the dataset identifier PXD047439. The mRNA and sRNA sequencing data generated in this study have been deposited to GEO under accession numbers GSE252804 [144] and GSE252805 [145]. The mRNA sequencing datasets of early developmental stages of Lake Malawi cichlids have been deposited to SRA under BioProject accession PRJNA1155688 [146]. Additional data and code related to this work are available on GitHub and Zenodo [89], under a LGPL-3.0 license. The genomic data of Lake Malawi cichlids used in this work are available on SRA (BioProjects PRJEB1254, PRJEB24325, PRJEB72478, PRJNA1144838, PRJNA60369, PRJEB75300, PRJNA344471, PRJNA609616, PRJNA1144831, PRJEB72381, PRJNA1144843, PRJEB75298, and PRJNA1144851, see a list of samples in Additional File 2: Table S1), on an open access basis for research use only. Any person who wishes to use this data for any form of commercial purpose must first enter into a commercial licensing and benefit sharing arrangement with the Government of Malawi, administered by the Department of Fisheries.

## Declarations

### Ethics approval and consent to participate
Fish feeding, housing, and handling were conducted in strict adherence to local regulations and with the protocols listed in Home Office project license PP9587325.

**Consent for publication**
Not applicable.

**Competing interests**
The authors declare that they have no competing interests.

**Author details**
[1]Department of Biochemistry, University of Cambridge, Tennis Court Road, Cambridge CB2 1GA, UK. [2]Wellcome/CRUK Gurdon Institute, University of Cambridge, Tennis Court Road, Cambridge CB2 1QN, UK. [3]Department of Genetics, University of Cambridge, Downing Street, Cambridge CB2 3EH, UK. [4]National Institute of Diabetes and Digestive and Kidney Diseases, National Institutes of Health, Bethesda, MD 20892, USA. [5]Biophysics Graduate Program, Institute for Physical Science and Technology, University of Maryland, College Park, Maryland 20742, USA. [6]Comparative Fungal Biology, Jodrell Laboratory, Royal Botanic Gardens Kew, Richmond TW9 3DS, UK. [7]Present Address: Zoological Institute, Department of Environmental Sciences, University of Basel, Vesalgasse 1, Basel 4051, Switzerland. [8]School of Natural Sciences, University of Hull, Hull HU6 7RX, UK. [9]Institute of Molecular Biology (IMB), Quantitative Proteomics, Ackermannweg 4, Mainz 55128, Germany. [10]Institute of Molecular Virology and Cell Biology, Friedrich-Loeffler-Institute, Südufer, Greifswald 17493, Germany. [11]Wellcome Sanger Institute, Tree of Life, Wellcome Genome Campus, Hinxton CB10 1SA, UK. [12]Department of Zoology, University of Cambridge, Downing Street, Cambridge CB2 3EJ, UK.

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

## 