## [Additional file 6. Peer review history. · Genome Biology]

Review history

First round of review

Reviewer 1

In this manuscript, the authors aim to address whether transposable elements (TEs) might be involved in the extreme diversification of cichlid fishes in the East African Great Lakes. They choose six cichlid species and analyze TE expression as well as the piRNA pathway (one of the TE silencing pathways) in the gonads and early development. This manuscript represents an interesting resource that should be useful to the community. I have one major concern and several questions, as follows.

Major concerns.

1. My main concern is that the most important conclusion of the manuscript is a coevolution of TEs and the piRNA pathway. However, this conclusion is not strongly supported by the data.

1.1. The last paragraph of data (p. 14, lines 464-486) concludes that "piRNAs are likely to be engaged in ongoing silencing of transpositionally active TE families". This conclusion is based on the inverse correlation between expression of active TE families and levels of the corresponding piRNAs. These data are not convincing. First, the authors do not describe how active TE families were identified (line 465). Second, this inverse correlation is shown for only two TE families, it does not stand for the other families (Fig. 6E, F).

1.2. One main argument for the coevolution of TEs and the piRNA pathway is the expansion of piwil1 genes. This part of the manuscript is misleading because it gives the impression of the presence of several piwil1 genes able to regulate TE expression. In fact of the three additional piwil1, one is a pseudogene, and of the other two, only one (piwil1.4) is expressed in gonads. But most importantly, both piwil1.3 and piwil1.4 are truncated forms of piwil1 and encode only the PIWI domain. As such, these proteins cannot bind piRNAs (this is explained in the Discussion) and thus are unlikely to regulate TEs. These proteins are expected to have new functions, but there is no evidence for functions in TE repression. They cannot target TE mRNAs due to their lack of piRNA loading and they have lost the domains involved in interaction with other piRNA factors. Therefore, increased number of piwil1 genes cannot be used as an argument for the coevolution between TEs and the piRNA pathway.

2. Fig. 4 shows that piRNAs are not produced by phasing in ovaries (except in the AC species). Two proteins, the endonuclease Pld6 and the RNA helicase Mov10L1 are specific to phasing in other species. It would be interesting to correlate the presence of phasing with the presence of these two proteins in the corresponding tissue.

Regarding Mov10L1, two proteins appear in the list in Fig. S2B, Mov10a and Mov10b.1. The authors should clarify the relationships between these two proteins and Mov10L1, as in other species Mov10 proteins (including Mov10a and Mov10b.1) have more general expression pattern and functions, while Mov10L1 is specific to piRNA biogenesis.

The presence of Pld6 in testes or ovaries does not correlate with the presence phasing signatures. This is unexpected. Could the authors comment on this?

3. The size differences in piRNAs between ovaries and testes might be explained by their loading in different PIWI proteins. However, Piwil1 and Piwil2 are not specifically expressed in either testes or ovaries (Fig. S2B). Could the piRNA size difference reflect the expression levels of Piwil1 and Piwil2 in ovaries and testes? Or what could be the basis of this size difference?

4. piRNAs have various functions in addition to repressing TEs. Since small RNAs of 24-35 nt show piRNA signatures, it would be interesting to indicate the percentage of them (potential piRNAs) corresponding to the TE sequences vs other sequences.

Reviewer 2

Almeida et al, 2024 present first in-depth study into the co-evolution of piRNAs and TEs in African cichlids. This study uses variety of data from transcriptomics, proteomics and evolutionary genomics to show piRNA and TE co-evolution is rapid and dynamic across African cichlids from three lakes. They also present many additional exciting observations regarding *piwil1* genes evolution and sex-bias in TE and piRNA cluster activity. Overall, it's potentially a landmark study for cichlid biology and opens a lot of fundamental questions regarding piRNA pathway evolution in vertebrates, especially during speciation events. There are several major concerns, however, which need to be addressed.

1. Analysis cut-offs are very relaxed and insufficient details of QC results:
mRNA expression analysis has very relaxed parameters. In addition, piRNA cluster annotation very stringent parameters for length of the cluster whereas relaxed parameter for alignments considered. The latter is probably why median cluster lengths are less than 5kb in some species/gonads. It is also unclear if the low-mappability regions of the clusters were considered for merging with flanking uniquely-mapping piRNA 'seed' windows.
2. Figures 1 and 6 are extremely information dense but most panels are only superficially discussed or mentioned. A reader who is not expert in TE or piRNAs reading the manuscript will have difficulty understanding the reasoning behind the analysis.
3. Authors also discuss and/or compare their findings regarding enrichment of TE expression in early embryo and *piwi* gene structure to that of model organism, zebrafish and sometimes widely to all animal models. This is a strength of the study and will make the paper broadly accessible to the community. However, in some places where authors try to connect diverse and dynamic TE expressions to cichlid species diversity, such comparisons are essentially missing from other main figure panels where comparative analysis between AC and ON would be insightful (such as Fig. 6H, Fig 4, Fig. 2E). Overall, if the authors mean to link cichlid diversity to divergence in genomic TE abundance, age or expression, this needs to be more directly discussed.
4. The conclusion of the paper is conflicting. One hand authors show that >90% of TE families are expressed in gonads and another hand that piRNA-mediated silencing of TE mRNA is active. This could be addressed by showing correlations between TE mRNA expression and piRNA abundance/ping-pong signal. Are the highly expressed TEs not regulated by piRNAs?

Specific Major Concerns:

Page 4, Line 56: Given the age distribution of LTR TEs between AB, PN and ON, AC genomes, its expected and reassuring that in Fig S1B, fewer LTR families are expressed in AB and PN because LTRs composition of their genome is smaller and older (10-20% divergence) compared to LTRs of ON and AC. However, the number of LINE families expressed in all 4 species are quite similar despite the abundance of young (<5% divergence) in ON and AC genomes. Have the authors investigated this further for any possible explanation?

Page 6, Line 3: Authors state that African Cichlids have substantial transcriptional TE activity. This can be misleading to non-TE researchers. First, a quantification of proportion of TE-derived mRNA across gonads and developmental stages is needed, which will help orient readers the contribution of TEs to the transcriptome in Lake Malawi cichlids compared to ON cichlid and zebrafish etc. Second, since authors have already quantified Locus-level TE expression, it shouldn't be too much work to distinguish how much of TE transcriptome is TE-derived and not from read-through transcription of nearby genes. These two analyses can help authors convince readers that African Cichlids are exceptional in their TE expression or abundance of TEs in their transcriptome which may contribute to their phenotypic diversity.

Page 23, Line 38: When mapping to TE family consensus, too many multi-mappers were allowed. TE family, by definition, are unique sequences for each designated family and thus If substantial multi-mapping reads were detected, it's very likely that some consensus sequences are not well resolved and thus expression of some families is overestimated. I suggest authors try stricter parameters such as `-outFilterMultimapNmax 3` and compare how much reads are unmapped compared to their original parameters. It seems the authors did reduce some of the overestimation by curating the AC consensus TE library but more details of the mapping statistics either in a supplemental table or figure is also needed.

Page 6, Line 55: Authors found similar PiggyBac insertions in 3' UTR of *piwil1* genes in AC is a very intriguing finding. It seems the authors are implying that *piwil1* could have been duplicated by PiggyBac transposition events. Putative TIR sequences shown at the borders of *piwil1* gene copies are convincing, but authors do not show the length of complete TIRs from the corresponding family consensus. Is the 9nt sequence shown in the white boxes the complete TIR sequence or only partial? Some PiggyBac TEs are well known to also have asymmetric TIR sequences.

Figure 2C: It is not clear where the LT and LV species belong in this plot. Are they even in this phylogeny? Annotation of species AC, MZ etc along the phylogeny will be helpful. Did the authors try to trace if the expansions of *piwil1* genes happened before the adaptive radiation in Lake Malawi or afterwards?

Page 8, Line 32: I also recommend authors consider alternative explanations, such as *PiggyBac* TEs maybe targeting *piwil1* genes. It seems the *piwil1.1* also has a more recent *PiggyBac* insertion in the S-region. Such targeting of another piRNA pathway genes has been reported for flies (PMID: 33347429). Since *PiggyBac* insertions are also apparently enriched near *piwil1* genes than the expected genome average, this might support the alternative hypothesis. I also suggest authors investigate if 'TTAA' (preferred site) tetranucleotides are more common in the *piwil1* genes neighborhood than other flanking genes? Overall, I think the conclusions made are very exciting and provocative, but the evidence presented for *PiggyBac* mediated expansion of *piwil1* is at best weak.

Figure 5 A&C: These circos plots are very information dense and can be very helpful in comparing the genomic distribution of clusters, association with TEs and conservation. However, some of the tracks do not add much more value to the plot. For example, piRNA coverage on both strands because of log2 transformation in such large window sizes is not informative. It is reassuring to see the apparent positive correlation between piRNA clusters and piRNA coverage, but this can be easily demonstrated by a dot plot such as shown in Figure S5. Plus, the gray background under the coverage track is very distracting. Additionally, the legend for tracks 5-6 is confusing. Does this indicate that the clusters were reproducibly detected across species or across tissues of the same species? Some explanation of what the Y-axis values here mean will be helpful.

Overall, I suggest the authors simplify these plots to highlight their salient points, which is rapid divergence of piRNA cluster activity across the species and strong sex-biased activity of many clusters. In addition, the TE distribution track is very small and thus difficult to correlate to piRNA cluster peaks. TE or any genomic features in the outermost track would help readers visualize the correlation much better.

Figures 5D and supplementary data: MZ testis cluster counts and ON ovary cluster counts are highly variable across replicates (>40% of ON ovary clusters in one replicate is not found in other replicates?). This indicates low quality libraries in one or more replications. Did the authors investigate this? Additionally, figure legend, results and even methods do not indicate any parameters of how the cluster overlap and sharing was evaluation between sexes and species. Was there a liftOver performed?

Page 26 and Figure 5A, C: Was there a separate annotation performed for uniquely mapping reads as well? This is necessary to show which clusters are truly high confidence and perhaps this will also improve the reproducibility issues earlier due to using multi-mapping reads.

Minor Concerns:

Page 5, Line 50: When describing the mRNA-seq results for TEs in gonads and embryos, authors highlight that >90% of TE families are expressed. This is very striking indeed (because it is from wildtype fish) but could be attributed to the low cutoff used for classifying TE expression (also see below in minor concerns). If this low cutoff was intentional (perhaps to account for piRNA-mediated slicing of TE mRNA in these Wildtype fish?), authors should state that in the result.

Figure 2A: Also, in the entire *piwil1* gene shown, are these 9nt sequences in either direction only present in the places shown with white boxes? Or are they more common? All the sites of the 5 or 9nt sequence should be shown if there are more.

Figure 2E: Grey data points need to be smaller in size to reduce the overlap between adjacent points. Additionally, zoomed in plot of *piwil1.2* is missing.

Figure S1A. TE divergence plot of subclasses for TM genome is missing.

Page 6, Line 12: It is unclear where this list of genes was curated from and reasoning behind including these specific genes. piRNA pathway genes are rapidly evolving and sometimes species-specific. Does the list used constitute the core-machinery expected to be widely conserved?

Page 6, Line 35: It is unclear which fishes are being referred to here? Did the authors conduct a wide search across teleost genomes for *piwil1* gene copies?

Figure 5G: How do the authors know if the testis specific cluster shown is Lake Malawai specific? And for that matter is indeed testis-specific in TM and MZ? I do not see the tracks for ON for the same cluster. I also cannot find any mention of TM and MZ ovary clusters.

Figure 5E-F: While TE annotation track is shown, no information regarding the TE class or subclass is shown. This will be insightful in understanding what kind of TEs are present and will serve as an example of enrichment analysis present in 6A-B.

Page 6, Line 60: "377 high quality copies" is vague. Did the authors mean near-complete/autonomous or just that their percent divergence is low, and insertions are highly similar to the consensus?

Figure 6: This figure contains many different observations, and some are unexpected. However, most of the figure panels are quickly skimmed in the results and not described or even discussed later. For example, while all TEs in general and LTR TEs are always enriched in clusters, LINE TEs are significantly depleted in AC and ON clusters of both gonads. Also, the bars for SINEs, RC and 3' UTR are difficult to read. I suggest authors present the y-axis values in log₁₀ increments.

Page 8, Line 58: If piwil1 is indeed heterozygous in some fish, there should be an empty site in the homologous chromosome with no piwil1 copy. Authors can eliminate or confirm this explanation by looking for this site using piwil1 5' and 3' junctions from either the long or short reads.

Page 24, Line 24: There are no stats for the mapping to genome and TE consensus provided for the small RNA libraries. A supplemental file with such details is needed to understand the quality of the libraries.

Page 23, Line 56: Authors state that 10 counts or more for each TE family was cutoff was TE family to be considered expressed. If I understand this correctly, this is raw read counts, which, depending on the depth of the library and multi-mappers, is a very relaxed cutoff. In addition, it would help convince readers of true TE expression if authors could add stranded normalized coverage plots for some of these TEs consensus sequences.

Figurer 3A: Are the 4 points in each tissue replicates libraries? This is not clear in the legend. Same issue with Supp Fig S1G heatmap. Are the four columns replicates? These should be made clear.

Page 26, Line 1: Is there a specific reason why STAR was used for aligning small RNA reads? STAR is better suited for paired-end and longer mRNA-seq data. Was the seed length adjusted to be half or lower than half of the mean piRNA length? If STAR used longer seed lengths (>20nt), then many potential alignments may be missed.

Authors' response to reviewers

We are grateful to the handling editor and the reviewers for their critical reading and comments on our manuscript. The original comments (in black font) and the author responses (in green font) can be found below.

Editor

"Following my own perusal of your engaging manuscript, I am curious about the methodologies employed to differentiate genuine TE transcripts from those resulting from the co-transcription of TEs embedded within UTRs of genes, which may differ across various tissues or developmental stages. A similar concern was raised by Reviewer 2 who noted, "[...] TE-derived and not from read-through transcription of nearby genes."

This is an excellent point, and the revised manuscript now includes a supplemental panel on this.

We refrained from including such data in the initial submission as the available gene annotations for cichlid genomes could be improved and are not on par with gene annotations currently available for model organisms such as zebrafish. In fact, some of the authors are involved in ongoing efforts to produce higher quality reference genomes and gene annotations for East African cichlids. In any case, as this point was brought up by the handling editor and one reviewer, we now include this information in **Figure S2E** and mention it adequately in the text. This was most easily addressed in our embryogenesis datasets, where we had readily available analysis of TE expression at the single locus level. Due to the annotation issues mentioned above, we reasoned the “cleaner” way to do this analysis was to categorise mapped reads according to overlaps with annotated gene features. We defined overlaps with gene features identically to Chang et al., 2022 (PMID: 34987056) and assessed the proportion of counts in a library mapping to each feature. As can be observed in **Figure S2E**, most of the counts map to intergenic regions (between 38.5-49.3%). Between 0.2-0.6% of TE counts are embedded within 5' UTRs, and between 5.9-10.8% within 3' UTRs. The 12.2-17.1% proportion of counts overlapping with exons is likely due to transposons misannotated as protein-coding genes. We are currently curating the gene annotation to comprehensively remove such misannotations, but the curated version will not be available within the timeline of this revision. This analysis should be revisited once higher quality annotations are available for cichlids. We split intron mapping counts in two classes, as defined in Chang et al., 2022 (PMID: 34987056) and observe that 3.8-19.3% of counts map to TEs within introns of non-expressed genes, while 18.8-24.3% of counts map to TEs within introns of expressed genes. For the latter, TE transcription can be either independent or dependent on the expression of the gene.

In summary, these results do not affect our conclusions, with the majority of counts mapping to TEs within intergenic regions. This piece of information is now added to the manuscript.

“Additionally, it would be beneficial to discuss the proportion of TEs that might evade detection due to poly-A enrichment techniques”

That is also a good point, and one that we addressed in the very beginning of the project. We include a Figure in this response addressing this.

While we did not conduct a dedicated experiment to address this, we compared available cichlid RNA-sequencing data from Poly-A-enriched and rRNA-depleted libraries that were previously sequenced in our laboratory. We analysed this data with TEdetect, one of the most widely used tools in the field to quantify TE expression at the family level (PMID: 26206304, 29508296, 32576954). This is the same analytical tool that we used in our manuscript to assess TE expression at the family level. Only a minority of cichlid TE families is differentially expressed between the two treatments, showing that the two library

preparations are well correlated and are essentially equivalent in terms of determining TE expression at the family level (**Revision Figure 1**).

To summarise, we are confident we are not losing a significant amount of information by performing Poly-A enrichment in cichlid samples.

Revision Figure 1. Comparison of expression of TE families in RNA-sequencing data from Poly-A-enriched and rRNA-depleted libraries. (A) Scatter plot showing Counts per Million (CPM) in rRNA-depleted (y-axis) and Poly-A-enriched libraries (x-axis), shown as $\log_{10}(\text{CPM} + 1)$. This analysis incorporates data from all ovaries, testes, and brain samples. We highlighted two TE families, which have significantly higher expression in rRNA-depleted libraries (according to DESeq2 analysis). (B-C) Identical to (A) but displaying only CPM in testes (B) or ovaries (C), and split into different panels according to TE class. As in (A), it can be observed that TE family expression is well correlated

between these different library preparations.

“Although your primary focus has been examining the influence of piRNAs on TE expression, mentioning alternative silencing mechanisms would complement your argumentation.”

We mention in passing in the introduction that several pathways, not only the piRNA pathway, have evolved to silence TEs (see page 3, lines 141-143). To further elaborate on this, we now added a paragraph in the discussion (see page 14, line 682), mentioning other TE silencing pathways in more detail and urging future studies to address whether: 1) these pathways are active in cichlids; and 2) how these pathways interact/collaborate with the piRNA pathway in TE silencing.

“For identifying genomic regions subjected to positive selection, you utilized MEME; comparing these outcomes with a more stringent PAML analysis could provide robust evidence supporting such selection.”

We conducted the suggested analysis, and its results are included in the revised manuscript.

We would like to note that the strongest evidence supporting positive selection in Lake Malawi *piwil1* genes is not the MEME analysis, but the data shown in **Figure 2E** and (**Supplemental Table 1**), which was obtained by conducting Raised Accuracy in Sweep Detection (RAiSD) analysis (PMID: 30271960) using single-nucleotide polymorphism data from 79 Lake Malawi cichlid genomes (see methods). This integrative analysis shows that the genome regions encompassing *piwil1.3* and *piwil1.4* have strong evidence of selective sweeps (amongst the sweep signatures associated with all genes, these are within the 99th and 93rd percentile, respectively).

Besides this well-powered genomic approach, we tested for signatures of selection on the coding sequences of cichlid Piwil1 proteins. Of note, we use only 10 cichlid Piwil1 sequences, all the available annotated protein sequences in the cichlid genomes in Ensembl. For the selection tests, we used two analytical tools of the HyPhy package, available via the Datamonkey 2.0 webserver (PMID: 29301006), as detailed in the methods:

“A gene wide test was first performed using Branch-site Unrestricted Statistical Test for Episodic Diversification (BUSTED). We conducted the test in two ways, testing for selection across all branches and testing for selection only in radiating cichlids, with *O. niloticus* as an outgroup. BUSTED reported strong support for positive selection in both cases (p-value = 0). Figure S5A shows the subsequent analysis to identify residues likely to be under positive selection according to Mixed Effects Model of Evolution (MEME).”

After the editor’s comment we repeated the analysis using codeml of the PAML package v4.10.7. Results are now included in **Supplemental Table 1**. To detect positive selection, we used branch-site tests with codeml in two ways: 1) with the Piwil1 proteins of radiating cichlids in the foreground; and 2) with the branches including Piwil1.3 and Piwil1.4 (truncated duplications in Lake Malawi cichlids) in the foreground (because we do see stronger signatures of selection on these truncations in our genome-wide RAiSD analysis mentioned above). Both these approaches, much like the BUSTED approach we used initially, support positive selection in foreground branches versus background branches (Supplemental Figure 1, input alignment and control files can be found in our GitHub: https://github.com/migueldvalmeida/Cichlid_TEs_piRNAs2024). Of note, 7/13 amino acid residues identified as likely under positive selection by MEME were also identified by PAML’s analysis (**Supplemental Table 1**).

Overall, these results obtained with the PAML package are consistent with the Datamonkey/HyPhy framework we used initially. Our conclusion of positive selection on cichlid Piwil1 proteins is now more robustly supported. We adjusted the methods to describe the additional PAML analysis.

“Furthermore, please ensure accuracy when defining SINEs (Short INterspersed Elements) throughout the text.”

We replaced in page 5, line 227 “SINE (short interspersed nuclear element)” with “SINE (short interspersed element)” as requested.

“PS: Please note that Genome Biology requires all cited data to be made publicly available, either through publication or via a data deposit platform that assigns DOIs (such as FigShare). As an alternative, such data may be included as part of the supplementary material of your manuscript.”

Our revised manuscript fully complies with Genome Biology’s data availability policies. In our initial submission we relied on three instances of “unpublished results”, this was amended in our revised manuscript. We apologise for this, there was simply a mismatch in timing of the readiness of our work and that of other unpublished work we relied on.

-First, we relied on unpublished mRNA-sequencing datasets of early developmental stages of Lake Malawi cichlids. Now the data and metadata are available at SRA under accession number PRJNA1155688. Reviewer link here: <https://dataview.ncbi.nlm.nih.gov/object/PRJNA1155688?reviewer=o5tti1kfle6372vr0htdpt9jii>. Datasets will become publicly available upon publication.

-Second, we relied on unpublished methodology and code used for the identification of piRNA clusters. This refers to a novel tool, piRNA Cluster Builder (piCB) by Astrid Haase’s laboratory. In the meantime, a manuscript describing and benchmarking the method was published as a preprint and has now undergone peer-review (in press at Cell Reports). We adjusted the methods and cite the preprint.

-Third, the identification of TE families likely to be transpositionally active in Lake Malawi were unpublished results. As the work describing this in detail was not published in the meantime, we have now included the methodology used, a supplemental table with relevant data, and additional expression data on additional families.

Reviewer #1

In this manuscript, the authors aim to address whether transposable elements (TEs) might be involved in the extreme diversification of cichlid fishes in the East African Great Lakes. They choose six cichlid species and analyze TE expression as well as the piRNA pathway (one of the TE silencing pathways) in the gonads and early development. This manuscript represents an interesting resource that should be useful to the community. I have one major concern and several questions, as follows.

We thank the reviewer for critically reading the manuscript and for the fair comments. Please find below our responses to the reviewer's comments (in green font).

Major concerns.

1. My main concern is that the most important conclusion of the manuscript is a coevolution of TEs and the piRNA pathway. However, this conclusion is not strongly supported by the data.

We do concede that the evidence supporting co-evolution between the piRNA pathway and TEs is indirect, as in other studies in the field. However, we provide a body of evidence suggesting co-evolution:

- Dynamic TE expression particularly in embryogenesis and early development (including TE families likely to be transpositionally active).
- Piwi genes are evolving fast and under positive selection.
- Divergence in piRNA-producing loci in closely related cichlid species.
- piRNA pathway targeting of TEs (including TE families likely to be transpositionally active).

Positive selection of piRNA factors, in particular, has been interpreted as co-evolution, characteristic of an arms race between piRNA pathway and TEs (e.g. PMIDs: 16103923, 18926973, 20971974, 22997235, 23550757, 24846630, 26868596, 28018141, 30958115)

In addition, we mention in the introduction and discussion another important piece of evidence from other work in line with co-evolution. We find evidence supporting recent TE transpositional activity in Lake Malawi cichlids (in an evolutionary timescale). This is demonstrated by a pangenomic approach, which revealed extensive TE-derived structural variation in Lake Malawi cichlids (<https://www.biorxiv.org/content/10.1101/2024.03.28.587230v1>, currently in revision), and by additional unpublished work that is in preparation (by P. Sierra & Richard Durbin, co-authors in our manuscript). The latter unpublished work characterises in detail TE variation in Lake Malawi using extensive genomic data. As this unpublished work informed our own study by giving us likely transpositionally active TE families, we now include additional information in the methods informing how this was determined and include additional data in **Supplemental Table 3** and **Figures 6** and **S11**.

In conclusion, we argue that when all the evidence is considered, it does altogether point towards a scenario of co-evolution. We note that in the abstract we refer to this with caution: “Our findings **suggest** dynamic co-evolution of TEs and host silencing pathways in the African cichlid radiations.” In our replies to 1.1 and 1.2 below, we further elaborate on how our data supports co-evolution between TEs and the piRNA pathway.

1.1. The last paragraph of data (p. 14, lines 464-486) concludes that “piRNAs are likely to be engaged in ongoing silencing of transpositionally active TE families”. This conclusion is based on the inverse correlation between expression of active TE families and levels of the corresponding piRNAs. These data are not convincing. First, the authors do not describe how active TE families were identified (line 465). Second, this inverse correlation is shown for only two TE families, it does not stand for the other families (Fig. 6E, F).

We apologise for the lack of clarity here. The revised manuscript includes textual changes and additional data to address this.

The description of how the transpositional activity of particular TE families was determined is unpublished work nearing final preparation stages. As we could not publish these results during the period of review of this work, we now describe in our work the methodology employed (see page 20, methods section “Inference of transpositional activity of transposable elements from genomic data”), contextualise the results in more detail (see pages 11-12, lines 527-561), and, importantly, include **Supplemental Table 3** with relevant data and additional data on **Figures 6** and **S11**.

To identify TE families active in AC, we used publicly available AC genomic data sampled from two small Lakes around Lake Malawi, Lakes Masoko and Kingiri, and quantified intrapopulation TE diversity π_{TE} and interpopulations TE diversity D_{xyTE} , based in the allele frequencies of the polymorphisms (see **Methods** section “Inference of likely transpositional activity of transposable elements from genomic data” for details). Then, the top 10 TE families with at least 100 identified polymorphisms, and highest values of the $\frac{\pi_{TE}}{D_{xyTE}}$ ratio in either Lake Masoko or Lake Kingiri populations of AC were selected, because in our interpretation these are strong signs of transpositional activity in recent evolutionary time. This led to 15 TE families (**Supplemental Table 3**) with strong support for recent transpositional activity. We note that this is a fairly stringent and conservative cutoff, meaning the number of families with recent signs of transpositional activity may be underestimated. This is likely given the abundance of TE-derived structural variation in Lake Malawi cichlids (<https://www.biorxiv.org/content/10.1101/2024.03.28.587230v1>).

Furthermore, we have now included additional data in **Figures 6** and **S11** showing the expression and mapped piRNAs of additional TE superfamilies, which include TE families likely to be transpositionally active. As we indicated in the main text, we focused on the TE superfamilies which displayed statistically significant differences in expression between active versus inactive TE families in at least one organ. The other families did not produce statistically

significant results, although in some cases the trends persist (e.g. active hAT-Ac in testes seems to be more highly expressed than inactive families, and the opposite trend is true for mapped piRNAs). For the other potentially active families in which we do not observe the inverse correlation, this may be due to piRNA silencing not being fully established yet. We adapted the text accordingly to describe these results more comprehensively as these are also interesting results.

In summary, to address this comment we added more detail to the methods and main text regarding how transpositional activity was assessed, added a supplemental table with this information along with the data on expression levels and mapped piRNAs of likely transpositionally active vs inactive elements.

1.2. One main argument for the coevolution of TEs and the piRNA pathway is the expansion of *piwil1* genes. This part of the manuscript is misleading because it gives the impression of the presence of several *piwil1* genes able to regulate TE expression. In fact of the three additional *piwil1*, one is a pseudogene, and of the other two, only one (*piwil1.4*) is expressed in gonads. But most importantly, both *piwil1.3* and *piwil1.4* are truncated forms of *piwil1* and encode only the PIWI domain. As such, these proteins cannot bind piRNAs (this is explained in the Discussion) and thus are unlikely to regulate TEs. These proteins are expected to have new functions, but there is no evidence for functions in TE repression. They cannot target TE mRNAs due to their lack of piRNA loading and they have lost the domains involved in interaction with other piRNA factors. Therefore, increased number of *piwil1* genes cannot be used as an argument for the coevolution between TEs and the piRNA pathway.

There are several points to unpack here:

1. “One main argument for the coevolution of TEs and the piRNA pathway is the expansion of *piwil1* genes.” We agree that our phrasing may be misleading. *piwil1* loci are evolving fast (and under positive selection) and this may be suggestive of coevolution, not the *piwil1* expansion per se. We now altered the text in our manuscript accordingly and believe the message is clearer. As mentioned above, this is an interpretation that is commonly found in the literature, particularly when positive selection is detected (e.g. PMIDs: 16103923, 18926973, 20971974, 22997235, 23550757, 24846630, 26868596, 28018141, 30958115).

2. “But most importantly, both *piwil1.3* and *piwil1.4* are truncated forms of *piwil1* and encode only the PIWI domain, so won’t bind piRNAs and are unlikely to regulate TEs.” We agree that right now we have no functional links between the truncated Piwil1 proteins and TE repression or expression. Now that we understand the polymorphic nature of these loci, we are ready to embark on functional studies. We believe these are beyond the scope of this manuscript. A first step we are planning in the near future is to create mutants of *piwil1.4*, which seems to be nearly fixed in Lake Malawi, and address the phenotypic and molecular impact of this perturbation. In the meantime, we would like to note that the truncated Piwil1 proteins may still have functions related to TE silencing or expression. A provocative hypothesis that we had

initially mentioned in the discussion, but removed in the meantime as it was too speculative is that these extra PIWI domains, which are not expected to bind piRNAs directly, may be weakening the piRNA pathway via a dominant negative-like effect. This could involve a hypothetical scenario where TEs hijacked the piRNA pathway (please note that piggyBac TEs are likely to be at least partially involved in the *piwil1* expansion) making it less effective and therefore benefiting TE activity. Mechanistically, this could be achieved for example if the truncated PIWI domains are able to affect piRNA strand-bias (PMID: 24757166) or interact with and “sponge” piRNA related machinery away from functional effector Piwi-piRNA complexes. This goes against what the reviewer suggested (“they have lost the domains involved in interaction with other piRNA factors”), but we do find in the literature instances where PIWI domains of Piwi proteins and other Argonaute proteins are able to interact with protein co-factors (PMID: 14749716, 17891150, 21427766). Also, the PIWI domain of these AC proteins contains 13-15 Arginines, which may be methylated and thus bound by Tudor proteins. For hypotheses such as these, no matter how unlikely, we cannot yet fully exclude a role related to TE silencing/expression. We note that we do observe other degenerated potentially full-length *piwil1* copies in the *A. calliptera* reference genome that were not originally annotated as protein coding genes. Also, we cannot exclude that there may be other additional non-reference *piwil1* copies in Lake Malawi that we could not identify from our abundant short-read genomic resources. We did not go into detail on this aspect as we feel that we lack the tools and resources to explore this adequately on a population level at this point in time. With more long-read genomic data we are currently obtaining from across Lake Malawi, we will be able to address this question more thoroughly.

3. “In fact of the three additional *piwil1*, one is a pseudogene, and of the other two, only one (*piwil1.4*) is expressed in gonads.” While it is true that of the expanded *piwil1* genes only *piwil1.4* appears to have detectable expression in gonads, we cannot ensure this is true across the entire Lake Malawi. We checked for *piwil1* expression in gonads of *A. calliptera* currently growing in our facility (originally isolated from Salima around Lake Malawi). Therefore, given the wide distribution of expanded *piwil1* genes across Lake Malawi, checking RNA expression levels on a population level across many Lake Malawi species and individuals would be more informative on the germline expression of the additional *piwil1* genes. Expression in gonads of *piwil1.3* cannot be excluded in particular species/individuals which carry this gene. We add two sentences to the **Discussion** addressing this aspect. Lastly, as mentioned above, there may be extra non-reference *piwil1* genes currently segregating in Lake Malawi and these could potentially be expressed in the germline.

In conclusion, we have now included textual changes to further clarify some of these aspects. With these changes in mind and taking into account the other complicated population-level aspects of the evolution and expression of *piwil1* genes in Lake Malawi cichlids, we still hold the firm belief that our suggestion of fast evolution of *piwil1* genes is reminiscent of an arms race interaction between piRNAs and TEs.

2. Fig. 4 shows that piRNAs are not produced by phasing in ovaries (except in the AC species). Two proteins, the endonuclease Pld6 and the RNA helicase Mov10L1 are specific to phasing in other species. It would be interesting to correlate the presence of phasing with the presence of these two proteins in the corresponding tissue.

Regarding Mov10L1, two proteins appear in the list in Fig. S2B, Mov10a and Mov10b.1. The authors should clarify the relationships between these two proteins and Mov10L1, as in other species Mov10 proteins (including Mov10a and Mov10b.1) have more general expression pattern and functions, while Mov10L1 is specific to piRNA biogenesis.

The presence of Pld6 in testes or ovaries does not correlate with the presence phasing signatures. This is unexpected. Could the authors comment on this?

That is a good point and one that is puzzling us as well. We have amended a figure panel and included additional data to address this point.

The expression of Pld6 in testes or ovaries according to our proteomics data (now **Figure S3B**) indeed does not correlate with presence/absence of phasing signatures. That may well be due to lack of depth and resolution in our mass spectrometry approach to thoroughly detect the entire proteome. In fact, at ~200 amino acids long PLD6 is fairly short, which may complicate its detection. Also, as we noted in our manuscript, the abundant proteins of the yolk fraction of ovaries have precluded protein detection in ovaries to levels like testes or brain. Thus, just because we do not detect PLD6 in some of our testes and ovary samples, it does not mean it is not expressed.

Besides the proteomics, we did check our mRNA-sequencing datasets for expression of *pld6*. We found it to be lowly expressed in ovaries versus testes, but this is the case in *A. calliptera* as well. In conclusion, expression of *pld6* alone does not explain the absence of phasing. This data is now included in **Figure S6H** and we comment on this directly in the text.

Thank you for your comment regarding Mov10L1. We have now noted our lapse of including data only on Mov10a and Mov10b.1, but not Mov10L1, which does participate in the piRNA pathway in other species. We did not include Mov10L1 because the zebrafish genome does not encode an ortholog and we did not check whether cichlid genomes had Mov10L1. We searched our OrthoFinder results for proteins in the same orthogroup as human and mouse Mov10L1 and found indeed cichlid Mov10L1 orthologs. Now we have changed **Figure S3B**, removing Mov10a and Mov10b.1 and including Mov10L1.

In addition, the expression of *mov10l1* in our mRNA-seq data (now included in **Figure S6H** and mentioned in text) also does not provide any obvious insights into how phasing is absent in cichlid ovaries besides *A. calliptera*.

In summary, we do not yet understand what determines the absence of phasing signatures in most cichlid ovaries. We propose that other factors may be at play, or it may be related with the catalytic activity of some of the proteins involved. It will be interesting to explore this in the future.

3. The size differences in piRNAs between ovaries and testes might be explained by their loading in different PIWI proteins. However, Piwil1 and Piwil2 are not specifically expressed in either tested or ovaries (Fig. S2B). Could the piRNA size difference reflect the expression levels of Piwil1 and Piwil2 in ovaries and testes? Or what could be the basis of this size difference?

Indeed, Piwil1 and Piwil2 are not specifically expressed in testes or ovaries but they do seem to be expressed at different levels (**Figures 3A** and **S3B**). Hence, the relative amounts of Piwil1 and Piwil2 and their favoured piRNA length may underlie this size difference. This is now rephrased in the discussion. At the moment, we lack the tools to experimentally determine the piRNA pool directly bound by Piwil1 and Piwil2, but this is something we would like to pursue in the future.

4. piRNAs have various functions in addition to repressing TEs. Since small RNAs of 24-35 nt show piRNA signatures, it would be interesting to indicate the percentage of them (potential piRNAs) corresponding the TE sequences vs other sequences.

That is true, it is clear that piRNAs have roles beyond TE repression, but these were not in line with the main purpose of our study, of acquiring first insights on piRNA-driven TE silencing in cichlid fishes.

Nevertheless, this is a good point and it is something we have shown in **Figures 6A-B** and **S10A-B**. While the vast majority of piRNAs overlaps with TEs and intergenic features, a minority (<6% of all piRNA bases) does overlap with promoters, exons and UTRs. We have now adjusted these figures, as suggested by another reviewer, in order to make clearly visible the percent of small RNA bases overlapping with UTRs, exons, and promoters. Also, we have added a sentence to the main text (page 11, lines 502-505) to highlight that some small RNAs do map outside of TEs and may have functions beyond TE silencing.

Reviewer # 2

Almeida et al, 2024 present first in-depth study into the co-evolution of piRNAs and TEs in African cichlids. This study uses variety of data from transcriptomics, proteomics and evolutionary genomics to show piRNA and TE co-evolution is rapid and dynamic across African cichlids from three lakes. They also present many additional exciting observations regarding piwil1 genes evolution and sex-bias in TE and piRNA cluster activity. Overall, it's potentially a landmark study for cichlid biology and opens a lot of fundamental questions regarding piRNA pathway evolution in vertebrates, especially during speciation events. There are several major concerns, however, which need to be addressed.

We are grateful to the reviewer for carefully reading our manuscript and for the extensive suggestions for improvement. Please find below our responses to the reviewer's comments.

1. Analysis cut-offs are very relaxed and insufficient details of QC results:

mRNA expression analysis has very relaxed parameters. In addition, piRNA cluster annotation very stringent parameters for length of the cluster whereas relaxed parameter for alignments considered. The latter is probably why median cluster lengths are less than 5kb in some species/gonads. It is also unclear if the low-mappability regions of the clusters were considered for merging with flanking uniquely-mapping piRNA 'seed' windows.

We appreciate the reviewers' comment and have now added clarifications on the parameters of our analyses in the main manuscript and below. Overall, we believe our chosen analysis parameters are robust and adequate for the analysis conducted.

The mRNA expression analysis was conducted with adequate parameters for interrogating TE expression at the family and single locus levels (with Tetrascripts and SQulRE, respectively), as we describe below in our responses to several of the reviewer's more specific comments. Following several of these comments, we have expanded/rewritten some sections of the methods. TE expression was quantified from alignments generated using parameters identical to the those advised by the developers of Tetrascripts and SQulRE. These programs were also ran using advised parameters, please see **Methods** for details. We provide clarification on the aspect of read-through transcription. Please let us know if there is still any aspect that requires clarification.

Regarding piRNA cluster annotation, we used a novel tool named piRNA cluster builder (piCB), to map piRNA clusters. Our initial collaborative manuscript that benchmarked piCB on seven different species - including *A. calliptera*, one cichlid species - has been accepted for publication in Cell Reports. A pre-print is available through SSRN (https://papers.ssrn.com/sol3/papers.cfm?abstract_id=4822917) and the piCB code is available on GitHub <https://github.com/HaaseLab/PICB> and <https://zenodo.org/doi/10.5281/zenodo.13376884>.

In brief, the piCB algorithm is based on the initial definition of piRNA clusters by the Hannon and Tuschl groups. The initial minimal piRNA cluster length of 5kb (Hannon lab) was based on technical parameters required to assemble piRNA clusters from first generation piRNA sequencing data. The small size of these data sets required a large sliding window (1 kb), a minimal coverage of only 1 read per kb, and an additional requirement for 5 positive scoring 1 kb windows to call a piRNA cluster. Modern sequencing data provide a much deeper insight into piRNAs and allow for a more precise definition, which enables the detection of piRNA clusters that are smaller than 5 kb. The large size and depth of current sequencing data required a new computational tool.

piCB improves upon the initial code by stepwise integration of unique mappers, primary alignments for multi-mappers and then all possible mapping positions (up to 100 according to our mapping cut-off) to build three types of intervals. First, “seeds” are assembled using only uniquely-mapping reads. Then, primary alignments of multi-mappers (one alignment at random for each read) are integrated and used to extend seeds to “cores”. Lastly, all multi-mapping alignments (up to 100 according to our mapping limit) are integrated, thus extending and potentially merging “cores” into “clusters”. This results in three cluster types: (1) clusters that are solely defined by unique mappers called “Solo-Core-Clusters”, (2) clusters that contain a single seed and are extended by multi-mappers called “extended-Core Clusters”, and (3) clusters that contain multiple seeds that are connected by multi-mappers “Multi-Core-Clusters” (Figure 1A-B of the preprint https://papers.ssrn.com/sol3/papers.cfm?abstract_id=4822917).

piCB contains a feature to optimise parameters according to the depth of the piRNA sequencing data. We used the default parameters of piCB to map cichlid piRNA clusters as these represent ideal conditions for high-quality data. This was benchmarked and we now include an additional supplemental figure with this benchmarking for two replicates of AC testes (Figure S7), as well as another Excel sheet with benchmarking data added to Supplemental Table 2. Benchmarking is described below.

To determine the optimal number of unique mappers per sliding window we generated cichlid piRNA clusters from with a range of values of unique mapping reads per 350 nt sliding window (Figure S7A-B and Supplemental Table 2). The default RPKM is 2, which in this case corresponds to 7 unique mapping reads per window. Stricter parameters require more unique mapping reads per window and result in less clusters, less genome space and lower fraction of the library explained by the clusters (towards the left of the panels in Figure S7A-B). It is up to the researcher to a decide regarding strictness of clustering based on the curves shape and specific task (Figure S7A-B and details on Supplemental Table 2). The default parameter corresponds to 3494 clusters, occupying 4% of the genome space, and explaining 80% of the mapped reads.

To illustrate the adequacy of the default parameters we chose, we can interrogate cluster productivity when using stricter and more relaxed parameters corresponding to 20 and 2 unique mappers per window, respectively (Revision Figure 2). With the stricter parameters (we can explain only 73% of the mapped reads (vs 80% in default), while with the more relaxed settings we obtain a large number of additional clusters that are lowly productive (12,320 vs 3494 in default, Revision Figure 2). Therefore, we decided to settle on the default settings,

which explain a higher proportion of the mapped reads and do not add many lowly productive clusters.

Revision Figure 2. Comparing different piCB parameters in one small RNA library of *A. calliptera* testis. (A-C) Ranking of piRNA clusters according to productivity in one library of *A. calliptera* testis. Different panels use loose (A), default (B), or stricter (C) parameters. Cumulative fraction shown as an orange line. Dashed red line indicates the 90th percentile of piRNA production and the number of clusters producing 90% of piRNA reads in clusters.

With these parameters, the number of clusters is substantially lower than the number of seeds and cores, and their total length is larger (**Figure S7C**). This is caused by aggregation of a few cores into multi-core clusters (**Figure S7D**). In fact, multi-core clusters are a minority, but correspond to approximately half of the genomic space of all clusters, explaining the majority of the reads. No biases are observable upon a closer look at the distribution of clusters in both strands of AC chromosomes (**Figure S7E**).

To address reproducibility of the clusters we compared the clusters in replicate 1 of AC testis with a second replicate of AC testis. In terms of genomic space, these two replicates share 56.3 Mb of sequences annotated as piRNA clusters (out of 72.1 Mb total in replicates 1). Then, we compared the reproducibility of these clusters to a random process by computing the ratio of common (reproducible) genomic space length to reduced (either of the replicates) genomic space length, using a bootstrapping approach. Bootstrapping was conducted by relocating the clusters to random coordinates and strand, while keeping the number and length of the clusters.

After 10,000 iterations:

Number of BootstrappingScores: 10000

Number of BootstrappingScores higher or equal than the observed: 0

Max of BootstrappingScores: 0.0275460790993278

Observed: 0.601170716593635

These results have an associated p-value of less than 10^{-4} , suggesting that the overlaps we observe between clusters are not random. To further address reproducibility, we verified how reproducible is the length of the cluster and how this relates with productivity. Most of the clusters identified by piCB are highly reproducible (in terms of their width) and productivity affects this reproducibility, with lowly productive clusters being less reproducible overall in terms of their length (**Figure S7F**). With this in mind, we now include two new panels B and C in **Figure S9**, which repeat the cluster intersects of **Figure 5A-B**, but using only the 50% most productive clusters in each library. **Figure S9B-C** show results identical to those originally presented, suggesting that the divergence in piRNA clusters in cichlids is not due only to lowly productive clusters.

We hope this brings clarity and illustrates the robustness of the approach and parameters we used. Please let us know if there is still any aspect that requires clarification.

2. Figures 1 and 6 are extremely information dense but most panels are only superficially discussed or mentioned. A reader who is not expert in TE or piRNAs reading the manuscript will have difficulty understanding the reasoning behind the analysis.

Thank you, we agree, and we have now made important adjustments to efficiently report results to a wide audience. We focused on textual changes to better contextualise the results, because we believe the figure panels included in Figures 1 and 6 are important and should remain in these main figures.

For **Figure 1** we do believe the first paragraph of the results section “TE transcriptional activity in cichlid gonads and early development” serves as a good introduction into why gonads were chosen and why we chose each representative species. Likewise, we explain the rationale of looking at early development in the beginning of the last paragraph of that section. In this section, we now make clearer the rationale for doing locus level analysis of TE expression and redirect the reader to the discussion where these results are put into context.

Regarding the text section referring to **Figure 6**, in the revised version we provide an additional supplementary table and details in the methods into how we defined TE families likely to be transpositionally active, which was one of the aspects generating some confusion. Importantly, we now split the main text relating to **Figures 5 and 6** in two separate sections. This, along with several textual changes added to further smoothen transitions and the reasoning behind particular analysis makes the text clearer and easier to follow.

3. Authors also discuss and/or compare their findings regarding enrichment of TE expression in early embryo and piwi gene structure to that of model organism, zebrafish and sometimes widely to all animal models. This is a strength of the study and will make the paper broadly accessible to the community. However, in some places where authors try to connect diverse and dynamic TE expressions to cichlid species diversity, such comparisons are essentially missing from other main figure panels where comparative analysis between AC and ON would be insightful (such as Fig. 6H, Fig 4, Fig. 2E). Overall, if the authors mean to link cichlid diversity to divergence in genomic TE abundance, age or expression, this needs to be more directly discussed.

We appreciate the reviewer’s comment that the comparative aspect is a strength of the study. While we indeed frame a portion of our results in a teleost or vertebrate “big picture”, it was not our intention to make this work a comparative resource per se. Given their history of adaptive radiation characterised by extensive phenotypic diversification in a background of

low genetic divergence, East African cichlid fishes are unique, full-fledged models in their own right and a thriving research community studies these fish models to understand the genetic basis of phenotypes and of fast adaptation. Our intention was to characterise TE activity and the small RNA-driven pathways of cichlids that antagonise TE activity. This was important groundwork to understand TE biology in cichlids better, before moving on to address how TEs and their transcriptional/transpositional activity are relevant to cichlid phenotypic diversity. This is ongoing work well beyond the scope of the work described in this manuscript. We do discuss in the last two paragraphs of the discussion how we currently conceptualise the role of TEs in cichlid diversification. As further data is needed to establish clear correlations between aspects of TE biology and cichlid diversification, we prefer to refrain from adding much more speculative text to our manuscript.

Regarding the comparisons between AC and ON the reviewer suggested:

-Figure 6H (now Figure 6G in the revised version of the manuscript). In this figure panel we show the embryonic expression of TE families likely to be recently active versus related families (of the same superfamily) for which we have no evidence of transpositional activity in Lake Malawi in terms of allele frequencies and number of polymorphisms. This analysis is now clearly described in the methods and in the main text. Importantly, this assessment of activity was obtained from genomic resources the Cambridge cichlid research community, including the Durbin, Miska and Santos groups has accrued in recent years, of cichlid fish populations of Lake Malawi and surrounding bodies of water. These represent abundant genomic resources at a population level that we do not have, and which we do not think exist, for fishes of the other Great Lakes and Nile Tilapia. As such, at the moment it is impossible for us to assess TE transpositional activity in other cichlids with the same degree of confidence by applying the same methodology. Furthermore, besides the fine-scale early development RNA-sequencing datasets we collected for two Lake Malawi species, there is currently no comparable temporal time-course data for early development available for other cichlid and tilapia species. In conclusion, there are no resources to extend this analysis to other species, and consequently it is currently impossible to introduce a comparative angle to this figure panel.

-Figure 4. This figure is by itself comparative, with the small RNA information for each cichlid species laid out side-by-side. All these species share identical length distribution profiles and sequence signatures, and this is mentioned in the text. The only diverging aspect is also mentioned in the text – the signature of phased piRNA biogenesis in ovaries observable only in AC ovaries, but not on those of AB, PN, and ON.

-Figure 2E. The *piwil1* expansion was detected only in cichlids of Lake Malawi, not in cichlids of other Lakes. Consequently, we cannot expand this test for signatures of selective sweeps as in Figure 2E to other East African cichlids, as these lack the extra *piwi* genes. We did test whether annotated *Piwil1* coding sequences of representative African cichlid genomes have signatures of positive selection using Datamonkey/HyPhy and PAML packages and found support for positive selection in specific amino acid residues of *Piwil1* proteins (**Figure S5A** and **Supplemental Table 1**). These results are described in the main text and methods.

Overall, we believe we have added a comparative dimension throughout our manuscript wherever it made sense to add or emphasise such comparisons.

4. The conclusion of the paper is conflicting. One hand authors show that >90% of TE families are expressed in gonads and another hand that piRNA-mediated silencing of TE mRNA is active. This could be addressed by showing correlations between TE mRNA expression and piRNA abundance/ping-pong signal. Are the highly expressed TEs not regulated by piRNAs?

We apologise for the confusion. The revised manuscript has altered text passages with this aspect clarified.

This is related to a seemingly paradoxical paradigm well known in the small RNA field: a degree of gene expression is required to have gene silencing. Without an RNA to serve as a template and substrate for small RNA production, no small RNAs could be produced, and no silencing could be established. Under this light, it follows that even robust silencing will not be, by definition, equating to 100% silencing of the target gene. As we discuss in detail below in response to another comment, we did not mean to equate “expression of TE families” with “high expression of TE families”. By “expression” we meant “detectable expression” of TE families. That is why we settled on a fairly relaxed cutoff for expression (please see below the response to the other comment on this). We have now altered the text referring to TE family expression accordingly to clarify this point. Despite this, our main conclusion reporting dynamic expression of TEs is supported by our data, because: 1) we are able to detect expression of > 90% of TE families in gonads and during early development; 2) we detect TE families and individual TEs with distinct expression patterns in gonads and during early development, including of TEs which are likely to be transpositionally active (**Figures 1, 6, S1, S2, and S11**).

Because TE transcripts targeted by piRNAs are processed into piRNAs themselves, highly expressed TEs are expected to also have high levels of piRNAs mapped to them, i.e. a positive correlation between TE (or TE family) expression and piRNA levels mapping to that TE (or TE family). Indeed, this is what we observe in our cichlid libraries (**Revision Figure 3**).

Revision Figure 3. Correlation between mRNA expression and levels of mapped piRNAs to TE families. (A-B) Scatter plots showing the mRNA expression levels of TE families (x-axis) and the levels of piRNAs mapped to TE families (y-axis). Each datapoint represents the aggregated data for one TE family. (A) shows data from testes libraries, whereas (B) shows data for ovary libraries. Adjusted R^2 and associated p-value are shown in each panel. AC, *Astatotilapia calliptera*; rlog, regularised log.

In summary, we have made textual changes to address this point and include a revision figure for clarification. However, we believe the inclusion of these figures in the manuscript will cause confusion to researchers who are not small RNA “aficionados” and detract from main message and findings of our work.

Specific Major Concerns:

Page 4, Line 56: Given the age distribution of LTR TEs between AB, PN and ON, AC genomes, its expected and reassuring that in Fig S1B, fewer LTR families are expressed in AB and PN because LTRs composition of their genome is smaller and older (10-20% divergence) compared to LTRs of ON and AC. However, the number of LINE families expressed in all 4 species are quite similar despite the abundance of young (<5% divergence) in ON and AC genomes. Have the authors investigated this further for any possible explanation?

The “age” of LINE insertions as measured by divergence from a modelled consensus should be interpreted with caution. It is likely that the LINES in the genomes of AC and ON, the best cichlid genomes currently available (not very fragmented, chromosome-level assemblies), have a “younger age” because these LINE repeat consensus could be better modelled due to the higher quality of the genomes. Curation of the modelled TE libraries is key here as well.

Note that when we map TE expression to our non-curated TE annotation for AC, there is a comparable number of expressed LINE families to the other species for which we use non-curated TE libraries (AC - 178, AB – 213, PN – 202, ON – 202), but distinct from the number of expressed LINE families obtained using AC's curated TE library (70, see **Figure S1B**).

As LINEs often truncate the 5' end, it is complicated to model full-length LINE elements and to obtain reliable measures of divergence. We did not do a dedicated analysis to address this, and it is thus not advisable to conclude much from LINE "age". To establish the age of LINE elements, a careful alignment and observation of the models in the TE libraries obtained for the different cichlid genes is necessary, but we believe this level of detail goes beyond the purpose and scope of our study. Once improved genomes and curated TE libraries are produced for representative cichlids of all East African Great Lakes, this issue will be more easily addressed.

Page 6, Line 3: Authors state that African Cichlids have substantial transcriptional TE activity. This can be misleading to non-TE researchers. First, a quantification of proportion of TE-derived mRNA across gonads and developmental stages is needed, which will help orient readers the contribution of TEs to the transcriptome in Lake Malawi cichlids compared to ON cichlid and zebrafish etc. Second, since authors have already quantified Locus-level TE expression, it shouldn't be too much work to distinguish how much of TE transcriptome is TE-derived and not from read-through transcription of nearby genes. These two analyses can help authors convince readers that African Cichlids are exceptional in their TE expression or abundance of TEs in their transcriptome which may contribute to their phenotypic diversity.

These are good points. We include a revision figure and added a new figure panel to the manuscript addressing this.

Of note, we do not think that the role of TEs in the cichlid radiations and their phenotypic diversification will be necessarily correlated with high TE expression (versus other genes) and/or increased TE abundance in the genome, and it was not our intention to convince readers of such. Rather, we think it may be a combination of TE recent transpositional activity and the polymorphism it generated and the actual genes whose gene expression may be rewired by TEs. We do articulate our thoughts on this matter in the last paragraph of the Discussion and in the Conclusions. We are further exploring the connection between TEs and the astonishing phenotypic diversification of these fishes. This work is ongoing and is beyond the scope of this manuscript.

That said, the reviewer does bring important aspects to consider. We have answered the first point by checking the proportion of library counts assigned to genes and TEs, using the tables of counts produced by TEtranscripts. Overall, only 1-2% of the libraries correspond to TE-mapped reads (**Revision Figure 4**). This does not represent substantial TE activity in absolute terms. We rephrased the sentence the reviewer referred to (now in page 5, lines

234-236), to make it reflect more accurately what we originally meant that many TE families seem to have detectable expression and dynamic expression patterns.

Revision Figure 4. Proportion of sequencing libraries mapping to protein-coding genes or TEs. (A-B) Proportion of counts in sequencing libraries in gonads of representative East African cichlid species **(A)** and early development of Lake Malawi cichlids **(B)** that map to protein-coding genes or TEs. In both cases, the vast majority of the counts map to protein-coding genes. AB, *Astatotilapia burtoni*; AC, *Astatotilapia calliptera*; MZ, *Maylandia zebra*; ON, *Oreochromis niloticus*; PN, *Pundamilia nyererei*; TM, *Tropheops* sp. ‘mauve’.

Then, there is the relevant point of read-through transcription. We refrained from including such data in the initial submission as the available gene annotations for cichlid genomes could be improved and are not on par with gene annotations currently available for model organisms such as zebrafish. In fact, some of the authors are involved in ongoing efforts to produce higher quality reference genomes and gene annotations for East African cichlids. In any case, as this point was brought up by the reviewer and the handling editor, we now include this information in **Figure S2E** and mention it adequately in the text. This was most easily addressed in our embryogenesis datasets, where we had readily available analysis of TE expression at the single locus level. Due to the annotation issues mentioned above, we reasoned the “cleaner” way to do this analysis was to categorise mapped reads according to overlaps with annotated gene features. We defined overlaps with gene features identically to Chang et al., 2022 (PMID: 34987056) and assessed the proportion of counts in a library mapping to each feature. As can be observed in **Figure S2E**, most of the counts map to intergenic regions (between 38.5-49.3%). Between 0.2-0.6% of TE counts are embedded within 5' UTRs, and between 5.9-10.8% within 3' UTRs. The 12.2-17.1% proportion of counts overlapping with exons is likely due to transposons misannotated as protein-coding genes. We are currently curating the gene annotation to comprehensively remove such

misannotations, but the curated version will not be available within the timeline of this revision. This analysis should be revisited once higher quality annotations are available for cichlids. We split intron mapping counts in two classes, as defined in Chang et al., 2022 (PMID: 34987056) and observe that 3.8-19.3% of counts map to TEs within introns of non-expressed genes, while 18.8-24.3% of counts map to TEs within introns of expressed genes. For the latter, TE transcription can be either independent or dependent on the expression of the gene.

These results do not affect our conclusions, with the majority of counts mapping to TEs within intergenic regions. This piece of information is now added to the manuscript.

Page 23, Line 38: When mapping to TE family consensus, too many multi-mappers were allowed. TE family, by definition, are unique sequences for each designated family and thus

If substantial multi-mapping reads were detected, it's very likely that some consensus sequences are not well resolved and thus expression of some families is overestimated. I suggest authors try stricter parameters such as `-outFilterMultimapNmax 3` and compare how much reads are unmapped compared to their original parameters. It seems the authors did reduce some of the overestimation by curating the AC consensus TE library but more details of the mapping statistics either in a supplemental table or figure is also needed.

We apologise, but there appears to be some confusion. We did not map mRNA reads to TE family consensus *per se*, but to TEs annotated in the genome, as indicated in the methods (page 22, lines 1050-1066). The resulting BAM files were used for quantification of TE expression at the family level using Tetrascripts. Tetrascripts is one of the most widely used tools in the field to quantify TE expression at the family level (PMID: 26206304, 29508296, 32576954). The mapping used is indeed relaxed, but that is what is intended (PMID: 26206304, 29508296, 32576954). These settings have been successfully used in our laboratory previously and are similar to the settings recommended by the authors of Tetrascripts of the Hammell laboratory: <https://github.com/mhammell-laboratory/Tetrascripts> (section: *specific recommendations when using STAR*).

In the revised version of the manuscript, we now include **Supplemental Table 4**, with all the small RNA and mRNA mapping statistics, which the reviewer can consult.

Page 6, Line 55: Authors found similar PiggyBac insertions in 3' UTR of *piwil1* genes in AC is a very intriguing finding. It seems the authors are implying that *piwil1* could have been duplicated by PiggyBac transposition events. Putative TIR sequences shown at the borders of *piwil1* gene copies are convincing, but authors do not show the length of complete TIRs from the corresponding family consensus. Is the 9nt sequence shown in the white boxes the complete TIR sequence or only partial? Some PiggyBac TEs are well known to also have asymmetric TIR sequences.

Thank you for your comment. It prompted our revisiting of the TIR sequences and the revised manuscript includes a more comprehensive analysis of these sequences.

First, a brief clarification. We are proposing that PiggyBac transposition could have had a role in the expansion of *piwil1* genes, presumably early on in the radiation of Lake Malawi. We are not claiming in the manuscript that the four *piwil1* copies we detected are exclusively originating from transposition of such Piwi-associated PiggyBac. Also, we do not have a way to conclusively reconstruct the succession of events. Thus, we cannot exclude that recombination events played a role, as we discuss in page 13, lines 624-626. Also, as we note in lines 350-354, pages 7-8, we do observe instances where *piwil1* genes are not in linkage with PiggyBac-1 elements, suggesting that recombination is a force at play.

The completeness of the TIR sequence was an excellent point, thank you. We initially had a look using the consensus from DFAM (<https://dfam.org/classification/dna-termini>), however, after the reviewer's comment we had a closer look and found that the TIR of the consensus of the PiggyBac-1 family (the *piwil1*-associated PiggyBac family) in our curated TE library is symmetrical but actually slightly longer, comprising a total of 16 nucleotides:

(ttaa)cccttgtaggtgttc and gaacaccacacaaggg(ttaa)

The sequence between parenthesis is the duplicated target side. In red, there is a mismatch. With this complete TIR sequence at hand, we revisited our analysis of the *piwil1* loci sequences to reannotate the complete TIR. Indeed, we could find the complete TIR in the right flank of all three *piwil1*-associated PiggyBac-1 TEs (please see new version of **Figure 2A**). In the PiggyBac-1 associated with *piwil1.4* we even found an intact TIR in its left flank. Finding potential alternative TIRs upstream of "Region S" and the *piwil1* genes was challenging, as in our original analysis. We did TBLASTN of the left and right complete TIRs and found no convincing hits with at least 9 matches, allowing one mismatch. We were more successful, as in our original analysis, in identifying sequences upstream of *piwil1* genes that resemble TTAACCCTT (allowing one mismatch), the expected left target site and the extremity of the TIR. In the revised version of figure 2A, we now annotated additional TTAACCCTT sites (with one mismatch) upstream of *piwil1* genes that could have presumably been used by a PiggyBac transposase as an alternative cut site. Besides amending **Figure 2A**, we added the BLASTN results to **Supplemental Table 1** and amended the main text and methods to incorporate this new analysis.

Overall, the revised manuscript includes more comprehensive analysis of the TIR sequences of the *piwil1*-associated PiggyBac elements, and we thank the reviewer for the suggestion. These results do not change our initial conclusions and interpretation.

Figure 2C: It is not clear where the LT and LV species belong in this plot. Are they even in this phylogeny? Annotation of species AC, MZ etc along the phylogeny will be helpful. Did the

authors try to trace if the expansions of *piwil1* genes happened before the adaptive radiation in Lake Malawi or afterwards?

Species from lakes Tanganyika (LT) and Victoria (LV) are purposefully not included in **Figure 2C**. This is stated in the legend of **Figure 2C** (“Presence (green)/absence (black) of each *piwil1* gene in genomes of **Lake Malawi and Tilapia cichlids.**”), as well as in the main text (see first paragraph of results section “Evolution and functional potential of *piwil1* genes in Lake Malawi cichlids”).

We did not include LT and LV species in the figure precisely because we found evidence supporting duplications of *piwil1* genes in Lake Malawi (LM) cichlids, but not in representatives of LT and LV (*Pundamilia nyererei*, *Astatotilapia burtoni*, and *Neolamprologus brichardi* genomes currently available in Ensembl), according to OrthoFinder results (**Supplemental Table 1**) and as mentioned in the main text (see first paragraph of results section “Evolution and functional potential of *piwil1* genes in Lake Malawi cichlids”). This argues against the *piwil1* gene expansion happening in a common ancestor of the three Lake radiations, but is in line, as we argue in our manuscript, with *piwil1* expansion in an ancestor of the LM radiation. Thus, we restricted analysis shown in **Figure 2C** (summarised in **Supplemental Table 1**, sheets **D** and **E**) to LM haplochromine cichlids (radiating lineages) and Tilapias from the Malawi catchment. Our analysis of the distribution of *piwil1* genes in LM cichlids was conducted using the extensive genomic resources from LM that the Cambridge cichlid research community has accrued in recent years, including both short-read and long-read genomes (summary of the analysis in **Supplemental Table 1**, sheets **D** and **E**).

According to the OrthoFinder analysis (**Supplemental Table 1**), representative species of LT and LV did not show extra *piwi* genes. While we cannot fully exclude that smaller scale *piwi* gene expansions have happened in cichlids of other lakes, in parallel to the LM radiation, we also do not have any evidence supporting such occurrences. Nevertheless, we note that our OrthoFinder analysis was sufficient to identify the LM expansion. Also, to our knowledge, all publicly available genomic resources of LT and LV cichlids consist of very fragmented short-read genomes, not up to modern standards in genomics. Thus, we do not think that the exploration of *piwi* expansions in LV and LT cichlids using subpar short-read genomic data would allow us to produce robust conclusions.

Page 8, Line 32: I also recommend authors consider alternative explanations, such as PiggyBac TEs maybe targeting *piwil1* genes. It seems the *piwil1.1* also has a more recent PiggyBac insertion in the S-region. Such targeting of another piRNA pathway genes has been reported for flies (PMID: 33347429). Since PiggyBac insertions are also apparently enriched near *piwil1* genes than the expected genome average, this might support the alternative hypothesis. I also suggest authors investigate if ‘TTAA’ (preferred site) tetranucleotides are more common in the *piwil1* genes neighborhood than other flanking genes? Overall, I think the conclusions made are very exciting and provocative, but the evidence presented for PiggyBac mediated expansion of *piwil1* is at best weak.

To address this point, we add data in the revised manuscript on TTAA distributions and provide clarification on the PiggyBac insertions the referee is referring to.

Regarding *piwil1.1* having a more recent PiggyBac insertion in the S-region: that is not accurate, and we do not make any such claims. In **Figure 2A** PiggyBacs are annotated in orange, while other non-PiggyBac DNA transposons are annotated in brown. Within the Region S of *Piwil1.1* we find DNA transposons (annotated in brown), but these are not PiggyBacs. Granted, the orange and the brown colours shown in the figure may be similar in hue, but we think they are still distinguishable. The similarity in hue was intentional as PiggyBac transposons are a class of DNA transposons.

Regarding the TTAA distribution in *piwi* genes vs others, we thank the reviewer for bringing this up, it is an excellent point. We have now tested whether the flanking regions (+/- 5kb) of *piwi* genes have a distinct frequency of TTAA when compared to the flanking regions (+/- 5 kb) of other genes (new data included in **Figure S4E**). Also, we have conducted a similar test comparing *piwi* gene sequences (including exons and introns) with the sequences of other genes (including exons and introns, new data included in **Figure S4F**). The flanking regions of *piwi* genes do not have significantly different numbers of TTAA sequences versus the flanking regions of all other genes (Welch's t-test, p-value = 0.86, **Figure S4E**), but the *piwi* sequences do show slightly higher TTAA numbers than all other genes (Welch's t-test, p-value = 0.018, **Figure S4F**). Of note, *piwil1.1* (the original copy) shows the greatest number of genic TTAAAs (**Figure S4E**). If there would be active targeting of *piwi* genes by PiggyBac TEs, one would expect that *piwil1.1*, the *piwi* gene with most TTAAAs, would have PiggyBacs in its vicinity, but this is not the case. Thus, we believe this data does not support the alternative model of active targeting. That said, there does seem indeed to be a slight tendency for some *piwi* genes in cichlid genomes to have more TTAA. We have included these results in **Figure S4E-F** and adjusted the text accordingly.

A note on the strength of the conclusions. How can we know for sure, with 100% certainty, whether transposition drove *piwil1* expansion? We simply cannot, such is often the case with studies on genome evolution. However, we do present several lines of evidence in line with transposition playing a role: 1) association of *piwil1* genes, besides the original *piwil1.1*, with one family of PiggyBac TEs; 2) the *piwil1*-associated PiggyBac TEs are closely related even within the same family; 3) we find sequence signatures of PiggyBac required for their transposition in the borders of Region S and the PiggyBac: their terminal inverted repeats and the preferred target sequence TTAA. We do not claim transposition was the sole driver of the expansion of *piwil1* genes: as we mention in the discussion, transposition may only partially explain what happened, with recombination-based gene expansion playing a role. Thus, the sentence that the reviewer pointed out (Page 7, lines 329-332 in the revised manuscript version) still holds true: these observations are compatible with a model involving transposition (although not exclusively, which we disclose).

To summarise, the new data we include in the revised manuscript do not alter any of our initial conclusions, instead making our findings more robust.

Figure 5 A&C: These circos plots are very information dense and can be very helpful in comparing the genomic distribution of clusters, association with TEs and conservation. However, some of the tracks do not add much more value to the plot. For example, piRNA coverage on both strands because of log2 transformation in such large window sizes is not informative. It is reassuring to see the apparent positive correlation between piRNA clusters and piRNA coverage, but this can be easily demonstrated by a dot plot such as shown in Figure S5. Plus, the gray background under the coverage track is very distracting. Additionally, the legend for tracks 5-6 is confusing. Does this indicate that the clusters were reproducibly detected across species or across tissues of the same species? Some explanation of what the Y-axis values here mean will be helpful.

Overall, I suggest the authors simplify these plots to highlight their salient points, which is rapid divergence of piRNA cluster activity across the species and strong sex-biased activity of many clusters. In addition, the TE distribution track is very small and thus difficult to correlate to piRNA cluster peaks. TE or any genomic features in the outermost track would help readers visualize the correlation much better.

Thank you, we agree that visualisation of the circos plots could be improved to increase their effectiveness of communicating the main message of cluster divergence. We now provide new versions of the circos plots in **Figure 5** including the shared and divergent clusters in Lake Malawi species and ON, which is in line with the main message we are trying to convey. In addition, we now add two circos plots to the supplement (**Figure S8**), which provide a visual complementation of the data in **Supplemental Table 2**, allowing visualisation of all the clusters. We believe these changes improved the manuscript.

Figures 5D and supplementary data: MZ testis cluster counts and ON ovary cluster counts are highly variable across replicates (>40% of ON ovary clusters in one replicate is not found in other replicates?). This indicates low quality libraries in one or more replications. Did the authors investigate this? Additionally, figure legend, results and even methods do not indicate any parameters of how the cluster overlap and sharing was evaluation between sexes and species. Was there a liftOver performed?

According to the reviewers' suggestions, we have added a more detailed description in the main manuscript and below.

We do not think that cluster count variability represents low quality libraries for three main reasons, which are described below.

First, we have conducted extensive quality control on all our samples using: 1) fastQC v0.11.9; 2) confirming a high percent of reads mapped to the respective cichlid genome (results in **Supplemental Table 4**); and 3) by checking DESeq2 normalised counts of samples on PCA. The results depicting sequencing data that were included in the manuscript were obtained

using only samples that met our quality control requirements. The higher variability in MZ is most likely because we only used two biological replicates due to sample availability, versus three or four biological replicates for other species' samples.

Second, these are biological, not technical, replicates, and as such a degree of variability is expected. We believe that the reviewers will appreciate the reproducibility of the piRNA clusters identified (demonstrated in **Figure S7F** of our manuscript). Furthermore, these fishes have been shown to have substantial structural variation attributable to TEs, even within species (3-4% intraspecies divergence and 5-10% interspecies divergence, <https://www.biorxiv.org/content/10.1101/2024.03.28.587230v1>). We show in **Fig. S10C**, that piRNA clusters overlap with structural variants identified in the Lake Malawi using a pangenomic approach. This indicates that piRNAs are being produced from regions that are polymorphic in distinct closely related species and even within species, which necessarily implies biological variability. This is an interesting point and one that may be relevant to cichlid biology and their radiations in East Africa, but a dedicated study is required to adequately address this.

Third, the aim was to identify all possible regions producing piRNAs and showing the properties of these clusters, without any arbitrary filtering based on piRNA cluster size or piRNA abundance or other features. Instead, ranking approaches, such as the one we employ here with piCB, take into account more information from the libraries and may reveal novel aspects of piRNA biology. These criteria, together with the polymorphism described above may underlie the piRNA cluster count variability.

To clarify our methodology, we have used piRNA Cluster Builder (piCB), a novel tool available as an R package, to map piRNA clusters developed by the Haase laboratory. We have now updated our methods section "**Bioinformatic analysis, Small RNA-sequencing analysis**" describing the method and citing the original preprint describing the method. In the meantime the work describing piCB has undergone peer-review and is currently in press. Overlaps were calculated using function PICBCombine() of the PICB R package, which calculates overlaps based on GenomicRanges::intersect(). We now include information in the methods describing the procedure used to calculate overlaps.

Page 26 and Figure 5A, C: Was there a separate annotation performed for uniquely mapping reads as well? This is necessary to show which clusters are truly high confidence and perhaps this will also improve the reproducibility issues earlier due to using multi-mapping reads.

We apologize for the confusion about the new clustering tool (piCB) and have now added more information according to the reviewers' suggestions. piCB automatically performs a dedicated annotation of uniquely mapping reads to determine "seeds". Upon stepwise integration of multimappers, these "seeds" are extended and possibly merged with nearby "seeds". All clusters contain such "seeds" that depend on unique mappers. As the reviewers noted, the difficulties of attributing multimapping reads can result in slight changes in the

genomic coordinates of piRNA clusters. However, piCB's requirement for uniquely mapping "seeds" ensures high confidence 'anchors' in all assembled intervals.

There is still a considerable amount of discussion and many methodologies to map piRNA clusters, but there is no "gold standard" in the field. By definition, piRNA clusters are enriched in repetitive sequences and transposable elements. Using only uniquely-mapping reads when annotating piRNA clusters may therefore lead to a massive loss of information and paint an incomplete picture.

To illustrate this, consider a simple model genome consisting of two chromosomes. On chromosome 1, a TE is present and on chromosome 2 there is a piRNA cluster protecting the organism against the TE on chromosome 1 (**Revision Figure 5**).

Revision Figure 5. Simple two-chromosome model to illustrate reliance on uniquely-mapping and multimapping reads when annotating piRNA clusters. See text for details.

It is expected that most of the sequenced reads will have two potential matching positions in the genome due to the considerable sequence similarity between the TE and the piRNA cluster. Such multimapping alignments are shown in grey in the **Revision Figure 5**. A smaller subset of reads will uniquely align to a single location within the genome, resulting in unique mapping alignments, which are shown in black. These specific reads are expected at the cluster's 5' and 3' ends, where the sequence derived from the TE integrates with the remainder of chromosome 2. Relying exclusively on unique mapping alignments poses an obstacle as it would only detect a truncated version of the piRNA cluster. Instead of capturing the entire cluster, it would yield two short segments, each representing the terminal regions of the piRNA source. On the contrary, considering all available alignments for analysis introduces the risk of false positive identification of the TE on chromosome 1 as a piRNA cluster.

Our approach to map piRNA clusters using the piRNA cluster builder (piCB) tool was extensively benchmarked in vertebrate genomes, a preprint with detailed methodology and demonstration of its power is now available (https://papers.ssrn.com/sol3/papers.cfm?abstract_id=4822917), has undergone peer-review in Cell Reports and is currently in press. The main advantage of this approach is to comprehensively build clusters using uniquely-mapping and multi-mapping reads. Please see

the preprint, our methods section, and the response to your first comment above for additional details.

Minor Concerns:

Page 5, Line 50: When describing the mRNA-seq results for TEs in gonads and embryos, authors highlight that >90% of TE families are expressed. This is very striking indeed (because it is from wildtype fish) but could be attributed to the low cutoff used for classifying TE expression (also see below in minor concerns). If this low cutoff was intentional (perhaps to account for piRNA-mediated slicing of TE mRNA in these Wildtype fish?), authors should state that in the result.

This is a good point, thank you. Using a fairly relaxed cutoff was the purpose here. As discussed above, there is a seemingly paradoxical paradigm in the small RNA field: a degree of gene expression is required to have gene silencing. We did not mean to equate “expression of TE families” with “high expression of TE families”. By “expression” we meant “detectable expression” of TE families. We have now altered the text referring to TE family expression throughout the manuscript accordingly to make what we mean clearer.

When we explore the percentage of TE families detected using different read cutoffs, we can observe that our chosen cutoff (at least 10 reads in at least 2 samples) locates at or near the inflection point of the sigmoidal curve (**Revision Figure 6**). This is true both for our embryonic datasets (**Revision Figure 6A**) and for our gonad datasets (**Revision Figure 6B**). Thus, we think the chosen cutoff is adequate to show “detectable TE expression” in our datasets.

We have now added text to the methods section addressing this point.

Revision Figure 6. Percentage of TE families detected with different read cutoffs. (A) shows this analysis done with mRNA-sequencing data from embryo datasets of *A. calliptera*, whereas (B) shows the analysis with gonads mRNA-sequencing. In (B), the left panel shows the minimum number of supporting reads in at least four samples, while the right panel shows the minimum number of supporting reads in at least two samples.

Figure 2A: Also, in the entire *piwil1* gene shown, are these 9nt sequences in either direction only present in the places shown with white boxes? Or are they more common? All the sites of the 5 or 9nt sequence should be shown if there are more.

That is a good point, we thank the reviewer for bringing it up. We have now revisited the annotation of the putative TIRs, after one of the comments of the reviewer (see above) about the completeness of the TIR, and reannotated **Figure 2A**. As discussed above, we annotated the complete TIRs. Additionally, we did BLASTN using the complete TIR sequences, searching across the 50 kilobase regions of the *A. calliptera* genome including each *piwil1* gene, but this approach failed to reveal any convincing hit consistent with an alternative PiggyBac TIR and target site (BLASTN results are included in **Supplemental Table 1**, sheet C). Thus, we used the same approach we initially used, by searching for 5'-TTAACCTT-3' sequences (with the PiggyBac target site and extremity of the left TIR), allowing one mismatch, to identify potential alternative TIRs upstream of Region S of *piwil1* genes. These results did not change our initial interpretation and are shown in our revised **Figure 2A**, which we believe is now more informative and accurate.

Figure 2E: Grey data points need to be smaller in size to reduce the overlap between adjacent points. Additionally, zoomed in plot of *piwil1.2* is missing.

We agree with the reviewer regarding the visualisation of **Figure 2E**. We have now included a revised **Figure 2E** with smaller grey points to reduce the overlap between adjacent points. In addition, we added transparency to the points.

The zoomed-in plot of *piwil1.2* is not missing. As stated in the legend of Figure 2E:

“The plots show genome-wide results of Raised Accuracy in Sweep Detection (RAiSD). μ is a metric incorporating three selective sweep signatures, with higher μ values indicative of a stronger signature of selection. **Upper panels show μ across the entire chromosome, or entire scaffold in case of *piwil1.2*.** Lower panels are insets of the *piwil1* gene regions +/- 1 Megabase (Mb). **As the entire scaffold where *piwil1.2* resides is less than 2 Mb, no inset is shown...**”

Figure S1A. TE divergence plot of subclasses for TM genome is missing.

Its non-inclusion is intentional. We did not include TE divergence plot for *Tropheops sp. 'mauve'* (TM) and *Maylandia zebra* (MZ) because these are Lake Malawi cichlids and we used the *Astatotilapia calliptera* (AC) reference genome and its TE annotations for all Lake Malawi

cichlids. This is because these fishes are closely related at the genetic level (PMID: 33197206/ and because we did have no TM and MZ genomes of similar quality to that of AC (chromosome level assembly, not significantly fragmented (https://www.ebi.ac.uk/ena/browser/view/GCA_900246225.5 and https://www.ensembl.org/Astatotilapia_calliptera/Info/Annotation). Demonstrating the genetic relatedness of these fishes, one can appreciate no difference in small RNA and mRNA mapping rates to the AC genome (**Supplemental Table 4**), in samples originating from AC, MZ or TM fishes.

Page 6, Line 12: It is unclear where this list of genes was curated from and reasoning behind including these specific genes. piRNA pathway genes are rapidly evolving and sometimes species-specific. Does the list used constitute the core-machinery expected to be widely conserved?

Indeed, piRNA pathway genes are rapidly evolving and sometimes species-specific. This list however, as the reviewer also suggests, includes the core machinery known to be conserved in animals (according to PMID: 30446728 and 35307201). To make this clearer, we have adjusted the text in that paragraph.

Page 6, Line 35: It is unclear which fishes are being referred to here? Did the authors conduct a wide search across teleost genomes for piwil1 gene copies?

This refers to our knowledge of the literature and our OrthoFinder analysis, the results of which are shown in **Supplemental Table 1** (sheet S1A). The species used for this analysis are mentioned in the methods (see *Orthology analysis* section), and in the different columns of **Supplemental Table 1** (S1A). These include twelve species of fish, plus mouse and human.

We have now added to that sentence a reference to **Supplemental Table 1**, to guide the reader to the results within the supplemental table. In addition, we cited relevant literature related to fish *piwil1* genes.

Figure 5G: How do the authors know if the testis specific cluster shown is Lake Malawai specific? And for that matter is indeed testis-specific in TM and MZ? I do not the tracks for ON for the same cluster. I also cannot find any mention of TM and MZ ovary clusters.

We apologise for the confusion, the figure legend was misleading. As indicated in the main text, the examples provided are supposed to represent large, highly productive piRNA clusters expressed in testes and/or ovaries, both in Lake Malawi cichlids and *O. niloticus*. We did not

intend to claim that these examples of clusters are testes- or species-specific. We did not sequence ovaries of TM and MZ, as no samples of these species' organs were available, so indeed within Lake Malawi we can only establish testes-specificity of piRNA cluster expression for AC, not TM and MZ. In any case, as that was not our intention, we rephrased the figure legend to make this clear.

Figure 5E-F: While TE annotation track is shown, no information regarding the TE class or subclass is shown. This will be insightful in understanding what kind of TEs are present and will serve as an example of enrichment analysis present in 6A-B.

We have added information on the TE classes in the representative clusters shown in Figure 5G-I. While we recognise that this makes the panels more informative, it does not add much to the analysis. We believe it erroneous to assume that genome-wide enrichment or depletion of particular TE classes over thousands of piRNA clusters is apparent from one or two examples of clusters (picked based on their length and productivity as mentioned in the text). As these are long piRNA clusters, it is not easily apparent which TE classes are present from the figure. We feel that searching for a piRNA cluster which perfectly illustrates the enrichment patterns present in **Figure 6A-B** would be an intense “cherry-picking” exercise and we would therefore prefer not doing it.

Page 6, Line 60: “377 high quality copies” is vague. Did the authors mean near-complete/autonomous or just that their percent divergence is low, and insertions are highly similar to the consensus?

That is a good point, it is indeed not clear from the main text how this was defined. However, we did define the criteria we used in detail in the methods section *Piwil1 evolutionary analysis*. We have now slightly altered the text in the methods for clarity, please see below.

“To expand the analysis and identify high quality copies, we extracted all the PiggyBac-1 sequences annotated in the *A. calliptera* reference genome on scaffold ≥ 1 Mbp (according to the curated TE annotation) with SWscore > 1000 (details available at https://github.com/miguelvalmeida/Cichlid_TEs_piRNAs2024), and aligned them with MUSCLE v3.8.3185. We further filtered the alignment to contain only the region encompassed by the PiggyBac-1 elements associated with *piwil1.2*, *piwil1.3*, and *piwil1.4*, and removed alignment columns consisting almost exclusively of missing data (filtered alignment available at https://github.com/miguelvalmeida/Cichlid_TEs_piRNAs2024).”

In addition, we provide the details (coordinates and other information) of the filtered high quality PiggyBac-1 TEs and alignment as a supplement in our GitHub page. Of note, upon double-checking, we amended the 377 number to 315, which was the final number of

sequences used for the alignment of **Figure S4D** (377 was the number of sequences in an intermediate filtering step).

Lastly, we amended the main text in order to direct the reader to the methods.

Figure 6: This figure contains many different observations, and some are unexpected. However, most of the figure panels are quickly skimmed in the results and not described or even discussed later. For example, while all TEs in general and LTR TEs are always enriched in clusters, LINE TEs are significantly depleted in AC and ON clusters of both gonads. Also, the bars for SINEs, RC and 3' UTR are difficult to read. I suggest authors present the y-axis values in log₁₀ increments.

As the values represent a percentage, we do not think that log₁₀ increments will be adequate visualisation. We do agree, however, that a better visualisation is required for the bars with low values on y-axis. To address this, we have now altered **Figures 6A-B** and **S10A-C** to contain the y-axis values above the bars.

We now mention the depletion of LINEs from piRNAs clusters in the text. One possible reason for this is that perhaps LINEs are silenced by specialised pathways, such as the HUSH complex (homologs for HUSH complex factors exist in cichlid fishes, see **Supplemental Table 1**, OrthoFinder results), leading to no advantage in accumulating LINEs in piRNA clusters. This is highly speculative, and we would prefer not mentioning it in the discussion section.

Page 8, Line 58: If *piwil1* is indeed heterozygous in some fish, there should be an empty site in the homologous chromosome with no *piwil1* copy. Authors can eliminate or confirm this explanation by looking for this site using *piwil1* 5' and 3' junctions from either the long or short reads.

The reviewer makes a good point. We created **Figure 2C** (summary of the results available in **Supplemental Table 1**) by manual curation of aligned short- and long-reads (.bam files) spanning the *piwi* genes in all the Lake Malawi fish samples available (what the reviewer suggested). This analysis was conducted on IGV and is described in the methods. The sentence the reviewer pointed out indeed was misleading and inaccurate. The samples used were sequenced to an adequate depth to assess heterozygosity. We have now double-checked the individuals we referred to, and as can be observed in **Revision Figure 7**, we can confidently say that the individuals are not heterozygous. We have removed that sentence in the revised version of our manuscript.

Revision Figure 7. IGV screenshots with reads supporting absence of *piwi* gene and presence of PiggyBac. (A-B) IGV screenshots of short-read (A) or long-read (B) data mapping to the *piwil1.4* locus. PiggyBac and Maverick elements flanking the gene are indicated. *A. calliptera* genome used as reference.

Page 24, Line 24: There are no stats for the mapping to genome and TE consensus provided for the small RNA libraries. A supplemental file with such details is needed to understand the quality of the libraries.

QC is integral to our analysis, and as mentioned in the methods, we use fastQC v0.11.9, as well as considering mapping to genome. We have now created and included in the manuscript **Supplemental Table 4** with mapping statistics of the small RNA and mRNA libraries. A minority of samples did not pass QC and were not included in our analysis and results (also not included in **Supplemental table 4**). As can be observed in **Supplemental Table 4**, the mapping parameters used resulted in a high percentage of mapping reads per sample, mostly > 80% for small RNAs and > 90% for mRNAs. We generally did not face technical issues with samples originating from testis, muscle, and brain. However, ovary samples were sometimes problematic, and did not pass QC, and thus were not included in this analysis. This is most likely due to the vast amount of yolk in ovary, which may interfere with RNA purification.

Of note, we did not map small RNAs or mRNAs to the TE consensus, but to the TE annotations in our reference genomes.

Page 23, Line 56: Authors state that 10 counts or more for each TE family was cutoff was TE family to be considered expressed. If I understand this correctly, this is raw read counts, which, depending on the depth of the library and multi-mappers, is a very relaxed cutoff. In addition, it would help convince readers of true TE expression if authors could add stranded normalized coverage plots for some of these TEs consensus sequences.

This is a good point, and using a fairly relaxed cutoff was the purpose here. We did not mean to equate “expression of TE families” with “high expression of TE families”. By “expression” we meant “detectable expression” of TE families. We have now altered the text referring to TE family expression accordingly to make what we mean clearer. Of note, the counts were obtained with Tetranscripts v2.2.1 (PMID: 26206304).

When we explore the percentage of TE families detected using different read cutoffs, we can observe that our chosen cutoff (at least 10 reads in at least 2 samples) locates at or near the inflection point of the sigmoidal curve (**Revision Figure 6**). This is true both for our embryonic datasets (**Revision Figure 6A**) and for our gonad datasets (**Revision Figure 6B**). Thus, we think the chosen cutoff is adequate to show “detectable TE expression” in our datasets.

Whenever we do show TE expression, we show normalised counts, obtained by running DESeq2 with the Tetranscripts counts as input (**Figures 1B, 6E-G, S1C-E, S2B-C and S11**). We now split old **Figure S1** into two (**S1** and **S2**) and include **Figures S1D** and **S2C**, detailed plots showing the expression of distinct TE families, grouped by TE superfamily (using DESeq2 normalised counts). Our data is stranded so this information is included in our data.

Figurer 3A: Are the 4 points in each tissue replicates libraries? This is not clear in the legend. Same issue with Supp Fig S1G heatmap. Are the four columns replicates? These should be made clear.

Regarding **Figures 3A** and **S4B**, the four points in each tissue are biological replicates, i.e. RNA isolated from tissues/organs of different fishes. Regarding the heatmap on old **Figure S1G** (now **Figure S2D**), the four columns of each stage represent biological replicates, i.e. RNA isolated from different embryos at the same developmental stage. We have now clarified both in the respective figure legends.

Page 26, Line 1: Is there a specific reason why STAR was used for aligning small RNA reads? STAR is better suited for paired-end and longer mRNA-seq data. Was the seed length adjusted to be half or lower than half of the mean piRNA length? If STAR used longer seed lengths (>20nt), then many potential alignments may be missed.

As the reviewers' remark, STAR was originally developed to enable mapping of paired-end and longer RNA-sequencing reads. However, STAR is adequate to map short-reads as well.

STAR does not use a fixed seed (Dobin et al., 2013, PMID: 23104886). Our parameters are based on the recommendations of Alexander Dobin, the creator of STAR, for small RNA mapping using STAR (<https://groups.google.com/g/rna-star/c/RBWvAGFooMU>). The parameters we use have been adapted to piRNAs and improved in our laboratory over the years, to maximise piRNA mappability. In fact, our collaborators in the Haase use similar parameters, which have been optimised in direct collaboration with the Dobin group. The Dobin group provides active support for STAR and regular updates that facilitate its use.

STAR is used by several laboratories in the small RNA community to align small RNAs to genomes, including Astrid Haase's (for example PMID: 34667116 and the recent work describing piCB, the program used in this work to define piRNA clusters: https://papers.ssrn.com/sol3/papers.cfm?abstract_id=4822917), Petr Svoboda's (for example PMID: 36332606 and 34489573), and previous work in the Miska lab (for example PMID: 32843637 and 36473462).

We used STAR to map piRNAs in this work because there were previously established and optimised analytical tools in our laboratory using this program. You can now find in **Supplemental Table 4** the mapping statistics of STAR v2.5.4 using filtered 24-35 nucleotides long reads with the options described in methods (`--readFilesCommand zcat --outMultimapperOrder Random --outFilterMultimapNmax 100 --outFilterMismatchNmax 2 --alignIntronMax 1 --outSAMtype BAM SortedByCoordinate --outFilterType BySJout --winAnchorMultimapNmax 100 --alignEndsType EndToEnd --scoreDelOpen -10000 --scoreInsOpen -10000 --outSAMmultNmax 1 --outFileNamePrefix`). The table shows that the average percentage of mapped reads with

these settings was ~83% and the average percentage of unmapped reads was ~17%. This shows that STAR is suitable to align piRNAs to cichlid genomes.

Second round of review

Reviewer 1

The authors have made a thorough revision of the manuscript that is strengthened and easier to read. My concerns have been addressed.

Reviewer 2

The authors have addressed all my major and minor concerns. Revised version of Almeida et al, 2024 has significant improvements in clarity and interpretation of the results presented but it is still an extremely information-dense article. As I said before, I will state again that it's potentially a landmark study for cichlid biology and will open many fundamental questions regarding piRNA pathway evolution in vertebrates, especially during speciation events. However, there are still two major areas that need clarification, perhaps reanalysis and several minor concerns.

Major concerns:

1. Lack of interpretation of Ovary sRNA profile of *Astatotilapia burtoni* (LT) (Fig 4B) and explanation for why this library has only one replicate, while others have 2-3.
 - i. In first panel of 4B, there is no standard deviation for the small RNA profile due to only one replicate. The one replicate used is seemingly a low-quality library as 'total % mapped reads' is mere 58%, while all other libraries are >80%. There may be a completely reasonable explanation for this, perhaps input tissue was damaged or it is difficult to obtain *A. burtoni* ovaries, or number of females in the population is lower than males. If the authors wish to keep this library for AB in their analysis, a reasonable explanation and discussion of these caveats is needed to make readers aware.
 - ii. There is no indication of where the peaks of miRNAs of the libraries are. Is it due to unique sequence-only representation and extensive diversity of piRNAs, the miRNA peak is weak? miRNAs exhibit very high conservation than piRNAs and are often used as internal control to evaluate these sRNA profiles in invertebrates.
 - iii. Testis small RNA profile for species PN and ON are very broad for piRNAs. Did the authors check if the most abundant 23-25 sized small RNAs aren't degraded fragments from snoRNAs or snRNAs? Perhaps if sRNA libraries are filtered from known snRNA, snoRNA, miRNA and tRNAs of zebrafish, true piRNA length profiles will emerge. If you do not selectively remove (especially tRNA-fragments that can be of 22nt and 33nt), the 24-35nt sRNAs used for piCDB might have many false-positive and skew TE-mapping analysis as well.
2. Enrichment analyses presented in figure 6A-B are unconvincing and not explained in the Methods.
 - i. First, no details of how the permutation of TE expected in (bps) was calculated is stated in the Methods. How was shuffling performed? How much cutoff in bps was required for the overlap? Are the p-values in A also calculated from Wilcoxon-Ranked sum test as in E-G. If yes, that test is not appropriate for enrichment analysis as expected independent samples.
 - ii. Second 'proportion bases' is not a good measure of enrichment since the authors want to test enrichment or depletion of TE in clusters compared to as expected by chance. All TEs irrespective of the subclass or superfamily will exhibit some amount of structural variation. By measuring proportion bases, authors add unnecessary confounding factors in their analysis-TE length. I recommend authors use insertion count instead of bases. I assume the authors perhaps were worried about the fragmented insertion calls RepeatMasker makes where copy

number of a TE may become inflated. If this is the case, it should be stated as such, at least in the Methods.

- iii. The significance shown for differences is not convincing. How is Ovary 'RC' in TE classes not-significant but the SINEs, with less difference between expected and observed is significant? The range of p-value do the 'three stars' signify is not stated in the legend. Raw values presented in 6A-B are not easy to understand enrichment or depletion.
- iv. I recommend authors look at standards of enrichment or depletion analysis done in the TE field in recent literature.

Minor concerns:

1. In Fig 4A-D, left-most, middle panels have differing scales of Y-axis values from one species to another. For a reader to understand the comparisons discussed in the text, the same scale should be used as done for the ping-pong signature plot.
2. In discussion, line 593, "we found that the majority of TE families are most highly expressed during gastrulation" should be changed to "exhibit higher expression". Since the cut-off is so low, I strongly recommend authors refrain from using highly expressed at any point for the mRNA-seq results.
3. The concept of functional role for the three TE superfamilies in lines 604-605 is highly speculative. Unless authors are referring to any specific domesticated TE family, invoking functional role for an entire superfamily, based only on the stage-specific expression is simply unsubstantiated. If the authors do intend to speculate some functionality due to studies of such families serving function in mammalian development, then those studies should be cited and mentioned before this line.
4. Line 704, citation needed.
5. In discussion, several interesting and novel findings are discussed but rarely connected to the known body of piRNA literature. Is the sex-bias in piRNA and TE activity expected? Are cichlid clusters more rapidly evolving or comparable to those reported for mammals and flies?

Authors' response to reviewers

Reviewer #2

The authors have addressed all my major and minor concerns. Revised version of Almeida et al, 2024 has significant improvements in clarity and interpretation of the results presented but it is still an extremely information-dense article. As I said before, I will state again that it's potentially a landmark study for cichlid biology and will open many fundamental questions regarding piRNA pathway evolution in vertebrates, especially during speciation events. However, there are still two major areas that need clarification, perhaps reanalysis and several minor concerns.

We thank the reviewer for having another thorough look at our paper and for the additional comments. Regarding the comment on information density, we believe our manuscript is up to current standards observed in (epi)genomics papers, including of papers published in Genome Biology. This is in agreement with Reviewer #1's assessment of our revised manuscript: "*The authors have made a thorough revision of the manuscript that is strengthened and easier to read.*"

Major concerns:

1. 1. Lack of interpretation of Ovary sRNA profile of *Astatotilapia burtoni* (LT) (Fig 4B) and explanation for why this library has only one replicate, while others have 2-3.

We address the comments in detail below. Although we have addressed the comment and think it is useful to bring up this point, this is a minor point that does not affect the outcome of the study.

i. i. In first panel of 4B, there is no standard deviation for the small RNA profile due to only one replicate. The one replicate used is seemingly a low-quality library as 'total % mapped reads' is mere 58%, while all other libraries are >80%. There may be a completely reasonable explanation for this, perhaps input tissue was damaged or it is difficult to obtain *A. burtoni* ovaries, or number of females in the population is lower than males. If the authors wish to keep this library for AB in their analysis, a reasonable explanation and discussion of these caveats is needed to make readers aware.

To address the comment, we have now included a sentence in the methods, see page 23, lines 1104-1106. Also, we make it clear in the legend of Figure 4 that there is only one replicate for *A. burtoni* ovary, see page 39, line 1869.

As we noted in the response to one of the Reviewer #2's comments in the first round of revision:

"We generally did not face technical issues with samples originating from testis, muscle, and brain. However, ovary samples were sometimes problematic, and did not pass QC, and thus were not included in this analysis. This is most likely due to the vast amount of yolk in ovary, which may interfere with RNA purification."

This, together with issues with sample availability (we do not grow *Astatotilapia burtoni* in our fish facilities) precluded the acquisition of additional replicates. This is unfortunate, but we note that our subsequent in-depth analysis were conducted only with Lake Malawi cichlids, and Nile tilapia samples, which are duly replicated and mapped to a higher quality reference genome. No conclusions of our work are based merely on *A. burtoni* ovary samples, these are used only for comparative purposes.

In summary, the inclusion of only one replicate for *A. burtoni* ovary small RNAs does not affect any conclusion in our manuscript.

ii. ii. There is no indication of where the peaks of miRNAs of the libraries are. Is it due to unique sequence-only representation and extensive diversity of piRNAs, the miRNA peak is weak? miRNAs exhibit very high conservation than piRNAs and are often used as internal control to evaluate these sRNA profiles in invertebrates.

Figures 4 and S6 have the information the reviewer is alluding to. We clarify below.

In the main text we write:

“The sRNA length distribution profiles in gonads have prominent peaks at lengths of 21-22 nucleotides, likely corresponding to microRNAs (Additional File 1: Fig. S6A). Contrary to microRNAs, piRNAs have high sequence diversity [27]. When sRNA reads are collapsed into unique sequences, we observed prominent sRNA populations between 24-31 nucleotides long, consistent with the length distribution of piRNAs (Fig. 4, left panels).”

As can be read in the legend of Figure 4A-D:

“sRNA length profiles shown here (left-most panels) comprise only reads of unique sequence.”

In the legend of Figure S6A we write:

“sRNA length distribution profiles of all reads (without collapsing into unique reads) display prominent peaks at 21-22 nucleotides, likely attributable to abundant microRNAs.”

Briefly, figure 4 shows collapsed reads of unique sequence, which are enriched for sequence diverse piRNAs and generally depleted of miRNA sequences (many copies of specified sequences). Figure S6 shows all mapped reads, which include piRNAs and miRNAs (with the latter as the more prominent populations). We did not go deep into miRNA biology as the annotations currently available are sub-par. We have plans to revisit these annotations and have a dedicated, in-depth look at miRNA populations, but this is beyond the scope of this work.

In summary, we have separate figures with information on piRNA and miRNA classes. The information is present in our manuscript and adequately contextualised in the main text and figure legends.

iii. iii. Testis small RNA profile for species PN and ON are very broad for piRNAs. Did the authors check if the most abundant 23-25 sized small RNAs aren't degraded fragments from snoRNAs or snRNAs? Perhaps if sRNA libraries are filtered from known snRNA, snoRNA, miRNA and tRNAs of zebrafish, true piRNA length profiles will emerge. If you do not

selectively remove (especially tRNA-fragments that can be of 22nt and 33nt), the 24-35nt sRNAs used for piCDB might have many false-positive and skew TE-mapping analysis as well.

This is an interesting point. However, as noted by the reviewer, we only used small RNAs between 24-35 nucleotides long for subsequent analysis. The key point here is that these 24-35 nt long small RNAs have clear piRNA signatures, as shown in the rest of Figure 4. Furthermore, the signatures of PN and ON are identical to the signatures observed in AB and Lake Malawi cichlids. Also, the piRNA cluster metrics obtained from piCB outputs are identical between Lake Malawi cichlids and ON (Figures 5E and S9D-F). It seems unlikely to us that conducting the suggested analysis with zebrafish annotations will add a qualitative improvement to our work on cichlid data. To our knowledge, there are no adequate and curated annotations for those types of non-coding RNA in cichlids.

Taken together, it is unlikely that our size selection approach leads to abundant false positives as suggested by the reviewer.

1. 2. Enrichment analyses presented in figure 6A-B are unconvincing and not explained in the Methods.

We address the comments in detail below. We apologise for not initially including the detailed methods. We have done so now. Our analysis presents statistically significant differences for some of the overlaps, as indicated in the main text.

i. i. First, no details of how the permutation of TE expected in (bps) was calculated is stated in the Methods. How was shuffling performed? How much cutoff in bps was required for the overlap? Are the p-values in A also calculated from Wilcoxon-Ranked sum test as in E-G. If yes, that test is not appropriate for enrichment analysis as expected independent samples.

Thank you for bringing this up. Indeed, we have now included a dedicated methods section with this analysis. A brief methods snippet was only described in the legend of Figure 6 (A-B) in the previous version of the manuscript.

Please see pages 25-26, lines 1238-1247 to consult the method description.

ii. ii. Second 'proportion bases' is not a good measure of enrichment since the authors want to test enrichment or depletion of TE in clusters compared to as expected by chance. All TEs irrespective of the subclass or superfamily will exhibit some amount of structural variation. By measuring proportion bases, authors add unnecessary confounding factors in their analysis- TE length. I recommend authors use insertion count instead of bases. I assume the authors perhaps were worried about the fragmented insertion calls RepeatMasker makes where copy number of a TE may become inflated. If this is the case, it should be stated as such, at least in the Methods.

We appreciate the reviewer's concerns, but we beg to differ. In our opinion, using proportion bases instead of insertion counts is the best approach.

The use of proportion of overlapping bases accounts for both variations in genomic feature length, as well as the length of piRNA clusters they overlap with. Further to this, it accounts for TE annotations that could cause the piRNA cluster-TE intersects to become inflated due to fragmented annotations if we used individual insertions as a method. As an example, in the case of a fragmented TE annotation from RepeatMasker there may be multiple short fragments annotated as separate entities in the TE annotation. However, there is a possibility these should be annotated as a single TE. In the case of using insertion counts we would find an inflated count for this region as each fragment would be counted separately. However, using the proportion of bases display the accurate level of overlap for all features at each level.

We have now added a statement in the Methods to explain why we prefer to use the proportion of bases approach, please see pages 25-26, lines 1238-1247. To elaborate slightly on the method, the expected distributions (shown in orange in the figures) were estimated by calculating this overlap after randomly shuffling the piRNA cluster coordinates within $\pm 18,000$ bp of their original positions, as opposed to having them shuffled randomly across the genome.

iii. The significance shown for differences is not convincing. How is Ovary 'RC' in TE classes not-significant but the SINEs, with less difference between expected and observed is significant? The range of p-value do the 'three stars' signify is not stated in the legend. Raw values presented in 6A-B are not easy to understand enrichment or depletion.

In terms of the significance, we used the following convention:

* $0.01 \leq p\text{-value} < 0.05$

** $0.001 \leq p\text{-value} < 0.01$

*** $p\text{-value} < 0.001$

We have now added this to figure legends throughout the manuscript for clarity. Regarding the ovary RC not being significant (in AC), that is due to the proportion of the genome attributable to RC versus the higher fraction of the genome attributable to SINE elements.

2. iv. I recommend authors look at standards of enrichment or depletion analysis done in the TE field in recent literature.

As explained above, we believe we conducted the analysis to a robust standard.

Minor concerns:

iv. 1. In Fig 4A-D, left-most, middle panels have differing scales of Y-axis values from one species to another. For a reader to understand the comparisons discussed in the text, the same scale should be used as done for the ping-pong signature plot.

The important aspects that we wish to highlight in these panels are the profiles of the small RNA length distribution and ping-pong signatures in each species. The differences in Y-axis between the panels are irrelevant to the point we wish to convey. We would prefer to maintain Figure 4 as is.

v. 2. In discussion, line 593, “we found that the majority of TE families are most highly expressed during gastrulation” should be changed to “exhibit higher expression”. Since the cut-off is so low, I strongly recommend authors refrain from using highly expressed at any point for the mRNA-seq results.

We changed the text as suggested, please see page 12, line 592.

2. 3. The concept of functional role for the three TE superfamilies in lines 604-605 is highly speculative. Unless authors are referring to any specific domesticated TE family, invoking functional role for an entire superfamily, based only on the stage-specific expression is simply unsubstantiated. If the authors do intend to speculate some functionality due to studies of such families serving function in mammalian development, then those studies should be cited and mentioned before this line.

We did not mean to imply a functional role for the **entire** superfamilies. We have now amended the text to reflect this, please see page 13, line 604. This part of the discussion is indeed speculation, but we believe that is an adequate level of speculation to the discussion section and is not overly speculative.

3. 4. Line 704, citation needed.

Thank you, relevant citations were added.

4. 5. In discussion, several interesting and novel findings are discussed but rarely connected to the known body of piRNA literature. Is the sex-bias in piRNA and TE activity expected? Are cichlid clusters more rapidly evolving or comparable to those reported for mammals and flies?

Thank you. We preferred to focus on points we believe are more relevant to contextualise our results in the literature.

We avoided an in-depth dive in the sex-bias in piRNA/TE activity as a dedicated section would perhaps be warranted in the discussion, to introduce the topic and discuss our results in this light. Recent studies in *Drosophila* found sex-specific changes in piRNAs that match sex-specific TE expression (PMID: 33985970). We are unaware of similar authoritative studies in vertebrates (the conclusions of golden hamster papers argue for similarly important role of piRNAs in males and females). What we observe is sex-specific differences in TE expression that are not matched by sex-specific piRNA expression differences. The sex-specific differences in TE expression seem to follow a general transcriptional trend also observable in protein-coding genes (higher expression in testes, Fig. S1E). As we note in the discussion: "This asymmetry is likely to be the result of overall higher transcriptional output in testes". At this point we have no reason to believe that piRNAs are driving the sex-specific TE expression pattern. Thus, we would prefer not creating a discussion section on this topic, which would detract from the main key points we want to convey and bring unnecessary complexity and nuance to the discussion.

We also avoided doing an in-depth direct comparison with mammalian and fly piRNA clusters because we felt these are beyond the scope of this paper. Due to difficulties in dealing with multi-mapping piRNAs and subsequent definition of piRNA clusters (extensively discussed in the responses to the reviewer's comments in the first round of revision), different studies have used different methods & metrics to identify piRNA clusters. Hence, merely comparing published numbers of piRNA clusters in different species, obtained via substantially different methodologies, does not produce meaningful comparisons. To do this adequately, a careful

meta-analysis is required, in order to identify piRNA clusters in different species using the same computational analysis.